# HEAVY-TAILED DIFFUSION MODELS

**Kushagra Pandey**[1,2]*, **Jaideep Pathak**[1], **Yilun Xu**[1],
**Stephan Mandt**[2], **Michael Pritchard**[1,2], **Arash Vahdat**[1†], **Morteza Mardani**[1†]
NVIDIA[1], University of California, Irvine [2]
{pandeyk1,mandt}@uci.edu,
{jpathak,yilunx,mpritchard,avahdat,mmardani}@nvidia.com

## ABSTRACT

Diffusion models achieve state-of-the-art generation quality across many applications, but their ability to capture rare or extreme events in heavy-tailed distributions remains unclear. In this work, we show that traditional diffusion and flow-matching models with standard Gaussian priors fail to capture heavy-tailed behavior. We address this by repurposing the diffusion framework for heavy-tail estimation using multivariate Student-t distributions. We develop a tailored perturbation kernel and derive the denoising posterior based on the conditional Student-t distribution for the backward process. Inspired by $\gamma$-divergence for heavy-tailed distributions, we derive a training objective for heavy-tailed denoisers. The resulting framework introduces controllable tail generation using only a single scalar hyperparameter, making it easily tunable for diverse real-world distributions. As specific instantiations of our framework, we introduce *t-EDM* and *t-Flow*, extensions of existing diffusion and flow models that employ a Student-t prior. Remarkably, our approach is readily compatible with standard Gaussian diffusion models and requires only minimal code changes. Empirically, we show that our *t-EDM* and *t-Flow* outperform standard diffusion models in heavy-tail estimation on high-resolution weather datasets in which generating rare and extreme events is crucial.

## 1 INTRODUCTION

In many real-world applications, such as weather forecasting, rare or extreme events—like hurricanes or heatwaves—can have disproportionately larger impacts than more common occurrences. Therefore, building generative models capable of accurately capturing these extreme events is critically important (Gründemann et al., 2022). However, learning the distribution of such data from finite samples is particularly challenging, as the number of empirically observed tail events is typically small, making accurate estimation difficult.

One promising approach is to use heavy-tailed distributions, which allocate more density to the tails than light-tailed alternatives. In popular generative models like Normalizing Flows (Rezende & Mohamed, 2016) and Variational Autoencoders (VAEs) (Kingma & Welling, 2022), recent works address heavy-tail estimation by learning a mapping from a heavy-tailed prior to the target distribution (Jaini et al., 2020; Kim et al., 2024).

While these works advocate for heavy-tailed base distributions, their application to real-world, high-dimensional datasets remains limited, with empirical results focused on small-scale or toy datasets. In contrast, diffusion models (Ho et al., 2020; Song et al., 2020; Lipman et al., 2023) have demonstrated excellent synthesis quality in large-scale applications. However, it is unclear whether diffusion models with Gaussian priors can effectively model heavy-tailed distributions without significant modifications.

---

*Work during an internship at NVIDIA

†Equal Advising

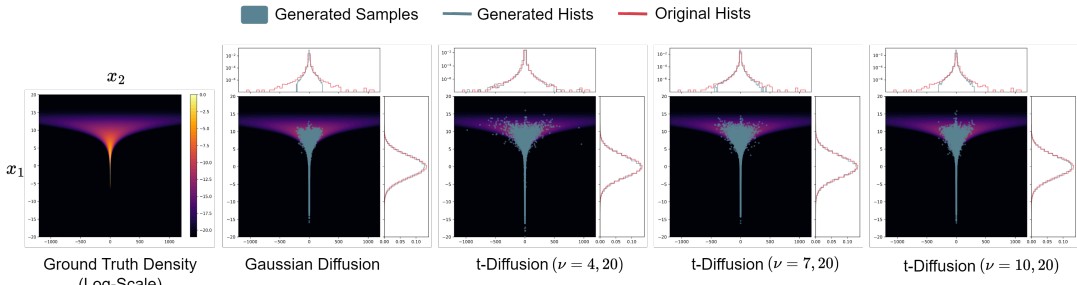

Figure 1: **Toy Illustration.** Our proposed diffusion model (*t-Diffusion*) captures heavy-tailed behavior more accurately than standard Gaussian diffusion (see histogram comparisons in the top panel, x-axis). The framework allows for **controllable tail estimation** using a hyperparameter $\nu$, which can be adjusted for each dimension. Lower $\nu$ values model heavier tails, while higher values approach Gaussian diffusion (Best viewed when zoomed in; see App. C.3 for details). Brighter colors indicate high-density regions

.

In this work, we first demonstrate through extensive experiments that traditional diffusion models—even with proper normalization, preconditioning, and noise schedule design (see Section 4)—fail to accurately capture the heavy-tailed behavior in target distributions (see Fig. 1 for a toy example). We hypothesize that, in high-dimensional spaces, the Gaussian distribution in standard diffusion models tends to concentrate on a spherical narrow shell, thereby neglecting the tails. To address this, we adopt the multivariate Student-t distribution as the base noise distribution, with its degrees of freedom providing controllability over tail estimation. Consequently, we reformulate the denoising diffusion framework using multivariate Student-t distributions by designing a tailored perturbation kernel and deriving the corresponding denoiser. Moreover, we draw inspiration from the $\gamma$-power Divergences (Eguchi, 2021; Kim et al., 2024) for heavy-tailed distributions to formulate the learning problem for our heavy-tailed denoiser.

We extend widely adopted diffusion models, such as EDM (Karras et al., 2022) and straight-line flows (Lipman et al., 2023; Liu et al., 2022), by introducing their Student-t counterparts: *t-EDM* and *t-Flow*. We derive the corresponding SDEs and ODEs for modeling heavy-tailed distributions. Through extensive experiments on the HRRR dataset (Dowell et al., 2022), train both unconditional and conditional versions of these models. The results show that standard EDM struggles to capture tails and extreme events, whereas *t-EDM* performs significantly better in modeling such phenomena. To summarize, we present,

- **Heavy-tailed Diffusion Models.** We repurpose the diffusion model framework for heavy-tail estimation by formulating both the forward and reverse processes using multivariate Student-t distributions. The denoiser is learned by minimizing the $\gamma$-power divergence (Kim et al., 2024) between the forward and reverse posteriors.

- **Continuous Counterparts.** We derive continuous formulations for heavy-tailed diffusion models and provide a principled approach to constructing ODE and SDE samplers. This enables the instantiation of *t-EDM* and *t-Flow* as heavy-tailed alternatives to standard diffusion and flow models.

- **Empirical Results.** Experiments on the HRRR dataset (Dowell et al., 2022), a high-resolution dataset for weather modeling, show that *t-EDM* significantly outperforms EDM in capturing tail distributions for both unconditional and conditional tasks.

- **Theoretical Connections.** To theoretically justify the effectiveness of our approach, we present several theoretical connections between our framework and existing work in diffusion models and robust statistical estimators (Futami et al., 2018).

## 2 BACKGROUND

As prerequisites underlying our method, we briefly summarize Gaussian diffusion models (as introduced in (Ho et al., 2020; Sohl-Dickstein et al., 2015)) and multivariate Student-t distributions.

### 2.1 DIFFUSION MODELS

Diffusion models define a *forward process* (usually with an affine drift and no learnable parameters) to convert data $\mathbf{x}_0 \sim p(\mathbf{x}_0), \mathbf{x}_0 \in \mathbb{R}^d$ to noise. A learnable *reverse* process is then trained to generate data from noise. In the discrete-time setting, the training objective for diffusion models can be specified as,

$$\mathbb{E}_q\bigg[ \underbrace{D_{\mathrm{KL}}(q(\mathbf{x}_T|\mathbf{x}_0) \parallel p(\mathbf{x}_T))}_{L_T} + \sum_{t>\Delta t} \underbrace{D_{\mathrm{KL}}(q(\mathbf{x}_{t-\Delta t}|\mathbf{x}_t, \mathbf{x}_0) \parallel p_\theta(\mathbf{x}_{t-\Delta t}|\mathbf{x}_t))}_{L_{t-1}} - \underbrace{\log p_\theta(\mathbf{x}_0|\mathbf{x}_{\Delta t})}_{L_0} \bigg], \quad (1)$$

where $T$ denotes the trajectory length while $\Delta t$ denotes the time increment between two consecutive time points. $D_{\mathrm{KL}}$ denotes the Kullback-Leibler (KL) divergence defined as, $D_{\mathrm{KL}}(q \parallel p) = \int q(\mathbf{x}) \log \frac{q(\mathbf{x})}{p(\mathbf{x})} d\mathbf{x}$. In the objective in Eq. 1, the trajectory length $T$ is chosen to match the generative prior $p(\mathbf{x}_T)$ and the forward marginal $q(\mathbf{x}_T|\mathbf{x}_0)$. The second term in Eqn. 1 proposes to minimize the KL divergence between the forward posterior $q(\mathbf{x}_{t-\Delta t}|\mathbf{x}_t, \mathbf{x}_0)$ and the learnable posterior $p_\theta(\mathbf{x}_{t-\Delta t}|\mathbf{x}_t)$ which corresponds to learning the *denoiser* (i.e. predicting a less noisy state from noise). The forward marginals, posterior, and reverse posterior are modeled using Gaussian distributions, which exhibit an analytical form of the KL divergence. The discrete-time diffusion framework can also be extended to the continuous time setting (Song et al., 2020; 2021; Karras et al., 2022). Recently, Lipman et al. (2023); Albergo et al. (2023) proposed stochastic interpolants (or flows), which allow flexible transport between two arbitrary distributions.

### 2.2 STUDENT-T DISTRIBUTIONS

The multivariate Student-t distribution $t_d(\mu, \mathbf{\Sigma}, \nu)$ with dimensionality d, location $\mu$, scale matrix $\mathbf{\Sigma}$ and degrees of freedom $\nu$ is defined as,

$$t_d(\mu, \mathbf{\Sigma}, \nu) = C_{\nu,d}\Big[1 + \frac{1}{\nu}(\mathbf{x} - \boldsymbol{\mu})^\top \mathbf{\Sigma}^{-1}(\mathbf{x} - \boldsymbol{\mu})\Big]^{-\frac{\nu+d}{2}}, \quad (2)$$

where $C_{\nu,d}$ is the normalizing factor. Since the multivariate Student-t distribution has polynomially decaying density, it can model heavy-tailed distributions. Interestingly, for $\nu = 1$, the Student-t distribution is analogous to the Cauchy distribution. As $\nu \to \infty$, the Student-t distribution converges to the Gaussian distribution. A Student-t distributed random variable $\mathbf{x}$ can be reparameterized as (Andrews & Mallows, 1974), $\mathbf{x} = \boldsymbol{\mu} + \mathbf{\Sigma}^{1/2}\mathbf{z}/\sqrt{\kappa}$, with $\mathbf{z} \sim \mathcal{N}(0, \boldsymbol{I}_d), \kappa \sim \chi^2(\nu)/\nu$ where $\chi^2$ denotes the Chi-squared distribution (See Fig. 8 for an illustration of the pdf of the $\chi^2$ distribution).

## 3 HEAVY-TAILED DIFFUSION MODELS

We now repurpose standard diffusion models using multivariate Student-t distributions. The main idea behind our design is the use of heavy-tailed generative priors (Jaini et al., 2020; Kim et al., 2024) for learning a transport map towards a potentially heavy-tailed target distribution. From Eqn. 1 we note three key requirements for training diffusion models: choice of the perturbation kernel $q(\mathbf{x}_t|\mathbf{x}_0)$, form of the target denoising posterior $q(\mathbf{x}_{t-\Delta t}|\mathbf{x}_t, \mathbf{x}_0)$ and the parameterization of the learnable reverse posterior $p_\theta(\mathbf{x}_{t-1}|\mathbf{x}_t)$. Therefore, we begin our discussion in the context of discrete-time diffusion models and later extend to the continuous regime. This has several advantages in terms of highlighting these three key design choices, which

might be obscured by the continuous-time framework of defining a forward and a reverse SDE (Karras et al., 2022) while at the same time leading to a simpler construction. Lastly, without loss of generality, we assume a scalar $\nu$ for subsequent analysis due to mathematical convenience.

### 3.1 Noising Process Design with Student-t Distributions.

Our construction of the noising process involves three key steps.

1. Firstly, given two consecutive noisy states $\mathbf{x}_t$ and $\mathbf{x}_{t-\Delta t}$, we specify a joint distribution $q(\mathbf{x}_t, \mathbf{x}_{t-\Delta t}|\mathbf{x}_0)$.

2. Secondly, given $q(\mathbf{x}_t, \mathbf{x}_{t-\Delta t}|\mathbf{x}_0)$, we construct the perturbation kernel $q(\mathbf{x}_t|\mathbf{x}_0) = \int q(\mathbf{x}_t, \mathbf{x}_{t-\Delta t}|\mathbf{x}_0)d\mathbf{x}_{t-\Delta t}$ which can be used as a noising process during training.

3. Lastly, from Steps 1 and 2, we construct the forward denoising posterior $q(\mathbf{x}_{t-\Delta t}|\mathbf{x}_t, \mathbf{x}_0) = \frac{q(\mathbf{x}_t, \mathbf{x}_{t-\Delta t}|\mathbf{x}_0)}{q(\mathbf{x}_t|\mathbf{x}_0)}$. We will later utilize the form of $q(\mathbf{x}_{t-\Delta t}|\mathbf{x}_t, \mathbf{x}_0)$ to parameterize the reverse posterior.

It is worth noting that our construction of the noising process bypasses the specification of the forward transition kernel $q(\mathbf{x}_t|\mathbf{x}_{t-\Delta t})$. This has the advantage that we can directly specify the form of the perturbation kernel parameters $\mu_t$ and $\sigma_t$ as in Karras et al. (2022) unlike Song et al. (2020); Ho et al. (2020). We next highlight the noising process construction in more detail.

**Specifiying the joint distribution** $q(\mathbf{x}_t, \mathbf{x}_{t-\Delta t}|\mathbf{x}_0)$**.** We parameterize the joint distribution $q(\mathbf{x}_t, \mathbf{x}_{t-\Delta t}|\mathbf{x}_0)$ as a multivariate Student-t distribution with the following form,

$$q(\mathbf{x}_t, \mathbf{x}_{t-\Delta t}|\mathbf{x}_0) = t_{2d}(\boldsymbol{\mu}, \boldsymbol{\Sigma}, \nu), \quad \boldsymbol{\mu} = [\mu_t; \mu_{t-\Delta t}]\mathbf{x}_0, \quad \boldsymbol{\Sigma} = \begin{pmatrix} \sigma_t^2 & \sigma_{12}^2(t) \\ \sigma_{21}^2(t) & \sigma_{t-\Delta t}^2 \end{pmatrix} \otimes \boldsymbol{I}_d, \quad (3)$$

where $\mu_t, \sigma_t, \sigma_{12}(t), \sigma_{21}(t)$ are time-dependent scalar design parameters. While the choice of the parameters $\mu_t$ and $\sigma_t$ determines the perturbation kernel used during training, the choice of $\sigma_{12}(t)$ and $\sigma_{21}(t)$ can affect the ODE/SDE formulation for the denoising process and will be clarified when discussing sampling.

**Constructing the perturbation kernel** $q(\mathbf{x}_t|\mathbf{x}_0)$**:** Given the joint distribution $q(\mathbf{x}_t, \mathbf{x}_{t-\Delta t}|\mathbf{x}_0)$ specified as a multivariate Student-t distribution, it follows that the perturbation kernel distribution $q(\mathbf{x}_t|\mathbf{x}_0)$ is also a Student-t distribution (Ding, 2016) parameterized as, $q(\mathbf{x}_t|\mathbf{x}_0) = t_d(\mu_t\mathbf{x}_0, \sigma_t^2\boldsymbol{I}_d, \nu)$ (proof in App. A.1). We choose the scalar coefficients $\mu_t$ and $\sigma_t$ such that the perturbation kernel at time $t = T$ converges to a standard Student-t distribution. Later, we will set our generative prior $p(\mathbf{x}_T) = q(\mathbf{x}_T|\mathbf{x}_0) = t_d(0, \boldsymbol{I}_d, \nu)$ to instantiate sample generation. We discuss practical choices of $\mu_t$ and $\sigma_t$ in Section 3.5.

**Estimating the reference denoising posterior.** Given the joint distribution $q(\mathbf{x}_t, \mathbf{x}_{t-\Delta t}|\mathbf{x}_0)$ and the perturbation kernel $q(\mathbf{x}_t|\mathbf{x}_0)$, the denoising posterior can be specified as (see Ding (2016)),

$$q(\mathbf{x}_{t-\Delta t}|\mathbf{x}_t, \mathbf{x}_0) = t_d(\bar{\boldsymbol{\mu}}_t, \frac{\nu + d_1}{\nu + d}\bar{\sigma}_t^2\boldsymbol{I}_d, \nu + d), \quad (4)$$

$$\bar{\boldsymbol{\mu}}_t = \mu_{t-\Delta t}\mathbf{x}_0 + \frac{\sigma_{21}^2(t)}{\sigma_t^2}(\mathbf{x}_t - \mu_t\mathbf{x}_0), \qquad \bar{\sigma}_t^2 = \left[\sigma_{t-\Delta t}^2 - \frac{\sigma_{21}^2(t)\sigma_{12}^2(t)}{\sigma_t^2}\right], \quad (5)$$

where $d_1 = \frac{1}{\sigma_t^2}\|\mathbf{x}_t - \mu_t\mathbf{x}_0\|^2$. Next, we formulate the training objective for heavy-tailed diffusions.

### 3.2 Parameterization of the Reverse Posterior

Following Eqn. 4, we parameterize the reverse (or the denoising) posterior distribution as:

$$p_{\boldsymbol{\theta}}(\mathbf{x}_{t-\Delta t}|\mathbf{x}_t) = t_d(\boldsymbol{\mu}_{\boldsymbol{\theta}}(\mathbf{x}_t, t), \bar{\sigma}_t^2\boldsymbol{I}_d, \nu + d), \quad (6)$$

where the denoiser mean $\boldsymbol{\mu_\theta}(\mathbf{x}_t, t)$ is further parameterized as follows:

$$\boldsymbol{\mu_\theta}(\mathbf{x}_t, t) = \frac{\sigma_{21}^2(t)}{\sigma_t^2}\mathbf{x}_t + \left[\mu_{t-\Delta t} - \frac{\sigma_{21}^2(t)}{\sigma_t^2}\mu_t\right]\boldsymbol{D}_\theta(\mathbf{x}_t, \sigma_t). \tag{7}$$

We discuss alternative posterior parameterizations in App. A.2. It is worth noting that while the noising process defined in Eq. 3 is non-markovian, our parameterization of the posterior is still Markovian. However, similar to Song et al. (2022a), this choice works well empirically (see Section 4). Moreover, when parameterizing the reverse posterior scale, we drop the data-dependent coefficient $(\nu + d_1)/(\nu + d)$. This choice is primarily inspired by simplicity in deriving preconditioners (Sec. 3.5) and developing continuous-time sampling methods (Sec. 3.4) for heavy-tailed diffusions, resulting in models that require minimal implementation overhead over standard diffusion models during training and sampling (see Fig. 2). However, heteroskedastic modeling of the denoiser is possible in our framework and could be an interesting direction for future work. Next, we reformulate the training objective in Eqn. 1 for heavy-tailed diffusions.

### 3.3 TRAINING WITH POWER DIVERGENCES

The optimization objective in Eqn. 1 primarily minimizes the KL-Divergence between a given pair of distributions. However, since we parameterize the distributions in Eqn. 1 using multivariate Student-t distributions, using the KL-Divergence might not be a suitable choice of divergence. This is because computing the KL divergence for Student-t distributions does not exhibit a closed-form expression. An alternative is the $\gamma$-Power Divergence (Eguchi, 2021; Kim et al., 2024) defined as,

$$D_\gamma(q \parallel p) = \frac{1}{\gamma}\big[\mathcal{C}_\gamma(q,p) - \mathcal{H}_\gamma(q)\big], \quad \gamma \in (-1,0) \cup (0,\infty)$$

$$\mathcal{H}_\gamma(p) = -\|p\|_{1+\gamma} = -\left(\int p(\mathbf{x})^{1+\gamma}d\mathbf{x}\right)^{\frac{1}{1+\gamma}} \qquad \mathcal{C}_\gamma(q,p) = -\int q(\mathbf{x})\left(\frac{p(\mathbf{x})}{\|p\|_{1+\gamma}}\right)^\gamma d\mathbf{x},$$

where, like Kim et al. (2024), we set $\gamma = -\frac{2}{\nu+d}$ for the remainder of our discussion. Moreover, $\mathcal{H}_\gamma$ and $\mathcal{C}_\gamma$ represent the $\gamma$-power entropy and cross-entropy, respectively. Interestingly, the $\gamma$-Power divergence between two multivariate Student-t distributions, $q_\nu = t_d(\boldsymbol{\mu}_0, \boldsymbol{\Sigma}_0, \nu)$ and $p_\nu = t_d(\boldsymbol{\mu}_1, \boldsymbol{\Sigma}_1, \nu)$, can be tractably computed in closed form and is defined as (see Kim et al. (2024) for a proof),

$$\begin{aligned}
\mathcal{D}_\gamma[q_\nu \| p_\nu] = -\frac{1}{\gamma}C_{\nu,d}^{\frac{\gamma}{1+\gamma}}&\left(1 + \frac{d}{\nu-2}\right)^{-\frac{\gamma}{1+\gamma}}\left[-|\boldsymbol{\Sigma}_0|^{-\frac{\gamma}{2(1+\gamma)}}\left(1 + \frac{d}{\nu-2}\right)\right. \\
&\left. + |\boldsymbol{\Sigma}_1|^{-\frac{\gamma}{2(1+\gamma)}}\left(1 + \frac{1}{\nu-2}\mathrm{tr}\left(\boldsymbol{\Sigma}_1^{-1}\boldsymbol{\Sigma}_0\right) + \frac{1}{\nu}(\boldsymbol{\mu}_0 - \boldsymbol{\mu}_1)^\top\boldsymbol{\Sigma}_1^{-1}(\boldsymbol{\mu}_0 - \boldsymbol{\mu}_1)\right)\right].
\end{aligned} \tag{8}$$

Therefore, analogous to Eqn. 1, we minimize the following optimization objective,

$$\mathbb{E}_q\left[D_\gamma(q(\mathbf{x}_T|\mathbf{x}_0) \parallel p(\mathbf{x}_T)) + \sum_{t>1} D_\gamma(q(\mathbf{x}_{t-\Delta t}|\mathbf{x}_t, \mathbf{x}_0) \parallel p_\theta(\mathbf{x}_{t-\Delta t}|\mathbf{x}_t)) - \log p_\theta(\mathbf{x}_0|\mathbf{x}_1)\right]. \tag{9}$$

Here, we note a couple of caveats. Firstly, while replacing the KL-Divergence with the $\gamma$-Power Divergence in the objective in Eqn. 1 might appear to be due to computational convenience, the $\gamma$-power divergence has several connections with robust estimators (Futami et al., 2018) in statistics and provides a tunable parameter $\gamma$ which can be used to control the model density assigned at the tail (see Section 5). Secondly, while the objective in Eqn. 1 is a valid ELBO, the objective in Eq. 9 is not. However, the following result provides a connection between the two objectives (see proof in App. A.3),

**Proposition 1.** *For arbitrary distributions $q$ and $p$, in the limit of $\gamma \to 0$, $D_\gamma(q \parallel p)$ converges to $D_{\mathrm{KL}}(q \parallel p)$. Consequently, for a finite-dimensional dataset with $\mathbf{x}_0 \in \mathbb{R}^d$ and $\gamma = -\frac{2}{\nu+d}$, under the limit of $\gamma \to 0$, the objective in Eqn. 9 converges to the objective in Eqn. 1.*

| Component | Gaussian Diffusion | (Ours) t-Diffusion |
|---|---|---|
| Perturbation Kernel ($q(\mathbf{x}_t\|\mathbf{x}_0)$) | $\mathcal{N}(\mu_t\mathbf{x}_0, \sigma_t^2 \boldsymbol{I}_d)$ | $t_d(\mu_t\mathbf{x}_0, \sigma_t^2 \boldsymbol{I}_d, \nu)$ |
| Forward Posterior ($q(\mathbf{x}_{t-\Delta t}\|\mathbf{x}_t, \mathbf{x}_0)$) | $\mathcal{N}(\bar{\boldsymbol{\mu}}_t, \bar{\sigma}_t^2 \boldsymbol{I}_d)$ | $t_d(\bar{\boldsymbol{\mu}}_t, \frac{\nu+d_1}{\nu+d}\bar{\sigma}_t^2 \boldsymbol{I}_d, \nu + d)$ |
| Reverse Posterior ($p_{\boldsymbol{\theta}}(\mathbf{x}_{t-\Delta t}\|\mathbf{x}_t)$) | $\mathcal{N}(\boldsymbol{\mu_\theta}(\mathbf{x}_t, t), \bar{\sigma}_t^2 \boldsymbol{I}_d)$ | $t_d(\boldsymbol{\mu_\theta}(\mathbf{x}_t, t), \bar{\sigma}_t^2 \boldsymbol{I}_d, \nu + d)$ |
| Divergence Measure | $D_{\mathrm{KL}}(q \, \| \, p)$ | $D_\gamma(q \, \| \, p)$ |
| Generative Prior ($p(\mathbf{x}_T)$) | $\mathcal{N}(0, \boldsymbol{I}_d)$ | $t_d(0, \boldsymbol{I}_d, \nu)$ |

Table 1: Comparison between different modeling components for constructing Gaussian vs Heavy-Tailed diffusion models. Under the limit of $\nu \to \infty$, our proposed t-Diffusion framework converges to Gaussian diffusion models.

Therefore, under the limit $\gamma \to 0$, the standard diffusion model framework becomes a special case of our proposed framework. Moreover, for $\gamma = -2/(\nu + d)$, this also explains the tail estimation moving towards Gaussian diffusion for an increasing $\nu$ (See Fig. 1 for an illustration). Lastly, the divergence-based interpretation of the ELBO loss in Eq. 1 has also been considered in prior work in generative models (Xiao et al.; Kim et al., 2024) and is also commonplace in M-estimators (Futami et al., 2018) used in robust statistics. Therefore, our choice of the training objective in Eq. 9 is quite principled.

**Simplifying the Training Objective.** Plugging the form of the forward posterior $q(\mathbf{x}_{t-\Delta t}|\mathbf{x}_t, \mathbf{x}_0)$ in Eqn. 4, the reverse posterior $p_{\boldsymbol{\theta}}(\mathbf{x}_{t-\Delta t}|\mathbf{x}_t)$ in the optimization objective in Eqn. 9, we obtain the following simplified training loss (proof in App. A.4),

$$\mathcal{L}(\theta) = \mathbb{E}_{\mathbf{x}_0 \sim p(\mathbf{x}_0)} \mathbb{E}_{t \sim p(t)} \mathbb{E}_{\epsilon \sim \mathcal{N}(0, \boldsymbol{I}_d)} \mathbb{E}_{\kappa \sim \frac{1}{\nu} \chi^2(\nu)} \left\| \boldsymbol{D_\theta}\left(\mu_t\mathbf{x}_0 + \sigma_t \frac{\epsilon}{\sqrt{\kappa}}, \sigma_t\right) - \mathbf{x}_0 \right\|_2^2. \tag{10}$$

Intuitively, the form of our training objective is similar to existing diffusion models (Ho et al., 2020; Karras et al., 2022). However, the only difference lies in sampling the noisy state $\mathbf{x}_t$ from a Student-t distribution based perturbation kernel instead of a Gaussian distribution. Next, we discuss sampling from our proposed framework under discrete and continuous-time settings.

## 3.4 SAMPLING

**Discrete-time Sampling.** For discrete-time settings, we can simply perform ancestral sampling from the learned reverse posterior distribution $p_{\boldsymbol{\theta}}(\mathbf{x}_{t-\Delta t}|\mathbf{x}_t)$. Therefore, following simple re-parameterization, an ancestral sampling update can be specified as,

$$\mathbf{x}_{t-\Delta t} = \boldsymbol{\mu_\theta}(\mathbf{x}_t, t) + \bar{\sigma}_t\mathbf{z}_t/\sqrt{\kappa_t}, \quad \mathbf{z} \sim \mathcal{N}(0, \boldsymbol{I}_d), \quad \kappa \sim \chi^2(\nu + d)/(\nu + d),$$

**Continuous-time Sampling.** Due to recent advances in accelerating sampling in continuous-time diffusion processes (Pandey et al., 2024a; Zhang & Chen, 2023; Lu et al., 2022; Song et al., 2022a; Xu et al., 2023a), we reformulate discrete-time dynamics in heavy-tailed diffusions to the continuous time regime. More specifically, we present a family of continuous-time processes in the following result (Proof in App. A.5).

**Proposition 2.** *The posterior parameterization $p_{\boldsymbol{\theta}}(\mathbf{x}_{t-\Delta t}|\mathbf{x}_t) = t_d(\boldsymbol{\mu_\theta}(\mathbf{x}_t, t), \bar{\sigma}_t^2 \boldsymbol{I}_d, \nu + d)$ induces the following continuous-time dynamics,*

$$d\mathbf{x}_t = \left[ \frac{\dot{\mu}_t}{\mu_t}\mathbf{x}_t - \left[ f(\sigma_t, \dot{\sigma}_t) + \frac{\dot{\mu}_t}{\mu_t} \right] (\mathbf{x}_t - \mu_t \boldsymbol{D_\theta}(\mathbf{x}_t, \sigma_t)) \right] dt + \sqrt{\beta(t)g(\sigma_t, \dot{\sigma}_t)}d\boldsymbol{S}_t, \tag{11}$$

*where $f : \mathbb{R}^+ \times \mathbb{R}^+ \to \mathbb{R}$ and $g : \mathbb{R}^+ \times \mathbb{R}^+ \to \mathbb{R}^+$ are user-specified functions, $\beta_t \in \mathbb{R}^+$ is a scaling coefficient such that $\frac{1}{\sigma_{12}^2(t)}\left(\sigma_{t-\Delta t}^2 - \beta(t)g(\sigma_t, \dot{\sigma}_t)\Delta t\right) - 1 = f(\sigma_t, \dot{\sigma}_t)\Delta t,$, where $\dot{\mu}_t, \dot{\sigma}_t$ denote the first-order time-derivatives of the perturbation kernel parameters $\mu_t$ and $\sigma_t$ respectively and the differential $d\boldsymbol{S}_t \sim t_d(0, dt, \nu + d)$.*

**Algorithm 1: Training (t-EDM)**

1: **repeat**
2: $\quad \mathbf{x}_0 \sim p(\mathbf{x}_0)$
3: $\quad \sigma \sim \text{LogNormal}(\pi_{\text{mean}}, \pi_{\text{std}})$
4: $\quad \mathbf{x} = \mathbf{x}_0 + \mathbf{n}, \quad \mathbf{n} \sim t_d(0, \sigma^2 \mathbf{I}_d, \nu)$
5: $\quad \sigma = \sigma\sqrt{\nu/(\nu-2)}$
6: $\quad D_\theta(\mathbf{x}, \sigma) =$
     $\quad c_{\text{skip}}(\sigma)\mathbf{x} + c_{\text{out}}(\sigma)F_\theta(c_{\text{in}}(\sigma)\mathbf{x}, c_{\text{noise}}(\sigma))$
7: $\quad \lambda(\sigma) = c_{\text{out}}^{-2}(\sigma)$
8: $\quad$ Take gradient descent step on
9: $\qquad \nabla_{\boldsymbol{\theta}}\Big[\lambda(\sigma)\|D_\theta(\mathbf{x}, \sigma) - \mathbf{x}_0\|^2\Big]$
10: **until** converged

**Algorithm 2: Sampling (t-EDM) ($\mu_t = 1, \sigma_t = t$)**

1: **sample $\mathbf{x}_0 \sim t_d\big(0, t_0^2\mathbf{I}_d, \nu\big)$**
2: **for** $i \in \{0, \dots, N-1\}$ **do**
3: $\quad \boldsymbol{d}_i \leftarrow \big(\mathbf{x}_i - D_{\boldsymbol{\theta}}(\mathbf{x}_i; t_i)\big)/t_i$
4: $\quad \mathbf{x}_{i+1} \leftarrow \mathbf{x}_i + (t_{i+1} - t_i)\boldsymbol{d}_i$
5: $\quad$ **if** $t_{i+1} \neq 0$ **then**
6: $\qquad \boldsymbol{d}_i \leftarrow \big(\mathbf{x}_{i+1} - D_{\boldsymbol{\theta}}(\mathbf{x}_{i+1}; t_{i+1})\big)/t_{i+1}$
7: $\qquad \mathbf{x}_{i+1} \leftarrow \mathbf{x}_i + (t_{i+1} - t_i)\big(\frac{1}{2}\boldsymbol{d}_i + \frac{1}{2}\boldsymbol{d}_i'\big)$
8: $\quad$ **end if**
9: **end for**
10: **return $\mathbf{x}_N$**

Figure 2: Training and Sampling algorithms for t-EDM ($\nu > 2$). Our proposed method requires minimal code updates (indicated with blue) over traditional Gaussian diffusion models and converges to the latter as $\nu \to \infty$.

Based on the result in Proposition 2, it is possible to construct deterministic/stochastic samplers for heavy-tailed diffusions. It is worth noting that the SDE in Eqn. 11 implies adding Student-t stochastic noise during inference (Bollerslev, 1987). This is intuitive since the denoising distribution $p_\theta(\mathbf{x}_{t-1}|\mathbf{x}_t)$ is modeled as a Student-t distribution. Next, we provide specific instantiations of the generic sampler in Eq. 11.

**Sampler Instantiations.** We instantiate the continuous-time SDE in Eqn. 11 by setting $g(\sigma_t, \dot{\sigma}_t) = 0$ and $\sigma_{12}(t) = \sigma_t\sigma_{t-\Delta t}$. Consequently, $f(\sigma_t, \dot{\sigma}_t) = -\frac{\dot{\sigma}_t}{\sigma_t}$. In this case, the SDE in Eqn. 11 reduces to an ODE,

$$\frac{d\mathbf{x}_t}{dt} = \frac{\dot{\mu}_t}{\mu_t}\mathbf{x}_t - \left[-\frac{\dot{\sigma}_t}{\sigma_t} + \frac{\dot{\mu}_t}{\mu_t}\right](\mathbf{x}_t - \mu_t\boldsymbol{D}_{\boldsymbol{\theta}}(\mathbf{x}_t, \sigma_t)). \tag{12}$$

**Summary.** Overall, we compare between Gaussian diffusion and heavy-tailed diffusion models in Table 1.

## 3.5 SPECIFIC INSTANTIATIONS: T-EDM

Karras et al. (2022) highlight several design choices during training and sampling, which significantly improve sample quality while reducing sampling budget for image datasets like CIFAR-10 (Krizhevsky, 2009) and ImageNet (Deng et al., 2009). With a similar motivation, we reformulate the perturbation kernel as $q(\mathbf{x}_t|\mathbf{x}_0) = t_d(s(t)\mathbf{x}_0, s(t)^2\sigma(t)^2\mathbf{I}_d, \nu)$ and denote the resulting diffusion model as *t-EDM*.

**Training.** During training, we set the perturbation kernel, $q(\mathbf{x}_t|\mathbf{x}_0)$, parameters $s(t) = 1$, $\sigma(t) = \sigma \sim$ LogNormal($P_{\text{mean}}, P_{\text{std}}$). We parameterize the denoiser $\boldsymbol{D}_{\boldsymbol{\theta}}(\mathbf{x}_t, \sigma_t)$ similar to Karras et al. (2022) with the difference that coefficients like $c_{\text{out}}$ additionally depend on $\nu$. We include full derivations in Appendix A.6. Consequently, our denoising loss can be specified as follows:

$$\mathcal{L}(\theta) \propto \mathbb{E}_{\mathbf{x}_0 \sim p(\mathbf{x}_0)}\mathbb{E}_\sigma\mathbb{E}_{\mathbf{n} \sim t_d(0, \sigma^2\mathbf{I}_d, \nu)}\big[\lambda(\sigma, \nu)\|\boldsymbol{D}_{\boldsymbol{\theta}}(\mathbf{x}_0 + \mathbf{n}, \sigma) - \mathbf{x}_0\|_2^2\big], \tag{13}$$

where $\lambda(\sigma, \nu)$ is a weighting function set to $\lambda(\sigma, \nu) = 1/c_{\text{out}}(\sigma, \nu)^2$.

**Sampling.** Interestingly, it can be shown that the ODE in Eqn. 12 is equivalent to the deterministic dynamics presented in Karras et al. (2022) (See Appendix A.7 for proof). Consequently, we choose $s(t) = 1$ and $\sigma(t) = t$ during sampling, further simplifying the dynamics in Eqn. 12 to $d\mathbf{x}_t/dt = (\mathbf{x}_t - \boldsymbol{D}_{\boldsymbol{\theta}}(\mathbf{x}_t, t))/t$. We adopt the timestep discretization schedule and the choice of the numerical ODE solver (Heun's method (Ascher & Petzold, 1998)) directly from EDM. Figure 2 illustrates the ease of transitioning from a Gaussian diffusion framework (EDM) to t-EDM. Under standard settings, transitioning to t-EDM requires as few as two lines of code change, making our method readily compatible with existing diffusion implementations. Similarly, we can also construct heavy-tailed flows (Albergo et al., 2023; Lipman et al., 2023). We denote the resulting model as *t-Flow* (see App. B for details).

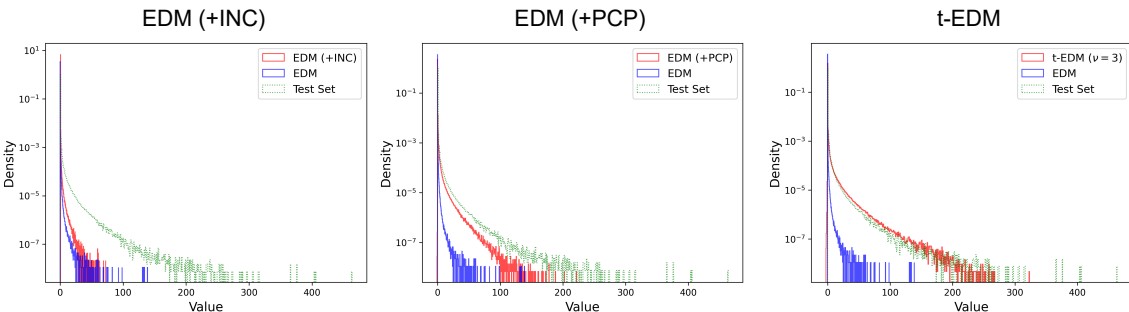

Figure 3: Sample 1-d histogram comparison between EDM and t-EDM on the test set for the Vertically Integrated Liquid (VIL) channel. t-EDM captures heavy-tailed behavior more accurately than other baselines. INC: Inverse CDF Normalization, PCP: Per-Channel Preconditioning

## 4 EXPERIMENTS

Next, we demonstrate the effectiveness of the proposed heavy-tailed diffusion models on real-world weather data for both unconditional and conditional generation tasks. We include full experimental details in App. C.

**Datasets.** We adopt the High-Resolution Rapid Refresh (HRRR) (Dowell et al., 2022) dataset, which is an operational archive of the US km-scale forecasting model. Among multiple dynamical variables in the dataset that exhibit heavy-tailed behavior, based on their dynamic range, we only consider the *Vertically Integrated Liquid* (VIL) and *Vertical Wind Velocity* at level 20 (denoted as w20) channels (see App. C.1 for more details). It is worth noting that the VIL and w20 channels have heavier right and left tails, respectively (See Fig. 6).

**Tasks and Metrics.** We consider both unconditional and conditional generative tasks relevant to weather and climate science. For unconditional modeling, we aim to generate the VIL and w20 physical variables in the HRRR dataset. For conditional modeling, we aim to generatively predict the hourly evolution of the target variable for the next lead-time $(\tau + 1)$ hour-ahead evolution of VIL and w20 based on information only at the current state time $\tau$; see more details in the appendix and see Pathak et al. (2024) for discussion of why hour-ahead, km-scale atmospheric prediction is a stochastic physical task appropriate for conditional generative models. To quantify the empirical performance of unconditional modeling, we rely on comparing 1-d statistics of generated and train/test set samples. More specifically, for quantitative analysis, we report the *Kurtosis Ratio* (KR), the *Skewness Ratio* (SR), and the *Kolmogorov-Smirnov* (KS)-2 sample statistic (at the tails) between the generated and train/test set samples. For qualitative analysis, we compare 1-d histograms between generated and train/test set samples. For the conditional task, we adopt standard probabilistic prediction score metrics such as the *Continuous Ranked Probability Score* (CRPS), the *Root-Mean Squared Error* (RMSE), and the *skill-spread ratio* (SSR); see, e.g., Mardani et al. (2024); Srivastava et al. (2023). A more detailed explanation of our evaluation protocol is provided in App. C.

**Methods and Baselines.** In addition to standard diffusion (EDM (Karras et al., 2022)) and flow models (Albergo et al., 2023) based on Gaussian priors, we introduce two additional baselines that are variants of EDM. To account for the high dynamic-range often exhibited by heavy-tailed distributions, we include *Inverse CDF Normalization* (INC) as an alternative data preprocessing step to z-score normalization. Using the former reduces dynamic range significantly and can make the data distribution closer to Gaussian. We denote this preprocessing scheme combined with standard EDM training as *EDM + INC*. Alternatively, we could instead modulate the noise levels used during EDM training as a function of the dynamic range of the input channel while keeping the data preprocessing unchanged. The main intuition is to use more heavy-tailed noise for large values. We denote this modulating scheme as *Per-Channel Preconditioning* (PCP) and denote the resulting baseline as *EDM + PCP*. We elaborate on these baselines in more detail in App. C.1.2

| | | | VIL (Train) | | | VIL (Test) | | | | w20 (Train) | | | w20 (Test) | |
|---|---|---|---|---|---|---|---|---|---|---|---|---|---|---|
| | Method | $\nu$ | KR ↓ | SR ↓ | KS ↓ | KR ↓ | SR ↓ | KS ↓ | $\nu$ | KR ↓ | SR ↓ | KS ↓ | KR ↓ | SR ↓ | KS ↓ |
| Baselines | EDM | $\infty$ | 210.11 | 10.79 | 0.997 | 45.35 | 5.23 | 0.991 | $\infty$ | 12.59 | 0.89 | 0.991 | 5.01 | 0.38 | 0.978 |
| | +INC | $\infty$ | 11.33 | 2.29 | 0.987 | 1.70 | 0.74 | 0.95 | $\infty$ | **1.80** | **0.18** | 0.909 | **0.23** | **0.13** | 0.763 |
| | +PCP | $\infty$ | 2.12 | 0.72 | 0.800 | **0.31** | **0.09** | 0.522 | $\infty$ | 2.17 | 0.70 | 0.838 | 0.40 | 0.24 | 0.648 |
| Ours | t-EDM | 3 | **1.06** | **0.43** | **0.431** | 0.54 | 0.23 | **0.114** | 3 | 2.44 | 0.65 | **0.683** | 0.52 | 0.21 | **0.286** |
| | t-EDM | 5 | 29.66 | 4.07 | 0.955 | 5.73 | 1.68 | 0.888 | 5 | 8.55 | 1.77 | 0.895 | 3.22 | 1.03 | 0.774 |
| | t-EDM | 7 | 24.35 | 4.14 | 0.959 | 4.57 | 1.72 | 0.908 | 7 | 7.03 | 1.58 | 0.82 | 2.55 | 0.89 | 0.622 |

Table 2: t-EDM outperforms standard diffusion models for unconditional generation on the HRRR dataset. For all metrics, lower is better. Values in **bold** indicate the best results in a column.

| | | | VIL (Train) | | | VIL (Test) | | | | w20 (Train) | | | w20 (Test) | |
|---|---|---|---|---|---|---|---|---|---|---|---|---|---|---|
| | Method | $\nu$ | KR ↓ | SR ↓ | KS ↓ | KR ↓ | SR ↓ | KS ↓ | $\nu$ | KR ↓ | SR ↓ | KS ↓ | KR ↓ | SR ↓ | KS ↓ |
| Baselines | Gaussian Flow | $\infty$ | **0.46** | **0.09** | 0.897 | 0.67 | 0.52 | 0.704 | $\infty$ | 2.03 | 0.36 | 0.294 | 0.34 | 0.01 | 0.384 |
| Ours | t-Flow | 3 | 1.39 | 0.37 | **0.711** | 0.47 | 0.27 | **0.275** | 5 | **1.08** | **0.21** | 0.333 | **0.07** | 0.42 | 0.512 |
| | t-Flow | 5 | 3.30 | 0.75 | 0.857 | 0.05 | 0.07 | 0.633 | 7 | 3.24 | 0.36 | **0.259** | 0.87 | **0.01** | 0.300 |
| | t-Flow | 7 | 3.36 | 0.84 | 0.844 | **0.04** | **0.02** | 0.603 | 9 | 5.47 | 0.41 | 0.478 | 1.86 | 0.034 | **0.289** |

Table 3: t-Flow outperforms standard Gaussian flows for unconditional generation on the HRRR dataset. For all metrics, lower is better. Values in **bold** indicate the best results in a column.

## 4.1 UNCONDITIONAL GENERATION

We assess the effectiveness of different methods on unconditional modeling for the VIL and w20 channels in the HRRR dataset. Fig. 3 qualitatively compares 1-d histograms of sample intensities between different methods for the VIL channel. We make the following key observations. Firstly, though EDM (with additional tricks like noise conditioning) can improve tail coverage, t-EDM covers a broader range of extreme values in the test set. Secondly, in addition to better dynamic range coverage, t-EDM qualitatively performs much better in capturing the density assigned to intermediate intensity levels under the model. We note similar observations from our quantitative results in Table 2, where t-EDM outperforms other baselines on the KS metric, implying our model exhibits better tail estimation over competing baselines for both the VIL and w20 channels. More importantly, unlike traditional Gaussian diffusion models like EDM, t-EDM enables controllable tail estimation by varying $\nu$, which could be useful when modeling a combination of channels with diverse statistical properties. Lastly, we note that while EDM combined with improved preprocessing (INC) or preconditioning (PCP) outperforms t-EDM in some cases, these techniques can also be readily integrated with t-EDM. Therefore, the main point of comparison in Table 2 should be primarily between t-EDM and the standard EDM baseline (row 1). We present similar quantitative results for t-Flow in Table 3 where the advantages of t-Flow over Gaussian Flows for heavy-tailed modeling are quite apparent. We also present additional results for unconditional modeling in App. C.1.7.

## 4.2 CONDITIONAL GENERATION

Next, we consider the task of conditional modeling, where we aim to predict the hourly evolution of a target variable for the next lead time $(\tau + \delta\tau)$ based on the current state at time $\tau$ with $\delta\tau = 1$hr . Table 4 illustrates the performance of EDM and t-EDM on this task for the VIL and w20 channels. We make the following key observations. Firstly, for both channels, t-EDM exhibits better CRPS and SSR scores, implying better probabilistic forecast skills and ensemble than EDM. Moreover, while t-EDM exhibits under-dispersion for VIL, while it is well-calibrated for w20, with its SSR close to an ideal score of 1. On the contrary, the baseline EDM model exhibits under-dispersion for both channels, thus implying overconfident predictions. Secondly, in addition to better calibration, t-EDM is better at tail estimation (as measured by the KS statistic) for the underlying conditional distribution. Lastly, we notice that different values of the parameter $\nu$ are optimal for

| | | | VIL (Test) | | | | | w20 (Test) | | | |
|---|---|---|---|---|---|---|---|---|---|---|---|
| | Method | $\nu$ | CRPS ↓ | RMSE ↓ | SSR ($\rightarrow$ 1) | KS ↓ | $\nu$ | CRPS ↓ | RMSE ↓ | SSR ($\rightarrow$ 1) | KS ↓ |
| Baselines | EDM | $\infty$ | 1.696 | 4.473 | 0.203 | 0.715 | $\infty$ | 0.304 | **0.664** | 0.865 | 0.345 |
| Ours | t-EDM | 3 | 1.649 | 4.526 | 0.255 | **0.419** | 3 | **0.295** | 0.734 | **1.045** | **0.111** |
| | t-EDM | 5 | **1.609** | **4.361** | **0.305** | 0.665 | 5 | 0.301 | 0.674 | 0.901 | 0.323 |

Table 4: t-EDM outperforms EDM for conditional next frame prediction for the HRRR dataset. Values in **bold** indicate the best results in each column. We note that VIL has a higher dynamic range over w20, and thus, the gains for VIL are more apparent (see hist plots in Fig. 6). ($\rightarrow$ 1) indicates values near 1 are better.

different channels, which suggests a more data-driven approach to learning the optimal $\nu$ directly. We present additional results for conditional modeling in App. C.2.

## 5 DISCUSSION AND THEORETICAL INSIGHTS

To conclude, we propose a framework for constructing heavy-tailed diffusion models and demonstrate their effectiveness over traditional diffusion models on unconditional and conditional generation tasks for a high-resolution weather dataset. Here, we highlight some theoretical connections that help gain insights into the effectiveness of our proposed framework while establishing connections with prior work.

**Exploring distribution tails during sampling.** The ODE in Eq. 12 can re-formulated as,

$$\frac{d\mathbf{x}_t}{dt} = \frac{\dot{\mu}_t}{\mu_t}\mathbf{x}_t + \sigma_t^2\left(\frac{\nu + d_1'}{\nu + d}\right)\left[\frac{\dot{\mu}_t}{\mu_t} - \frac{\dot{\sigma}_t}{\sigma_t}\right]\nabla_\mathbf{x}\log p(\mathbf{x}_t, t), \quad (14)$$

where $d_1' = (1/\sigma_t^2)\|\mathbf{x}_t - \mu_t\boldsymbol{D}_\theta(\mathbf{x}_t, \sigma_t)\|_2^2$. By formulating the ODE in terms of the score function, we can gain some intuition into the effectiveness of our model in modeling heavy-tailed distributions. Figure 4 illustrates the variation of the mean and variance of the multiplier $(\nu + d_1')/(\nu + d)$ along the diffusion trajectory across 1M samples generated from our toy models. Interestingly, as the value of $\nu$ decreases, the mean and variance of this multiplier increase significantly, which leads to large score multiplier weights. We hypothesize that this behavior

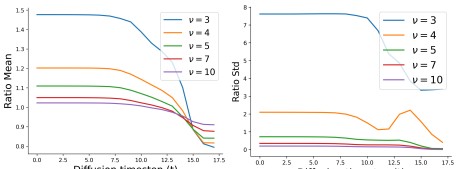

Figure 4: Variation of the mean and standard deviation of the ratio $(\nu + d_1)/(\nu + d)$ with $\nu$ across diffusion sampling trajectory for the toy dataset. As $\nu$ decreases, the mean ratio and its standard deviation increase, leading to large score multiplier weights.

allows our proposed model to explore more diverse regions during sampling (more details in App. A.10).

**Enabling efficient tail coverage during training.** The optimization objective in Eq. 9 has several connections with robust statistical estimators. More specifically, it can be shown that (proof in App. A.11),

$$\nabla_\theta D_\gamma(q \parallel p_\theta) = -\int q(\mathbf{x})\left(\frac{p_\theta(\mathbf{x})}{\|p_\theta\|_{1+\gamma}}\right)^\gamma\left(\nabla_\theta\log p_\theta(\mathbf{x}) - \mathbb{E}_{\tilde{p}_\theta(\mathbf{x})}[\nabla_\theta\log p_\theta(\mathbf{x})]\right)d\mathbf{x},$$

where $q$ and $p_\theta$ denote the forward ($q(\mathbf{x}_{t-\Delta t}|\mathbf{x}_t, \mathbf{x}_0)$) and reverse diffusion posteriors ($p_\theta(\mathbf{x}_{t-\Delta t}|\mathbf{x}_t)$), respectively. Intuitively, the coefficient $\gamma$ weighs the likelihood gradient, $\nabla_\theta\log p_\theta(\mathbf{x})$, and can be set accordingly to ignore or consider outliers when modeling the data distribution. Specifically, when $\gamma > 1$, the model would learn to ignore outliers (Futami et al., 2018; Fujisawa & Eguchi, 2008; Basu et al., 1998) since data points on the tails would be assigned low likelihood. On the contrary, a negative value of $\gamma$ (as is the case in this work since we set $\gamma = -2/(\nu + d)$), the model can assign more weights to capture these extreme values.

We discuss some other connections to prior work in heavy-tailed generative modeling and more recent work in diffusion models in App. F.1 and some limitations of our approach in App. F.2.

## ACKNOWLEDGEMENT

Stephan Mandt acknowledges support from the National Science Foundation (NSF) under an NSF CAREER Award IIS-2047418 and IIS-2007719, the NSF LEAP Center, by the Department of Energy under grant DE-SC0022331, the IARPA WRIVA program, the Hasso Plattner Research Center at UCI, the Chan Zuckerberg Initiative, and gifts from Qualcomm and Disney.

## REPRODUCIBILITY STATEMENT

We include proofs for all theoretical results introduced in the main text in Appendix A. We describe our complete experimental setup (including data processing steps, model specification for training and inference, description of evaluation metrics, and extended experimental results) in Appendix C.

## ETHICS STATEMENT

We develop a generative framework for modeling heavy-tailed distributions and demonstrate its effectiveness for scientific applications. In this context, we do not think our model poses a risk of misinformation or other ethical biases associated with large-scale image synthesis models. However, we would like to point out that similar to other generative models, our model can sometimes hallucinate predictions for certain channels, which could impact downstream applications like weather forecasting.

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

## CONTENTS

## A   PROOFS

### A.1   DERIVATION OF THE PERTURBATION KERNEL

*Proof.* The proof is an adaptation from Ding (2016) and included here for completeness. We define the joint state $\mathbf{x} = [\mathbf{x}_t; \mathbf{x}_{t-\Delta t}]$. Consequently,

$$q(\mathbf{x}|\mathbf{x}_0) = t_{2d}(\boldsymbol{\mu}, \boldsymbol{\Sigma}, \nu), \quad \boldsymbol{\mu} = [\mu_t; \mu_{t-\Delta t}]\mathbf{x}_0, \quad \boldsymbol{\Sigma} = \begin{pmatrix} \sigma_t^2 & \sigma_{12}^2(t) \\ \sigma_{21}^2(t) & \sigma_{t-\Delta t}^2 \end{pmatrix} \otimes \boldsymbol{I}_d,$$

By re-parameterization of the joint Student-t distribution $q(\mathbf{x}|\mathbf{x}_0)$, we have,

$$\mathbf{x} = \boldsymbol{\mu} + \boldsymbol{\Sigma}^{\frac{1}{2}}\mathbf{z}/\sqrt{\kappa}, \quad \mathbf{z} \sim \mathcal{N}(0, \boldsymbol{I}_d) \text{ and } \kappa \sim \chi^2(\nu)/\nu \tag{15}$$

This implies that the conditional distribution $q(\mathbf{x}|\mathbf{x}_0, \kappa) = \mathcal{N}(\boldsymbol{\mu}, \boldsymbol{\Sigma}/\kappa)$. Therefore, following properties of Gaussian distributions, $q(\mathbf{x}_t|\mathbf{x}_0, \kappa) = \mathcal{N}(\mu_t\mathbf{x}_0, \sigma_t^2/\kappa\boldsymbol{I}_d)$. Therefore, from reparameterization,

$$\mathbf{x}_t|\kappa = \mu_t\mathbf{x}_0 + \sigma_t\mathbf{z}/\sqrt{\kappa} \tag{16}$$

which implies that $q(\mathbf{x}_t|\mathbf{x}_0) = t_d(\mu_t\mathbf{x}_0, \sigma_t^2\boldsymbol{I}_d, \nu)$. This completes the proof.   □

### A.2   ON THE POSTERIOR PARAMETERIZATION

The perturbation kernel $q(\mathbf{x}_t|\mathbf{x}_0)$ for Student-t diffusions is parameterized as,

$$q(\mathbf{x}_t|\mathbf{x}_0) = t_d(\mu_t\mathbf{x}_0, \sigma_t^2\boldsymbol{I}_d, \nu) \tag{17}$$

Using re-parameterization,

$$\mathbf{x}_t = \mu_t \mathbf{x}_0 + \sigma_t \frac{\epsilon}{\sqrt{\kappa}}, \quad \epsilon \sim \mathcal{N}(0, \boldsymbol{I}_d), \kappa \sim \chi^2(\nu)/\nu \tag{18}$$

During inference, given a noisy state $\mathbf{x}_t$, we have the following estimation problem,

$$\mathbf{x}_t = \mu_t \mathbb{E}[\mathbf{x}_0|\mathbf{x}_t] + \sigma_t \mathbb{E}\big[\frac{\epsilon}{\sqrt{\kappa}}|\mathbf{x}_t\big] \tag{19}$$

Therefore, the task of denoising can be posed as either estimating $\mathbb{E}[\mathbf{x}_0|\mathbf{x}_t]$ or $\mathbb{E}\big[\frac{\epsilon}{\sqrt{\kappa}}|\mathbf{x}_t\big]$. With this motivation, the posterior $p_{\boldsymbol{\theta}}(\mathbf{x}_{t-\Delta t}|\mathbf{x}_t)$ can be parameterized appropriately. Recall the form of the forward posterior

$$q(\mathbf{x}_{t-\Delta t}|\mathbf{x}_t, \mathbf{x}_0) = t_d(\bar{\boldsymbol{\mu}}_t, \frac{\nu + d_1}{\nu + d}\bar{\sigma}_t^2 \boldsymbol{I}_d, \nu + d) \tag{20}$$

$$\bar{\boldsymbol{\mu}}_t = \mu_{t-\Delta t}\mathbf{x}_0 + \frac{\sigma_{21}^2(t)}{\sigma_t^2}(\mathbf{x}_t - \mu_t\mathbf{x}_0), \qquad \bar{\sigma}_t^2 = \Big[\sigma_{t-\Delta t}^2 - \frac{\sigma_{21}^2(t)\sigma_{12}^2(t)}{\sigma_t^2}\Big] \tag{21}$$

where $d_1 = \frac{1}{\sigma_t^2}\|\mathbf{x}_t - \mu_t\mathbf{x}_0\|^2$. Further simplifying the mean $\bar{\boldsymbol{\mu}}_t$,

$$\bar{\boldsymbol{\mu}}_t(\mathbf{x}_t, \mathbf{x}_0, t) = \mu_{t-\Delta t}\mathbf{x}_0 + \frac{\sigma_{21}^2(t)}{\sigma_t^2}(\mathbf{x}_t - \mu_t\mathbf{x}_0) \tag{22}$$

$$= \mu_{t-\Delta t}\mathbf{x}_0 + \frac{\sigma_{21}^2(t)}{\sigma_t^2}\mathbf{x}_t - \frac{\sigma_{21}^2(t)}{\sigma_t^2}\mu_t\mathbf{x}_0 \tag{23}$$

$$= \Big(\mu_{t-\Delta t} - \mu_t\frac{\sigma_{21}^2(t)}{\sigma_t^2}\Big)\mathbf{x}_0 + \frac{\sigma_{21}^2(t)}{\sigma_t^2}\mathbf{x}_t \tag{24}$$

Therefore, the mean $\boldsymbol{\mu}_{\boldsymbol{\theta}}(\mathbf{x}_t, t)$ of the reverse posterior $p_{\boldsymbol{\theta}}(\mathbf{x}_{t-\Delta t}|\mathbf{x}_t)$ can be parameterized as,

$$\bar{\boldsymbol{\mu}}_{\boldsymbol{\theta}}(\mathbf{x}_t, t) = \Big(\mu_{t-\Delta t} - \mu_t\frac{\sigma_{21}^2(t)}{\sigma_t^2}\Big)\mathbb{E}[\mathbf{x}_0|\mathbf{x}_t] + \frac{\sigma_{21}^2(t)}{\sigma_t^2}\mathbf{x}_t \tag{25}$$

$$\approx \Big(\mu_{t-\Delta t} - \mu_t\frac{\sigma_{21}^2(t)}{\sigma_t^2}\Big)D_{\boldsymbol{\theta}}(\mathbf{x}_t, \sigma_t) + \frac{\sigma_{21}^2(t)}{\sigma_t^2}\mathbf{x}_t \tag{26}$$

where $\mathbb{E}[\mathbf{x}_0|\mathbf{x}_t]$ is learned using a parametric estimator $D_{\boldsymbol{\theta}}(\mathbf{x}_t, \sigma_t)$. This corresponds to the $\mathbf{x}_0$-prediction parameterization presented in Eq. 7 in the main text. Alternatively, From Eqn. 19,

$$\bar{\boldsymbol{\mu}}_{\boldsymbol{\theta}}(\mathbf{x}_t, t) = \Big(\mu_{t-\Delta t} - \mu_t\frac{\sigma_{21}^2(t)}{\sigma_t^2}\Big)\mathbb{E}[\mathbf{x}_0|\mathbf{x}_t] + \frac{\sigma_{21}^2(t)}{\sigma_t^2}\mathbf{x}_t \tag{27}$$

$$= \frac{1}{\mu_t}\Big(\mu_{t-\Delta t} - \mu_t\frac{\sigma_{21}^2(t)}{\sigma_t^2}\Big)\Big(\mathbf{x}_t - \sigma_t\mathbb{E}\big[\frac{\epsilon}{\sqrt{\kappa}}|\mathbf{x}_t\big]\Big) + \frac{\sigma_{21}^2(t)}{\sigma_t^2}\mathbf{x}_t \tag{28}$$

$$= \frac{\mu_{t-\Delta t}}{\mu_t}\mathbf{x}_t - \frac{\sigma_t}{\mu_t}\Big(\mu_{t-\Delta t} - \mu_t\frac{\sigma_{21}^2(t)}{\sigma_t^2}\Big)\mathbb{E}\big[\frac{\epsilon}{\sqrt{\kappa}}|\mathbf{x}_t\big] \tag{29}$$

$$\approx \frac{\mu_{t-\Delta t}}{\mu_t}\mathbf{x}_t - \frac{\sigma_t}{\mu_t}\Big(\mu_{t-\Delta t} - \mu_t\frac{\sigma_{21}^2(t)}{\sigma_t^2}\Big)\epsilon_{\boldsymbol{\theta}}(\mathbf{x}_t, \sigma_t) \tag{30}$$

where $\mathbb{E}\big[\frac{\epsilon}{\sqrt{\kappa}}|\mathbf{x}_t\big]$ is learned using a parametric estimator $\epsilon_{\boldsymbol{\theta}}(\mathbf{x}_t, \sigma_t)$. This corresponds to the $\epsilon$-prediction parameterization (Ho et al., 2020).

### A.3 PROOF OF PROPOSITION 1

We restate the proposition here for convenience,

**Proposition 1.** *For arbitrary distributions $q$ and $p$, in the limit of $\gamma \to 0$, $D_\gamma(q \parallel p)$ converges to $D_{\mathrm{KL}}(q \parallel p)$. Consequently, the objective in Eqn. 9 converges to the DDPM objective stated in Eqn. 1.*

*Proof.* We present our proof in two parts:

1. Firstly, we establish the following relation between the $\gamma$-Power Divergence and the KL-Divergence between two distributions $q$ and $p$.

$$D_\gamma(q \parallel p) = D_{\mathrm{KL}}(q \parallel p) + \mathcal{O}(\gamma) \tag{31}$$

2. Next, for the choice of $\gamma = -\frac{2}{\nu+d}$, we show that in the limit of $\gamma \to 0$, the optimization objective in Eqn. 9 converges to the optimization objective in Eqn. 1

**Relation between $D_{\mathrm{KL}}(q \parallel p)$ and $D_\gamma(q \parallel p)$.** The $\gamma$-Power Divergence as stated in (Kim et al., 2024) assumes the following form:

$$D_\gamma(q \parallel p) = \frac{1}{\gamma}\big[\mathcal{C}_\gamma(q, p) - \mathcal{H}_\gamma(q)\big] \tag{32}$$

where $\gamma \in (-1, 0) \cup (0, \infty)$, and,

$$\mathcal{H}_\gamma(p) = -\|p\|_{1+\gamma} = -\left(\int p(\mathbf{x})^{1+\gamma} d\mathbf{x}\right)^{\frac{1}{1+\gamma}} \qquad \mathcal{C}_\gamma(q, p) = -\int q(\mathbf{x})\left(\frac{p(\mathbf{x})}{\|p\|_{1+\gamma}}\right)^\gamma d\mathbf{x} \tag{33}$$

For subsequent analysis, we assume $\gamma \to 0$. Under this assumption, we simplify $\mathcal{H}_\gamma(q)$ as follows. By definition,

$$\mathcal{H}_\gamma(q) = -\|q\|_{1+\gamma} = -\left(\int q(\mathbf{x})^{1+\gamma} d\mathbf{x}\right)^{\frac{1}{1+\gamma}} \tag{34}$$

$$= -\left(\int q(\mathbf{x}) q(\mathbf{x})^\gamma d\mathbf{x}\right)^{\frac{1}{1+\gamma}} \tag{35}$$

$$= -\left(\int q(\mathbf{x}) \exp(\gamma \log q(\mathbf{x})) d\mathbf{x}\right)^{\frac{1}{1+\gamma}} \tag{36}$$

$$= -\left(\int q(\mathbf{x})\big[1 + \gamma \log q(\mathbf{x}) + \mathcal{O}(\gamma^2)\big] d\mathbf{x}\right)^{\frac{1}{1+\gamma}} \tag{37}$$

$$= -\left(\int q(\mathbf{x}) d\mathbf{x} + \gamma \int q(\mathbf{x}) \log q(\mathbf{x}) d\mathbf{x} + \mathcal{O}(\gamma^2)\right)^{\frac{1}{1+\gamma}} \tag{38}$$

$$= -\left(1 + \gamma \int q(\mathbf{x}) \log q(\mathbf{x}) d\mathbf{x} + \mathcal{O}(\gamma^2)\right)^{\frac{1}{1+\gamma}} \tag{39}$$

Using the approximation $(1 + \delta x)^\alpha \approx 1 + \alpha \delta x$ for a small $\delta$ in the above equation, we have,

$$\mathcal{H}_\gamma(q) = -\|q\|_{1+\gamma} \approx -\left(1 + \frac{\gamma}{1+\gamma}\Big[\int q(\mathbf{x}) \log q(\mathbf{x}) d\mathbf{x} + \mathcal{O}(\gamma)\Big]\right) \tag{40}$$

$$\approx -\left(1 + \gamma(1 - \gamma)\Big[\int q(\mathbf{x}) \log q(\mathbf{x}) d\mathbf{x} + \mathcal{O}(\gamma)\Big]\right) \tag{41}$$

where we have used the approximation $\frac{1}{1+\gamma} \approx 1 - \gamma$ in the above equation. This is justified since $\gamma$ is assumed to be small enough. Therefore, we have,

$$\mathcal{H}_\gamma(q) = -\|q\|_{1+\gamma} \approx -\left(1 + \gamma \int q(\mathbf{x}) \log q(\mathbf{x}) d\mathbf{x} + \mathcal{O}(\gamma^2)\right) \tag{42}$$

Similarly, we now obtain an approximation for the power-cross entropy as follows. By definition,

$$\mathcal{C}_\gamma(q,p) = -\int q(\mathbf{x}) \left(\frac{p(\mathbf{x})}{\|p\|_{1+\gamma}}\right)^\gamma d\mathbf{x} \tag{43}$$

$$= -\int q(\mathbf{x}) \left(1 + \gamma \log \left(\frac{p(\mathbf{x})}{\|p\|_{1+\gamma}}\right) + \mathcal{O}(\gamma^2)\right) d\mathbf{x} \tag{44}$$

$$= -\left(\int q(\mathbf{x}) d\mathbf{x} + \gamma \int q(\mathbf{x}) \log \left(\frac{p(\mathbf{x})}{\|p\|_{1+\gamma}}\right) + \mathcal{O}(\gamma^2)\right) \tag{45}$$

$$= -\left(1 + \gamma \int q(\mathbf{x}) \log p(\mathbf{x}) d\mathbf{x} - \gamma \int q(\mathbf{x}) \log \|p\|_{1+\gamma} d\mathbf{x} + \mathcal{O}(\gamma^2)\right) \tag{46}$$

From Eqn. 42, it follows that,

$$\|p\|_{1+\gamma} \approx \left(1 + \gamma \int p(\mathbf{x}) \log p(\mathbf{x}) d\mathbf{x} + \mathcal{O}(\gamma^2)\right) \tag{47}$$

Therefore,

$$\log \|p\|_{1+\gamma} = \log \left(1 + \gamma \int p(\mathbf{x}) \log p(\mathbf{x}) d\mathbf{x} + \mathcal{O}(\gamma^2)\right) \tag{48}$$

$$\approx \gamma \int p(\mathbf{x}) \log p(\mathbf{x}) d\mathbf{x} \tag{49}$$

where the above result follows from the logarithmic series and ignores the terms of order $\mathcal{O}(\gamma^2)$ or higher. Plugging the approximation in Eqn. 49 in Eqn. 46, we have,

$$\mathcal{C}_\gamma(q,p) = -\left(1 + \gamma \int q(\mathbf{x}) \log p(\mathbf{x}) d\mathbf{x} - \gamma \int q(\mathbf{x}) \log \|p\|_{1+\gamma} d\mathbf{x} + \mathcal{O}(\gamma^2)\right) \tag{50}$$

$$\approx -\left(1 + \gamma \int q(\mathbf{x}) \log p(\mathbf{x}) d\mathbf{x} + \mathcal{O}(\gamma^2)\right) \tag{51}$$

Therefore,

$$D_\gamma(q \,\|\, p) = \frac{1}{\gamma}\left[\mathcal{C}_\gamma(q,p) - \mathcal{H}_\gamma(q)\right] \tag{52}$$

$$= \frac{1}{\gamma}\left[\gamma\left(\int q(\mathbf{x}) \log q(\mathbf{x}) d\mathbf{x} - \int q(\mathbf{x}) \log p(\mathbf{x}) d\mathbf{x} + \mathcal{O}(\gamma^2)\right)\right] \tag{53}$$

$$= \int q(\mathbf{x}) \log \frac{q(\mathbf{x})}{p(\mathbf{x})} d\mathbf{x} + \mathcal{O}(\gamma) \tag{54}$$

$$= D_{\text{KL}}(q \,\|\, p) + \mathcal{O}(\gamma) \tag{55}$$

This establishes the relationship between the KL and $\gamma$-Power divergence between two distributions. Therefore, for two distributions $q$ and $p$, the difference in the magnitude of $D_{\text{KL}}(q \,\|\, p)$ and $D_\gamma(q \,\|\, p)$ is of the order of $\mathcal{O}(\gamma)$. In the limit of $\gamma \to 0$, the $D_\gamma(q \,\|\, p) \to D_{\text{KL}}(q \,\|\, p)$. This concludes the first part of our proof.

**Equivalence between the objectives under $\gamma \to 0$.** For $\gamma = -\frac{2}{\gamma+d}$ and a finite-dimensional dataset with $\mathbf{x}_0 \in \mathbb{R}^d$, it follows that $\gamma \to 0$ implies $\nu \to \infty$. Moreover, in the limit of $\nu \to \infty$, the multivariate Student-t distribution converges to a Gaussian distribution. As already shown in the previous part, under this limit, $D_\gamma(q \parallel p)$ converges to $D_{\mathrm{KL}}(q \parallel p)$. Therefore, under this limit, the optimization objective in Eqn. 9 converges to the standard DDPM objective in Eqn. 1. This completes the proof. $\square$

### A.4 DERIVATION OF THE SIMPLIFIED DENOISING LOSS

Here, we derive the simplified denoising loss presented in Eq. 10 in the main text. We specifically consider the term $D_\gamma(q(\mathbf{x}_{t-\Delta t}|\mathbf{x}_t, \mathbf{x}_0) \parallel p_\theta(\mathbf{x}_{t-\Delta t}|\mathbf{x}_t))$ in Eq. 9. The $\gamma$-power divergence between two Student-t distributions is given by,

$$
\begin{aligned}
\mathcal{D}_\gamma[q_\nu \| p_\nu] = &-\tfrac{1}{\gamma} C_{\nu,d}^{\frac{\gamma}{1+\gamma}} \left(1 + \tfrac{d}{\nu-2}\right)^{-\frac{\gamma}{1+\gamma}} \Big[ -|\mathbf{\Sigma}_0|^{-\frac{\gamma}{2(1+\gamma)}} \left(1 + \tfrac{d}{\nu-2}\right) \\
&+ |\mathbf{\Sigma}_1|^{-\frac{\gamma}{2(1+\gamma)}} \left(1 + \tfrac{1}{\nu-2}\mathrm{tr}\left(\mathbf{\Sigma}_1^{-1}\mathbf{\Sigma}_0\right) + \tfrac{1}{\nu}(\boldsymbol{\mu}_0 - \boldsymbol{\mu}_1)^\top \mathbf{\Sigma}_1^{-1}(\boldsymbol{\mu}_0 - \boldsymbol{\mu}_1)\right) \Big].
\end{aligned}
\tag{56}
$$

where $\gamma = -\frac{2}{\nu+d}$. Recall, the definitions of the forward denoising posterior,

$$
q(\mathbf{x}_{t-\Delta t}|\mathbf{x}_t, \mathbf{x}_0) = t_d(\bar{\boldsymbol{\mu}}_t, \frac{\nu + d_1}{\nu + d}\bar{\sigma}_t^2 \boldsymbol{I}_d, \nu + d)
\tag{57}
$$

$$
\bar{\boldsymbol{\mu}}_t = \mu_{t-\Delta t}\mathbf{x}_0 + \frac{\sigma_{21}^2(t)}{\sigma_t^2}(\mathbf{x}_t - \mu_t\mathbf{x}_0), \qquad \bar{\sigma}_t^2 = \left[\sigma_{t-\Delta t}^2 - \frac{\sigma_{21}^2(t)\sigma_{12}^2(t)}{\sigma_t^2}\right]
\tag{58}
$$

and the reverse denoising posterior,

$$
p_\theta(\mathbf{x}_{t-\Delta t}|\mathbf{x}_t) = t_d(\boldsymbol{\mu_\theta}(\mathbf{x}_t, t), \bar{\sigma}_t^2 \boldsymbol{I}_d, \nu + d)
\tag{59}
$$

where the denoiser mean $\boldsymbol{\mu_\theta}(\mathbf{x}_t, t)$ is further parameterized as follows:

$$
\boldsymbol{\mu_\theta}(\mathbf{x}_t, t) = \frac{\sigma_{21}^2(t)}{\sigma_t^2}\mathbf{x}_t + \left[\mu_{t-\Delta t} - \frac{\sigma_{21}^2(t)}{\sigma_t^2}\mu_t\right]\boldsymbol{D_\theta}(\mathbf{x}_t, \sigma_t)
\tag{60}
$$

Since we only parameterize the mean of the reverse posterior, the majority of the terms in the $\gamma$-power divergence are independent of $\theta$ and can be ignored (or treated as scalar coefficients). Therefore,

$$
D_\gamma(q(\mathbf{x}_{t-\Delta t}|\mathbf{x}_t, \mathbf{x}_0) \parallel p_\theta(\mathbf{x}_{t-\Delta t}|\mathbf{x}_t)) \propto (\bar{\boldsymbol{\mu}}_t - \boldsymbol{\mu_\theta}(\mathbf{x}_t, t))^\top(\bar{\boldsymbol{\mu}}_t - \boldsymbol{\mu_\theta}(\mathbf{x}_t, t))
\tag{61}
$$

$$
\propto \|\bar{\boldsymbol{\mu}}_t - \boldsymbol{\mu_\theta}(\mathbf{x}_t, t)\|_2^2
\tag{62}
$$

$$
\propto \left[\mu_{t-\Delta t} - \frac{\sigma_{21}^2(t)}{\sigma_t^2}\mu_t\right]^2 \|\mathbf{x}_0 - D_\theta(\mathbf{x}_t, \sigma_t)\|_2^2
\tag{63}
$$

For better sample quality, it is common to ignore the scalar multiple in prior works (Ho et al., 2020; Song et al., 2020). Therefore, ignoring the time-dependent scalar multiple,

$$
D_\gamma(q(\mathbf{x}_{t-\Delta t}|\mathbf{x}_t, \mathbf{x}_0) \parallel p_\theta(\mathbf{x}_{t-\Delta t}|\mathbf{x}_t)) \propto \|\mathbf{x}_0 - D_\theta(\mathbf{x}_t, \sigma_t)\|_2^2
\tag{64}
$$

Therefore, the final loss function $\mathcal{L}_\theta$ can be stated as,

$$
\mathcal{L}(\theta) = \mathbb{E}_{\mathbf{x}_0 \sim p(\mathbf{x}_0)}\mathbb{E}_{t \sim p(t)}\mathbb{E}_{\epsilon \sim \mathcal{N}(0, \boldsymbol{I}_d)}\mathbb{E}_{\kappa \sim \frac{1}{\nu}\chi^2(\nu)}\left\|\boldsymbol{D_\theta}\left(\mu_t\mathbf{x}_0 + \sigma_t\frac{\epsilon}{\sqrt{\kappa}}, \sigma_t\right) - \mathbf{x}_0\right\|_2^2
\tag{65}
$$

A.5 PROOF OF PROPOSITION 2

We restate the proposition here for convenience,

**Proposition 2.** *The posterior parameterization in Eqn. 6 induces the following continuous-time dynamics,*

$$d\mathbf{x}_t = \left[\frac{\dot{\mu}_t}{\mu_t}\mathbf{x}_t - \left[f(\sigma_t, \dot{\sigma}_t) + \frac{\dot{\mu}_t}{\mu_t}\right](\mathbf{x}_t - \mu_t \boldsymbol{D}_{\boldsymbol{\theta}}(\mathbf{x}_t, t))\right]dt + \sqrt{\beta(t)g(\sigma_t, \dot{\sigma}_t)}d\boldsymbol{S}_t \tag{66}$$

*where $f : \mathbb{R}^+ \times \mathbb{R}^+ \to \mathbb{R}$ and $g : \mathbb{R}^+ \times \mathbb{R}^+ \to \mathbb{R}^+$ are scalar-valued functions, $\beta_t \in \mathbb{R}^+$ is a scaling coefficient such that the following condition holds,*

$$\frac{1}{\sigma_{12}^2(t)}\left(\sigma_{t-\Delta t}^2 - \beta(t)g(\sigma_t, \dot{\sigma}_t)\Delta t\right) - 1 = f(\sigma_t, \dot{\sigma}_t)\Delta t \tag{67}$$

*where $\dot{\mu}_t, \dot{\sigma}_t$ denote the first-order time-derivatives of the perturbation kernel parameters $\mu_t$ and $\sigma_t$ respectively and the differential $d\boldsymbol{S}_t \sim t_d(0, dt, \nu + d)$.*

*Proof.* We start by writing a single sampling step from our learned posterior distribution. Recall

$$p_\theta(\mathbf{x}_{t-\Delta t}|\mathbf{x}_t) = t_d(\boldsymbol{\mu}_\theta(\mathbf{x}_t, t), \bar{\sigma}_t^2 \boldsymbol{I}_d, \nu + d) \tag{68}$$

where (using the $\epsilon$-prediction parameterization in App. A.2),

$$\boldsymbol{\mu_\theta}(\mathbf{x}_t, t) = \frac{\mu_{t-\Delta t}}{\mu_t}\mathbf{x}_t + \frac{1}{\sigma_t}\left[\sigma_{21}^2(t) - \frac{\mu_{t-\Delta t}}{\mu_t}\sigma_t^2\right]\epsilon_\theta(\mathbf{x}_t, \sigma_t) \tag{69}$$

From re-parameterization, we have,

$$\mathbf{x}_{t-\Delta t} = \mu_\theta(\mathbf{x}_t, t) + \frac{\bar{\sigma}_t}{\sqrt{\kappa_t}}\mathbf{z}_t \quad \mathbf{z}_t \sim \mathcal{N}(0, \boldsymbol{I}_d), \quad \kappa_t \sim \chi^2(\nu + d)/(\nu + d) \tag{70}$$

$$\mathbf{x}_{t-\Delta t} = \frac{\mu_{t-\Delta t}}{\mu_t}\mathbf{x}_t + \frac{1}{\sigma_t}\left[\sigma_{21}^2(t) - \frac{\mu_{t-\Delta t}}{\mu_t}\sigma_t^2\right]\epsilon_\theta(\mathbf{x}_t, \sigma_t) + \frac{\bar{\sigma}_t}{\sqrt{\kappa_t}}\mathbf{z}_t \tag{71}$$

Moreover, we choose the posterior scale to be the same as the forward posterior $q(\mathbf{x}_{t-1}|\mathbf{x}_t, \mathbf{x}_0)$ i.e.

$$\bar{\sigma}_t^2 = \left[\sigma_{t-\Delta t}^2 - \frac{\sigma_{21}^2(t)\sigma_{12}^2(t)}{\sigma_t^2}\right]\boldsymbol{I}_d \tag{72}$$

This implies,

$$\sigma_{21}^2(t) = \frac{\sigma_t^2}{\sigma_{12}^2(t)}\left(\sigma_{t-\Delta t}^2 - \bar{\sigma}_t^2\right) \tag{73}$$

Substituting this form of $\sigma_{21}^2(t)$ into Eqn. 71, we have,

$$\mathbf{x}_{t-\Delta t} = \frac{\mu_{t-\Delta t}}{\mu_t}\mathbf{x}_t + \frac{1}{\sigma_t}\left[\frac{\sigma_t^2}{\sigma_{12}^2(t)}\left(\sigma_{t-\Delta t}^2 - \bar{\sigma}_t^2\right) - \frac{\mu_{t-\Delta t}}{\mu_t}\sigma_t^2\right]\epsilon_\theta(\mathbf{x}_t, \sigma_t) + \frac{\bar{\sigma}_t}{\sqrt{\kappa_t}}\mathbf{z}_t \tag{74}$$

$$= \frac{\mu_{t-\Delta t}}{\mu_t}\mathbf{x}_t + \sigma_t\left[\frac{1}{\sigma_{12}^2(t)}\left(\sigma_{t-\Delta t}^2 - \bar{\sigma}_t^2\right) - \frac{\mu_{t-\Delta t}}{\mu_t}\right]\epsilon_\theta(\mathbf{x}_t, \sigma_t) + \frac{\bar{\sigma}_t}{\sqrt{\kappa_t}}\mathbf{z}_t \tag{75}$$

$$= \frac{\mu_{t-\Delta t}}{\mu_t}\mathbf{x}_t + \sigma_t\left[\frac{1}{\sigma_{12}^2(t)}\left(\sigma_{t-\Delta t}^2 - \bar{\sigma}_t^2\right) - 1 + 1 - \frac{\mu_{t-\Delta t}}{\mu_t}\right]\epsilon_\theta(\mathbf{x}_t, \sigma_t) + \frac{\bar{\sigma}_t}{\sqrt{\kappa_t}}\mathbf{z}_t \tag{76}$$

$$= \frac{\mu_{t-\Delta t}}{\mu_t}\mathbf{x}_t + \sigma_t\left[\frac{1}{\sigma_{12}^2(t)}\left(\sigma_{t-\Delta t}^2 - \bar{\sigma}_t^2\right) - 1 + \frac{\dot{\mu}_t}{\mu_t}\Delta t\right]\epsilon_\theta(\mathbf{x}_t, \sigma_t) + \frac{\bar{\sigma}_t}{\sqrt{\kappa_t}}\mathbf{z}_t \tag{77}$$

where in the above equation we use the first-order approximation $\dot{\mu}_t = \frac{\mu_t - \mu_{t-\Delta t}}{\Delta t}$. Next, we make the following design choices:

1. Firstly, we assume the following form of the reverse posterior variance $\bar{\sigma}_t^2$:

$$\bar{\sigma}_t^2 = \beta(t)g(\sigma_t, \dot{\sigma}_t)\Delta t \tag{78}$$

where $g : \mathbb{R}^+ \times \mathbb{R}^+ \to \mathbb{R}^+$ and $\beta(t) \in \mathbb{R}^+$ represents a time-varying scaling factor chosen empirically which can be used to vary the noise injected at each sampling step. It is worth noting that a positive $\dot{\sigma}_t$ (as indicated in the definition of g) is a consequence of a monotonically increasing noise schedule $\sigma_t$ in diffusion model design.

2. Secondly, we make the following design choice:

$$\frac{1}{\sigma_{12}^2(t)}\left(\sigma_{t-\Delta t}^2 - \bar{\sigma}_t^2\right) - 1 = f(\sigma_t, \dot{\sigma}_t)\Delta t \tag{79}$$

where $f : \mathbb{R}^+ \times \mathbb{R}^+ \to \mathbb{R}$

With these two design choices, Eqn. 77 simplifies as:

$$\mathbf{x}_{t-\Delta t} = \frac{\mu_{t-\Delta t}}{\mu_t}\mathbf{x}_t + \sigma_t\left[\frac{1}{\sigma_{12}^2(t)}\left(\sigma_{t-\Delta t}^2 - \bar{\sigma}_t^2\right) - 1 + \frac{\dot{\mu}_t}{\mu_t}\Delta t\right]\epsilon_\theta(\mathbf{x}_t, \sigma_t) + \frac{\bar{\sigma}_t}{\sqrt{\kappa_t}}\mathbf{z} \tag{80}$$

$$= \frac{\mu_{t-\Delta t}}{\mu_t}\mathbf{x}_t + \sigma_t\left[f(\sigma_t, \dot{\sigma}_t)\Delta t + \frac{\dot{\mu}_t}{\mu_t}\Delta t\right]\epsilon_\theta(\mathbf{x}_t, \sigma_t) + \sqrt{\beta(t)g(\sigma_t, \dot{\sigma}_t)}\sqrt{\Delta t}\frac{\mathbf{z}_t}{\sqrt{\kappa_t}} \tag{81}$$

$$= \frac{\mu_{t-\Delta t}}{\mu_t}\mathbf{x}_t + \sigma_t\left[f(\sigma_t, \dot{\sigma}_t) + \frac{\dot{\mu}_t}{\mu_t}\right]\epsilon_\theta(\mathbf{x}_t, \sigma_t)\Delta t + \sqrt{\beta(t)g(\sigma_t, \dot{\sigma}_t)}\sqrt{\Delta t}\frac{\mathbf{z}_t}{\sqrt{\kappa_t}} \tag{82}$$

$$\mathbf{x}_{t-\Delta t} - \mathbf{x}_t = \left(\frac{\mu_{t-\Delta t}}{\mu_t} - 1\right)\mathbf{x}_t + \sigma_t\left[f(\sigma_t, \dot{\sigma}_t) + \frac{\dot{\mu}_t}{\mu_t}\right]\epsilon_\theta(\mathbf{x}_t, \sigma_t)\Delta t + \sqrt{\beta(t)g(\sigma_t, \dot{\sigma}_t)}\sqrt{\Delta t}\frac{\mathbf{z}_t}{\sqrt{\kappa_t}} \tag{83}$$

$$\mathbf{x}_{t-\Delta t} - \mathbf{x}_t = -\left[\frac{\dot{\mu}_t}{\mu_t}\mathbf{x}_t - \sigma_t\left[f(\sigma_t, \dot{\sigma}_t) + \frac{\dot{\mu}_t}{\mu_t}\right]\epsilon_\theta(\mathbf{x}_t, \sigma_t)\right]\Delta t + \sqrt{\beta(t)g(\sigma_t, \dot{\sigma}_t)}\sqrt{\Delta t}\frac{\mathbf{z}_t}{\sqrt{\kappa_t}} \tag{84}$$

In the limit of $\Delta t \to 0$:

$$d\mathbf{x}_t = \left[\frac{\dot{\mu}_t}{\mu_t}\mathbf{x}_t - \sigma_t\left[f(\sigma_t, \dot{\sigma}_t) + \frac{\dot{\mu}_t}{\mu_t}\right]\epsilon_\theta(\mathbf{x}_t, \sigma_t)\right]dt + \sqrt{\beta(t)g(\sigma_t, \dot{\sigma}_t)}\frac{dW_t}{\sqrt{\kappa_t}} \tag{85}$$

$$= \left[\frac{\dot{\mu}_t}{\mu_t}\mathbf{x}_t - \left[f(\sigma_t, \dot{\sigma}_t) + \frac{\dot{\mu}_t}{\mu_t}\right](\mathbf{x}_t - \mu_t\mathbf{D}_\theta(\mathbf{x}_t, \sigma_t))\right]dt + \sqrt{\beta(t)g(\sigma_t, \dot{\sigma}_t)}\frac{dW_t}{\sqrt{\kappa_t}} \tag{86}$$

Moreover, since $dW_t \sim \mathcal{N}(0, dt)$ and $\kappa_t \sim \chi^2(\nu + d)/(\nu + d)$, the term $d\mathbf{S}_t = dW_t/\sqrt{\kappa_t}$ is distributed as a Student-t random variable with $d\mathbf{S}_t \sim t_d(0, dt, \nu + d)$. Therefore,

$$d\mathbf{x}_t = \left[\frac{\dot{\mu}_t}{\mu_t}\mathbf{x}_t - \sigma_t\left[f(\sigma_t, \dot{\sigma}_t) + \frac{\dot{\mu}_t}{\mu_t}\right]\epsilon_\theta(\mathbf{x}_t, \sigma_t)\right]dt + \sqrt{\beta(t)g(\sigma_t, \dot{\sigma}_t)}d\mathbf{S}_t \tag{87}$$

which gives the required SDE formulation for the diffusion posterior sampling. Next, we discuss specific choices of $f(\sigma_t, \dot{\sigma}_t)$ and $g(\sigma_t, \dot{\sigma}_t)$.

**On the choice of $f(\sigma_t, \dot{\sigma}_t)$ and $g(\sigma_t, \dot{\sigma}_t)$:** Recall our design choices:

$$\bar{\sigma}_t^2 = \beta(t)g(\sigma_t, \dot{\sigma}_t)\Delta t \tag{88}$$

$$\frac{1}{\sigma_{12}^2(t)}\left(\sigma_{t-\Delta t}^2 - \bar{\sigma}_t^2\right) - 1 = f(\sigma_t, \dot{\sigma}_t)\Delta t \tag{89}$$

Substituting the value of $\bar{\sigma}_t^2$ from the first design choice to the second yields the following condition:

$$\frac{1}{\sigma_{12}^2(t)}\left(\sigma_{t-\Delta t}^2 - \beta(t)g(\sigma_t, \dot{\sigma}_t)\Delta t\right) - 1 = f(\sigma_t, \dot{\sigma}_t)\Delta t \tag{90}$$

This concludes the proof. It is worth noting that the above equation provides two degrees of freedom, i.e., we can choose two variables among $\sigma_{12}(t), g, f$ and automatically determine the third. However, it is more convenient to choose $\sigma_{12}(t)$ and $g$, since both these quantities should be positive. Different choices of these quantities yield different instantiations of the SDE in Eqn. 87 as illustrated in the main text. $\square$

### A.6 DERIVING THE DENOISER PRECONDITIONER FOR t-EDM

Recall our denoiser parameterization,

$$D_\theta(\mathbf{x}, \sigma) = c_{\text{skip}}(\sigma, \nu)\mathbf{x} + c_{\text{out}}(\sigma, \nu)\boldsymbol{F_\theta}(c_{\text{in}}(\sigma, \nu)\mathbf{x}, c_{\text{noise}}(\sigma)) \tag{91}$$

Karras et al. (2022) highlight the following design choices, which we adopt directly.

1. Derive $c_{\text{in}}$ based on constraining the input variance to 1
2. Derive $c_{\text{skip}}$ and $c_{\text{out}}$ to constrain output variance to 1 and additionally minimizing $c_{\text{out}}$ to bound scaling errors in $\boldsymbol{F_\theta}(\mathbf{x}, \sigma)$.

The coefficient $c_{\text{in}}$ can be derived by setting the network inputs to have unit variance. Therefore,

$$\text{Var}_{\mathbf{x}_0,\mathbf{n}}\left[c_{\text{in}}(\sigma)(\mathbf{x}_0 + \mathbf{n})\right] = 1 \tag{92}$$

$$c_{\text{in}}(\sigma, \nu)^2 \, \text{Var}_{\mathbf{x}_0,\mathbf{n}}\left[\mathbf{x}_0 + \mathbf{n}\right] = 1 \tag{93}$$

$$c_{\text{in}}(\sigma, \nu)^2\left(\sigma_{\text{data}}^2 + \frac{\nu}{\nu - 2}\sigma^2\right) = 1 \tag{94}$$

$$c_{\text{in}}(\sigma, \nu) = 1 \Big/ \sqrt{\frac{\nu}{\nu - 2}\sigma^2 + \sigma_{\text{data}}^2}. \tag{95}$$

The coefficients $c_{\text{skip}}$ and $c_{\text{out}}$ can be derived by setting the training target to have unit variance. Similar to Karras et al. (2022) our training target can be specified as:

$$\boldsymbol{F}_{\text{target}} = \frac{1}{c_{\text{out}}(\sigma, \nu)}\left(\mathbf{x}_0 - c_{\text{skip}}(\sigma, \nu)(\mathbf{x}_0 + \mathbf{n})\right) \tag{96}$$

$$\text{Var}_{\mathbf{x}_0,\mathbf{n}}\left[F_{\text{target}}(\mathbf{x}_0, \mathbf{n}; \sigma)\right] = 1 \tag{97}$$

$$\text{Var}_{\mathbf{x}_0,\mathbf{n}}\left[\frac{1}{c_{\text{out}}(\sigma, \nu)}\left(\mathbf{x}_0 - c_{\text{skip}}(\sigma, \nu)(\mathbf{x}_0 + \mathbf{n})\right)\right] = 1 \tag{98}$$

$$\frac{1}{c_{\text{out}}(\sigma, \nu)^2}\text{Var}_{\mathbf{x}_0,\mathbf{n}}\left[\mathbf{x}_0 - c_{\text{skip}}(\sigma, \nu)(\mathbf{x}_0 + \mathbf{n})\right] = 1 \tag{99}$$

$$c_{\text{out}}(\sigma, \nu)^2 = \text{Var}_{\mathbf{x}_0,\mathbf{n}}\left[\mathbf{x}_0 - c_{\text{skip}}(\sigma, \nu)(\mathbf{x}_0 + \mathbf{n})\right] \tag{100}$$

$$c_{\text{out}}(\sigma, \nu)^2 = \text{Var}_{\mathbf{x}_0,\mathbf{n}}\left[\left(1 - c_{\text{skip}}(\sigma, \nu)\right)\mathbf{x}_0 + c_{\text{skip}}(\sigma, \nu)\,\mathbf{n}\right] \tag{101}$$

$$c_{\text{out}}(\sigma, \nu)^2 = \left(1 - c_{\text{skip}}(\sigma, \nu)\right)^2 \sigma_{\text{data}}^2 + c_{\text{skip}}(\sigma, \nu)^2 \frac{\nu}{\nu - 2}\sigma^2 \tag{102}$$

Lastly, setting $c_{\text{skip}}(\sigma, \nu)$ to minimize $c_{\text{out}}(\sigma, \nu)$, we obtain,

$$c_{\text{skip}}(\sigma, \nu) = \sigma_{\text{data}}^2 \Big/ \left(\frac{\nu}{\nu - 2}\sigma^2 + \sigma_{\text{data}}^2\right). \tag{103}$$

Consequently $c_{\text{out}}(\sigma, \nu)$ can be specified as:

$$c_{\text{out}}(\sigma, \nu) = \sqrt{\frac{\nu}{\nu-2}}\sigma \cdot \sigma_{\text{data}} \Big/ \sqrt{\frac{\nu}{\nu-2}\sigma^2 + \sigma_{\text{data}}^2}. \tag{104}$$

## A.7 EQUIVALENCE WITH THE EDM ODE

Similar to Karras et al. (2022), we start by deriving the optimal denoiser for the denoising loss function. Moreover, we reformulate the form of the perturbation kernel as $q(\mathbf{x}_t|\mathbf{x}_0) = t_d(\mu_t\mathbf{x}_0, \sigma_t^2\boldsymbol{I}_d, \nu) = t_d(s(t)\mathbf{x}_0, s(t)^2\sigma(t)^2\boldsymbol{I}_d, \nu)$ by setting $\mu_t = s(t)$ and $\sigma_t = s(t)\sigma(t)$. The denoiser loss can then be specified as follows,

$$
\begin{aligned}
\mathcal{L}(D, \sigma) &= \mathbb{E}_{\mathbf{x}_0 \sim p(\mathbf{x}_0)}\mathbb{E}_{\mathbf{n}\sim t_d(0,\sigma^2\boldsymbol{I}_d,\nu)}\big[\lambda(\sigma,\nu)\|\boldsymbol{D}(s(t)[\mathbf{x}_0+\mathbf{n}],\sigma)-\mathbf{x}_0\|_2^2\big] &(105)\\
&= \mathbb{E}_{\mathbf{x}_0 \sim p(\mathbf{x}_0)}\mathbb{E}_{\mathbf{x}\sim t_d(s(t)\mathbf{x}_0,s(t)^2\sigma^2\boldsymbol{I}_d,\nu)}\big[\lambda(\sigma,\nu)\|\boldsymbol{D}(\mathbf{x},\sigma)-\mathbf{x}_0\|_2^2\big] &(106)\\
&= \mathbb{E}_{\mathbf{x}_0 \sim p(\mathbf{x}_0)}\Big[\int t_d(\mathbf{x}; s(t)\mathbf{x}_i, s(t)^2\sigma^2\boldsymbol{I}_d,\nu)\big[\lambda(\sigma,\nu)\|\boldsymbol{D}(\mathbf{x},\sigma)-\mathbf{x}_0\|_2^2\big]d\mathbf{x}\Big] &(107)\\
&= \frac{1}{N}\sum_i \int t_d(\mathbf{x}; s(t)\mathbf{x}_i, s(t)^2\sigma^2\boldsymbol{I}_d,\nu)\big[\lambda(\sigma,\nu)\|\boldsymbol{D}(\mathbf{x},\sigma)-\mathbf{x}_i\|_2^2\big]d\mathbf{x} &(108)
\end{aligned}
$$

where the last result follows from assuming $p(\mathbf{x}_0)$ as the empirical data distribution. Thus, the optimal denoiser can be specified by setting $\nabla_D\mathcal{L}(D, \sigma) = 0$. Therefore,

$$\nabla_D\mathcal{L}(D, \sigma) = 0 \tag{109}$$

Consequently,

$$\nabla_D\frac{1}{N}\sum_i\int t_d(\mathbf{x}; s(t)\mathbf{x}_i, s(t)^2\sigma^2\boldsymbol{I}_d,\nu)\big[\lambda(\sigma,\nu)\|\boldsymbol{D}(\mathbf{x},\sigma)-\mathbf{x}_i\|_2^2\big]d\mathbf{x} = 0 \tag{110}$$

$$\frac{1}{N}\sum_i\int t_d(\mathbf{x}; s(t)\mathbf{x}_i, s(t)^2\sigma^2\boldsymbol{I}_d,\nu)\big[\lambda(\sigma,\nu)(\boldsymbol{D}(\mathbf{x},\sigma)-\mathbf{x}_i)\big]d\mathbf{x} = 0 \tag{111}$$

$$\int\sum_i t_d(\mathbf{x}; s(t)\mathbf{x}_i, s(t)^2\sigma^2\boldsymbol{I}_d,\nu)(\boldsymbol{D}(\mathbf{x},\sigma)-\mathbf{x}_i)d\mathbf{x} = 0 \tag{112}$$

The optimal denoiser $\boldsymbol{D}(\mathbf{x}, \sigma)$, can be obtained from Eq. 112 as,

$$\boldsymbol{D}(\mathbf{x},\sigma) = \frac{\sum_i t_d(\mathbf{x}; s(t)\mathbf{x}_i, s(t)^2\sigma^2\boldsymbol{I}_d,\nu)\mathbf{x}_i}{\sum_i t_d(\mathbf{x}; s(t)\mathbf{x}_i, s(t)^2\sigma^2\boldsymbol{I}_d,\nu)} \tag{113}$$

We can further simplify the optimal denoiser as,

$$
\begin{aligned}
\boldsymbol{D}(\mathbf{x},\sigma) &= \frac{\sum_i t_d(\mathbf{x}; s(t)\mathbf{x}_i, s(t)^2\sigma^2\boldsymbol{I}_d,\nu)\mathbf{x}_i}{\sum_i t_d(\mathbf{x}; s(t)\mathbf{x}_i, s(t)^2\sigma^2\boldsymbol{I}_d,\nu)} &(114)\\
&= \frac{\sum_i t_d(\frac{\mathbf{x}}{s(t)}; \mathbf{x}_i, \sigma^2\boldsymbol{I}_d,\nu)\mathbf{x}_i}{\sum_i t_d(\frac{\mathbf{x}}{s(t)}; \mathbf{x}_i, \sigma^2\boldsymbol{I}_d,\nu)} &(115)\\
&= \boldsymbol{D}\Big(\frac{\mathbf{x}}{s(t)},\sigma\Big) &(116)
\end{aligned}
$$

Next, recall the ODE dynamics (Eqn. 12) in our formulation,

$$\frac{d\mathbf{x}_t}{dt} = \frac{\dot{\mu}_t}{\mu_t}\mathbf{x}_t - \Big[-\frac{\dot{\sigma}_t}{\sigma_t} + \frac{\dot{\mu}_t}{\mu_t}\Big](\mathbf{x}_t - \mu_t\boldsymbol{D}_{\boldsymbol{\theta}}(\mathbf{x}_t, \sigma(t))) \tag{117}$$

Reparameterizing the above ODE by setting $\mu_t = s(t)$ and $\sigma_t = s(t)\sigma(t)$,

$$\frac{d\mathbf{x}_t}{dt} = \frac{\dot{\mu}_t}{\mu_t}\mathbf{x}_t - \left[ -\frac{\dot{\sigma}_t}{\sigma_t} + \frac{\dot{\mu}_t}{\mu_t} \right](\mathbf{x}_t - s(t)\boldsymbol{D}_{\boldsymbol{\theta}}(\mathbf{x}_t, \sigma(t))) \tag{118}$$

$$= \frac{\dot{s}(t)}{s(t)}\mathbf{x}_t - \left[ \frac{\dot{s}(t)}{s(t)} - \frac{\dot{\sigma}(t)s(t) + \sigma(t)\dot{s}(t)}{\sigma(t)s(t)} \right](\mathbf{x}_t - s(t)\boldsymbol{D}_{\boldsymbol{\theta}}(\mathbf{x}_t, \sigma(t))) \tag{119}$$

$$= \frac{\dot{s}(t)}{s(t)}\mathbf{x}_t - \left[ -\frac{\dot{\sigma}(t)}{\sigma(t)} \right](\mathbf{x}_t - s(t)\boldsymbol{D}_{\boldsymbol{\theta}}(\mathbf{x}_t, \sigma(t))) \tag{120}$$

$$= \frac{\dot{s}(t)}{s(t)}\mathbf{x}_t + \frac{\dot{\sigma}(t)}{\sigma(t)}(\mathbf{x}_t - s(t)\boldsymbol{D}_{\boldsymbol{\theta}}(\mathbf{x}_t, \sigma(t))) \tag{121}$$

Lastly, since Karras et al. (2022) propose to train the denoiser $\boldsymbol{D}_{\boldsymbol{\theta}}$ using unscaled noisy state, from Eqn. 116, we can re-write the above ODE as,

$$\frac{d\mathbf{x}_t}{dt} = \left[ \frac{\dot{s}(t)}{s(t)} + \frac{\dot{\sigma}(t)}{\sigma(t)} \right]\mathbf{x}_t - \frac{\dot{\sigma}(t)s(t)}{\sigma(t)}\boldsymbol{D}_{\boldsymbol{\theta}}\left( \frac{\mathbf{x}_t}{s(t)}, \sigma(t) \right) \tag{122}$$

The form of the ODE in Eqn. 122 is equivalent to the ODE presented in Karras et al. (2022) (Algorithm 1 line 4 in their paper). This concludes the proof.

## A.8   CONDITIONAL VECTOR FIELD FOR T-FLOW

Here, we derive the conditional vector field for the *t-Flow* interpolant. Recall, the interpolant in *t-Flow* is derived as follows,

$$\mathbf{x}_t = t\mathbf{x}_0 + (1 - t)\frac{\epsilon}{\sqrt{\kappa}}, \quad \epsilon \sim \mathcal{N}(0, \boldsymbol{I}_d), \kappa \sim \chi^2(\nu)/\nu \tag{123}$$

It follows that,

$$\mathbf{x}_t = t\mathbb{E}[\mathbf{x}_0|\mathbf{x}_t] + (1 - t)\mathbb{E}\left[ \frac{\epsilon}{\sqrt{\kappa}}|\mathbf{x}_t \right] \tag{124}$$

Moreover, following Eq. 2.10 in Albergo et al. (2023), the conditional vector field $\boldsymbol{b}(\mathbf{x}_t, t)$ can be defined as,

$$\boldsymbol{b}(\mathbf{x}_t, t) = \mathbb{E}[\dot{\mathbf{x}}_t|\mathbf{x}_t] = \mathbb{E}[\mathbf{x}_0|\mathbf{x}_t] - \mathbb{E}\left[ \frac{\epsilon}{\sqrt{\kappa}}|\mathbf{x}_t \right] \tag{125}$$

From Eqns. 124 and 125, the conditional vector field can be simplified as,

$$\boldsymbol{b}(\mathbf{x}_t, t) = \frac{\mathbf{x}_t - \mathbb{E}\left[ \frac{\epsilon}{\sqrt{\kappa}}|\mathbf{x}_t \right]}{t} \tag{126}$$

This concludes the proof.

## A.9   CONNECTION TO DENOISING SCORE MATCHING

We start by defining the score for the perturbation kernel $q(\mathbf{x}_t|\mathbf{x}_0)$. The pdf for the perturbation kernel $q(\mathbf{x}_t|\mathbf{x}_0) = t_d(\mu_t\mathbf{x}_0, \sigma_t^2\boldsymbol{I}_d, \nu)$ can be expressed as (ignoring the normalization constant):

$$q(\mathbf{x}_t|\mathbf{x}_0) \propto \left[ 1 + \frac{1}{\nu\sigma_t^2}(\mathbf{x}_t - \mu_t\mathbf{x}_0)^\top(\mathbf{x}_t - \mu_t\mathbf{x}_0) \right]^{\frac{-(\nu+d)}{2}} \tag{127}$$

Therefore,

$$\nabla_{\mathbf{x}_t} \log q(\mathbf{x}_t | \mathbf{x}_0) = -\frac{\nu + d}{2} \nabla_{\mathbf{x}_t} \log \left[ 1 + \frac{1}{\nu \sigma_t^2} \|\mathbf{x}_t - \mu_t \mathbf{x}_0\|_2^2 \right] \tag{128}$$

$$= -\frac{\nu + d}{2} \frac{1}{1 + \frac{1}{\nu \sigma_t^2} \|\mathbf{x}_t - \mu_t \mathbf{x}_0\|_2^2} \frac{2}{\nu \sigma_t^2} \left( \mathbf{x}_t - \mu_t \mathbf{x}_0 \right) \tag{129}$$

Denoting $d_1 = \frac{1}{\sigma_t^2} \|\mathbf{x}_t - \mu_t \mathbf{x}_0\|_2^2$ for convenience, we have,

$$\nabla_{\mathbf{x}_t} \log q(\mathbf{x}_t | \mathbf{x}_0) = -\left( \frac{\nu + d}{\nu + d_1} \right) \frac{1}{\sigma_t^2} \left( \mathbf{x}_t - \mu_t \mathbf{x}_0 \right) \tag{130}$$

In Denoising Score Matching (DSM) (Vincent, 2011), the following objective is minimized,

$$L_{\text{DSM}} = \mathbb{E}_{t \sim p(t), \mathbf{x}_0 \sim p(\mathbf{x}_0), \mathbf{x}_t \sim q(\mathbf{x}_t | \mathbf{x}_0)} \left[ \gamma(t) \|\nabla_{\mathbf{x}_t} \log q(\mathbf{x}_t | \mathbf{x}_0) - \mathbf{s}_\theta(\mathbf{x}_t, t)\|_2^2 \right] \tag{131}$$

for some loss weighting function $\gamma(t)$. Parameterizing the score estimator $\mathbf{s}_\theta(\mathbf{x}_t, t)$ as:

$$\mathbf{s}_\theta(\mathbf{x}_t, t) = -\left( \frac{\nu + d}{\nu + d_1} \right) \frac{1}{\sigma_t^2} \left( \mathbf{x}_t - \mu_t D_\theta(\mathbf{x}_t, \sigma_t) \right) \tag{132}$$

With this parameterization of $\mathbf{s}_\theta(\mathbf{x}_t, t)$ and some choice of $\gamma(t)$, the DSM objective can be further simplified as follows,

$$L_{\text{DSM}} = \mathbb{E}_{\mathbf{x}_0 \sim p(\mathbf{x}_0)} \mathbb{E}_t \mathbb{E}_{\epsilon \sim \mathcal{N}(0,I), \kappa \sim \chi^2(\nu)/\nu} \left[ \left( \frac{\nu + d}{\nu + d_1} \right)^2 \left\| \mathbf{x}_0 - D_\theta\left( \mu_t \mathbf{x}_0 + \sigma_t \frac{\epsilon}{\sqrt{\kappa}}, \sigma_t \right) \right\|_2^2 \right] \tag{133}$$

$$= \mathbb{E}_{\mathbf{x}_0 \sim p(\mathbf{x}_0)} \mathbb{E}_t \mathbb{E}_{\epsilon \sim \mathcal{N}(0,I), \kappa \sim \chi^2(\nu)/\nu} \left[ \lambda(\mathbf{x}_t, \nu, t) \left\| \mathbf{x}_0 - D_\theta\left( \mu_t \mathbf{x}_0 + \sigma_t \frac{\epsilon}{\sqrt{\kappa}}, \sigma_t \right) \right\|_2^2 \right] \tag{134}$$

where $\lambda(\mathbf{x}_t, \nu, t) = \left( \frac{\nu + d}{\nu + d_1} \right)^2$, which is equivalent to a scaled version of the simplified denoising loss Eq. 10. This concludes the proof. As an additional caveat, the score parameterization in Eq. 132 depends on $d_1 = \frac{1}{\sigma_t^2} \|\mathbf{x}_t - \mu_t \mathbf{x}_0\|_2^2$, which can be approximated during inference as, $d_1 \approx \frac{1}{\sigma_t^2} \|\mathbf{x}_t - \mu_t D_\theta(\mathbf{x}_t, \sigma_t)\|_2^2$

## A.10 ODE REFORMULATION AND CONNECTIONS TO INVERSE PROBLEMS.

Recall the ODE dynamics in terms of the denoiser can be specified as,

$$\frac{d\mathbf{x}_t}{dt} = \frac{\dot{\mu}_t}{\mu_t} \mathbf{x}_t - \left[ -\frac{\dot{\sigma}_t}{\sigma_t} + \frac{\dot{\mu}_t}{\mu_t} \right] (\mathbf{x}_t - \mu_t D_\theta(\mathbf{x}_t, \sigma_t)) \tag{135}$$

From Eq. 132, we have,

$$\mathbf{x}_t - \mu_t D_\theta(\mathbf{x}_t, \sigma_t) = -\sigma_t^2 \left( \frac{\nu + d_1}{\nu + d} \right) \nabla_{\mathbf{x}_t} \log p(\mathbf{x}_t, t) \tag{136}$$

where $d_1 = \frac{1}{\sigma_t^2} \|\mathbf{x}_t - \mu_t \mathbf{x}_0\|_2^2$. Substituting the above result in the ODE dynamics, we obtain the reformulated ODE in Eq. 14.

$$\frac{d\mathbf{x}_t}{dt} = \frac{\dot{\mu}_t}{\mu_t} \mathbf{x}_t + \sigma_t^2 \left( \frac{\nu + d_1}{\nu + d} \right) \left[ \frac{\dot{\mu}_t}{\mu_t} - \frac{\dot{\sigma}_t}{\sigma_t} \right] \nabla_{\mathbf{x}} \log p(\mathbf{x}_t, t) \tag{137}$$

Since the term $d_1$ is data-dependent, we can estimate it during inference as $d_1 \approx d_1' = \frac{1}{\sigma_t^2}\|\mathbf{x}_t - \mu_t \boldsymbol{D}_\theta(\mathbf{x}_t, \sigma_t)\|_2^2$. Thus, the above ODE can be reformulated as,

$$\frac{d\mathbf{x}_t}{dt} = \frac{\dot{\mu}_t}{\mu_t}\mathbf{x}_t + \sigma_t^2\Big(\frac{\nu + d_1'}{\nu + d}\Big)\Big[\frac{\dot{\mu}_t}{\mu_t} - \frac{\dot{\sigma}_t}{\sigma_t}\Big]\nabla_\mathbf{x}\log p(\mathbf{x}_t, t) \tag{138}$$

**Tweedie's Estimate and Inverse Problems.** Given the perturbation kernel $q(\mathbf{x}_t|\mathbf{x}_0) = t_d(\mu_t \mathbf{x}_0, \sigma_t^2 \boldsymbol{I}_d, \nu)$, we have,

$$\mathbf{x}_t = \mu_t \mathbf{x}_0 + \sigma_t \frac{\epsilon}{\sqrt{\kappa}}, \quad \epsilon \sim \mathcal{N}(0, \boldsymbol{I}_d), \kappa \sim \chi^2(\nu)/\nu \tag{139}$$

It follows that,

$$\mathbf{x}_t = \mu_t \mathbb{E}[\mathbf{x}_0|\mathbf{x}_t] + \sigma_t \mathbb{E}\Big[\frac{\epsilon}{\sqrt{\kappa}}\Big|\mathbf{x}_t\Big] \tag{140}$$

$$\mathbb{E}[\mathbf{x}_0|\mathbf{x}_t] = \frac{1}{\mu_t}\left(\mathbf{x}_t - \sigma_t \mathbb{E}\Big[\frac{\epsilon}{\sqrt{\kappa}}\Big|\mathbf{x}_t\Big]\right) \tag{141}$$

$$\approx \frac{1}{\mu_t}\left(\mathbf{x}_t + \sigma_t^2\Big(\frac{\nu + d_1'}{\nu + d}\Big)\nabla_\mathbf{x}\log p(\mathbf{x}_t, t)\right) \tag{142}$$

which gives us an estimate of the predicted $\mathbf{x}_0$ at any time $t$. Moreover, the framework can also be extended for solving inverse problems using diffusion models. More specifically, given a conditional signal $\mathbf{y}$, the goal is to simulate the ODE,

$$\frac{d\mathbf{x}_t}{dt} = \frac{\dot{\mu}_t}{\mu_t}\mathbf{x}_t + \sigma_t^2\Big(\frac{\nu + d_1'}{\nu + d}\Big)\Big[\frac{\dot{\mu}_t}{\mu_t} - \frac{\dot{\sigma}_t}{\sigma_t}\Big]\nabla_\mathbf{x}\log p(\mathbf{x}_t|\mathbf{y}) \tag{143}$$

$$= \frac{\dot{\mu}_t}{\mu_t}\mathbf{x}_t + \sigma_t^2\Big(\frac{\nu + d_1'}{\nu + d}\Big)\Big[\frac{\dot{\mu}_t}{\mu_t} - \frac{\dot{\sigma}_t}{\sigma_t}\Big]\Big[w_t\nabla_\mathbf{x}\log p(\mathbf{y}|\mathbf{x}_t) + \nabla_\mathbf{x}\log p(\mathbf{x}_t)\Big] \tag{144}$$

where the above decomposition follows from $p(\mathbf{x}_t|y) \propto p(\mathbf{y}|\mathbf{x}_t)^{w_t} p(\mathbf{x}_t)$ and the weight $w_t$ represents the *guidance weight* of the distribution $p(\mathbf{y}|\mathbf{x}_t)$. The term $\log p(\mathbf{y}|\mathbf{x}_t)$ can now be approximated using existing posterior approximation methods in inverse problems (Chung et al., 2022; Song et al., 2022b; Mardani et al., 2023; Pandey et al., 2024b)

## A.11    CONNECTIONS TO ROBUST STATISTICAL ESTIMATORS

Here, we derive the expression for the gradient of the $\gamma$-power divergence between the forward and the reverse posteriors (denoted by $q$ and $p_\theta$, respectively for notational convenience) i.e., $\nabla_\theta D_\gamma(q \parallel p_\theta)$. By the definition of the $\gamma$-power divergence,

$$D_\gamma(q \parallel p_\theta) = \frac{1}{\gamma}\big[\mathcal{C}_\gamma(q, p_\theta) - \mathcal{H}_\gamma(q)\big], \quad \gamma \in (-1, 0) \cup (0, \infty) \tag{145}$$

$$\mathcal{H}_\gamma(p) = -\|p\|_{1+\gamma} = -\left(\int p(\mathbf{x})^{1+\gamma}d\mathbf{x}\right)^{\frac{1}{1+\gamma}} \qquad \mathcal{C}_\gamma(q, p) = -\int q(\mathbf{x})\left(\frac{p(\mathbf{x})}{\|p\|_{1+\gamma}}\right)^\gamma d\mathbf{x} \tag{146}$$

Therefore,

$$\nabla_\theta D_\gamma(q \parallel p_\theta) = \frac{1}{\gamma}\nabla_\theta\mathcal{C}_\gamma(q, p_\theta) \tag{147}$$

Consequently, we next derive an expression for $\nabla_\theta \mathcal{C}_\gamma(q, p_\theta)$.

$$\nabla_\theta \mathcal{C}_\gamma(q, p_\theta) = -\nabla_\theta \int q(\mathbf{x}) \left( \frac{p_\theta(\mathbf{x})}{\|p_\theta\|_{1+\gamma}} \right)^\gamma d\mathbf{x} \tag{148}$$

$$= -\gamma \int q(\mathbf{x}) \left( \frac{p_\theta(\mathbf{x})}{\|p_\theta\|_{1+\gamma}} \right)^{\gamma-1} \nabla_\theta \left( \frac{p_\theta(\mathbf{x})}{\|p_\theta\|_{1+\gamma}} \right) d\mathbf{x} \tag{149}$$

$$= -\gamma \int q(\mathbf{x}) \left( \frac{p_\theta(\mathbf{x})}{\|p_\theta\|_{1+\gamma}} \right)^{\gamma-1} \frac{\|p_\theta\|_{1+\gamma} \nabla_\theta p_\theta(\mathbf{x}) - p_\theta(\mathbf{x}) \nabla_\theta \|p_\theta\|_{1+\gamma}}{\|p_\theta\|_{1+\gamma}^2} d\mathbf{x} \tag{150}$$

From the definition of $\|p_\theta\|_{1+\gamma}$,

$$\|p_\theta\|_{1+\gamma} = \left( \int p_\theta(\mathbf{x})^{1+\gamma} d\mathbf{x} \right)^{\frac{1}{1+\gamma}} \tag{151}$$

$$\nabla_\theta \|p_\theta\|_{1+\gamma} = \nabla_\theta \left( \int p_\theta(\mathbf{x})^{1+\gamma} d\mathbf{x} \right)^{\frac{1}{1+\gamma}} \tag{152}$$

$$= \frac{1}{1+\gamma} \left( \int p_\theta(\mathbf{x})^{1+\gamma} d\mathbf{x} \right)^{-\frac{\gamma}{1+\gamma}} \int \nabla_\theta \left( p_\theta(\mathbf{x})^{1+\gamma} \right) d\mathbf{x} \tag{153}$$

$$= \frac{1}{1+\gamma} \left( \int p_\theta(\mathbf{x})^{1+\gamma} d\mathbf{x} \right)^{-\frac{\gamma}{1+\gamma}} (1+\gamma) \int p_\theta(\mathbf{x})^\gamma \nabla_\theta p_\theta(\mathbf{x}) d\mathbf{x} \tag{154}$$

$$\nabla_\theta \|p_\theta\|_{1+\gamma} = \left( \int p_\theta(\mathbf{x})^{1+\gamma} d\mathbf{x} \right)^{-\frac{\gamma}{1+\gamma}} \int p_\theta(\mathbf{x})^\gamma \nabla_\theta p_\theta(\mathbf{x}) d\mathbf{x} \tag{155}$$

$$= \frac{\|p_\theta\|_{1+\gamma}}{\left( \int p_\theta(\mathbf{x})^{1+\gamma} d\mathbf{x} \right)} \int p_\theta(\mathbf{x})^\gamma \nabla_\theta p_\theta(\mathbf{x}) d\mathbf{x} \tag{156}$$

$$= \frac{\|p_\theta\|_{1+\gamma}}{\left( \int p_\theta(\mathbf{x})^{1+\gamma} d\mathbf{x} \right)} \int p_\theta(\mathbf{x})^{1+\gamma} \nabla_\theta \log p_\theta(\mathbf{x}) d\mathbf{x} \tag{157}$$

$$= \|p_\theta\|_{1+\gamma} \int \underbrace{\frac{p_\theta(\mathbf{x})^{1+\gamma}}{\left( \int p_\theta(\mathbf{x})^{1+\gamma} d\mathbf{x} \right)}}_{=\tilde{p}_\theta(\mathbf{x})} \nabla_\theta \log p_\theta(\mathbf{x}) d\mathbf{x} \tag{158}$$

$$= \|p_\theta\|_{1+\gamma} \mathbb{E}_{\tilde{p}_\theta(\mathbf{x})} [\nabla_\theta \log p_\theta(\mathbf{x})] \tag{159}$$

where we denote $\tilde{p}_\theta(\mathbf{x}) = \frac{p_\theta(\mathbf{x})^{1+\gamma}}{\left( \int p_\theta(\mathbf{x})^{1+\gamma} d\mathbf{x} \right)}$ for notational convenience. Substituting the above result in Eq. 150, we have the following simplification,

**Algorithm 3:** Training (t-Flow)

1: **repeat**
2:    $\mathbf{x}_1 \sim p(\mathbf{x}_1)$
3:    $t \sim \text{Uniform}(\{1, \dots, T\})$
4:    $\mu_t = t, \sigma_t = 1 - t$
5:    $\mathbf{x}_t = \mu_t \mathbf{x}_1 + \sigma_t \mathbf{n}, \quad \mathbf{n} \sim t_d(0, \boldsymbol{I}_d, \nu)$
6:    Take gradient descent step on
7:       $\nabla_{\boldsymbol{\theta}} \|\mathbf{n} - \boldsymbol{\epsilon_\theta}(\mathbf{x}_t, \sigma_t)\|^2$
8: **until** converged

**Algorithm 4:** Sampling (t-Flow)

1: **sample** $\mathbf{x}_0 \sim t_d(0, \boldsymbol{I}_d, \nu)$
2: **for** $i \in \{0, \dots, N-1\}$ **do**
3:    $\boldsymbol{d}_i \leftarrow (\mathbf{x}_i - \boldsymbol{\epsilon_\theta}(\mathbf{x}_i; \sigma_{t_i}))/t_i$
4:    $\mathbf{x}_{i+1} \leftarrow \mathbf{x}_i + (t_{i+1} - t_i)\boldsymbol{d}_i$
5:    **if** $t_{i+1} \neq 0$ **then**
6:       $\boldsymbol{d}_i \leftarrow (\mathbf{x}_{i+1} - \boldsymbol{\epsilon_\theta}(\mathbf{x}_{i+1}; \sigma_{t_{i+1}}))/t_{i+1}$
7:       $\mathbf{x}_{i+1} \leftarrow \mathbf{x}_i + (t_{i+1} - t_i)(\frac{1}{2}\boldsymbol{d}_i + \frac{1}{2}\boldsymbol{d}_i')$
8:    **end if**
9: **end for**
10: **return** $\mathbf{x}_N$

Figure 5: Training and Sampling algorithms for t-Flow. Our proposed method requires minimal code updates (indicated with blue) over traditional Gaussian flow models and converges to the latter as $\nu \to \infty$.

$$\nabla_\theta \mathcal{C}_\gamma(q, p_\theta) = -\gamma \int q(\mathbf{x}) \left( \frac{p_\theta(\mathbf{x})}{\|p_\theta\|_{1+\gamma}} \right)^{\gamma-1} \frac{\|p_\theta\|_{1+\gamma} \nabla_\theta p_\theta(\mathbf{x}) - p_\theta(\mathbf{x})\nabla_\theta \|p_\theta\|_{1+\gamma}}{\|p_\theta\|_{1+\gamma}^2} d\mathbf{x} \tag{160}$$

$$= -\gamma \int q(\mathbf{x}) \left( \frac{p_\theta(\mathbf{x})}{\|p_\theta\|_{1+\gamma}} \right)^{\gamma-1} \frac{\|p_\theta\|_{1+\gamma} \nabla_\theta p_\theta(\mathbf{x}) - p_\theta(\mathbf{x}) \|p_\theta\|_{1+\gamma} \mathbb{E}_{\tilde{p}_\theta(\mathbf{x})}[\nabla_\theta \log p_\theta(\mathbf{x})]}{\|p_\theta\|_{1+\gamma}^2} d\mathbf{x} \tag{161}$$

$$= -\gamma \int q(\mathbf{x}) \left( \frac{p_\theta(\mathbf{x})}{\|p_\theta\|_{1+\gamma}} \right)^{\gamma-1} \frac{\nabla_\theta p_\theta(\mathbf{x}) - p_\theta(\mathbf{x})\mathbb{E}_{\tilde{p}_\theta(\mathbf{x})}[\nabla_\theta \log p_\theta(\mathbf{x})]}{\|p_\theta\|_{1+\gamma}} d\mathbf{x} \tag{162}$$

$$= -\gamma \int q(\mathbf{x}) \left( \frac{p_\theta(\mathbf{x})}{\|p_\theta\|_{1+\gamma}} \right)^{\gamma-1} \frac{p_\theta(\mathbf{x})\nabla_\theta \log p_\theta(\mathbf{x}) - p_\theta(\mathbf{x})\mathbb{E}_{\tilde{p}_\theta(\mathbf{x})}[\nabla_\theta \log p_\theta(\mathbf{x})]}{\|p_\theta\|_{1+\gamma}} d\mathbf{x} \tag{163}$$

$$\tag{164}$$

$$\nabla_\theta \mathcal{C}_\gamma(q, p_\theta) = -\gamma \int q(\mathbf{x}) \left( \frac{p_\theta(\mathbf{x})}{\|p_\theta\|_{1+\gamma}} \right)^{\gamma} \left( \nabla_\theta \log p_\theta(\mathbf{x}) - \mathbb{E}_{\tilde{p}_\theta(\mathbf{x})}[\nabla_\theta \log p_\theta(\mathbf{x})] \right) d\mathbf{x} \tag{165}$$

Plugging this result in Eq. 147, we have the following result,

$$\nabla_\theta D_\gamma(q \parallel p_\theta) = \frac{1}{\gamma}\nabla_\theta \mathcal{C}_\gamma(q, p_\theta) = -\int q(\mathbf{x}) \left( \frac{p_\theta(\mathbf{x})}{\|p_\theta\|_{1+\gamma}} \right)^{\gamma} \left( \nabla_\theta \log p_\theta(\mathbf{x}) - \mathbb{E}_{\tilde{p}_\theta(\mathbf{x})}[\nabla_\theta \log p_\theta(\mathbf{x})] \right) d\mathbf{x} \tag{166}$$

This completes the proof. Intuitively, the second term inside the integral in Eq. 166 ensures *unbiasedness* of the gradients. Therefore, the scalar coefficient $\gamma$ controls the weighting on the likelihood gradient and can be set accordingly to ignore or model outliers when modeling the data distribution.

## B   EXTENSION TO FLOWS

Here, we discuss an extension of our framework to flow matching models (Albergo et al., 2023; Lipman et al., 2023) with a Student-t base distribution. More specifically, we define a *straight-line* flow of the form,

$$\mathbf{x}_t = t\mathbf{x}_1 + (1 - t)\frac{\epsilon}{\sqrt{\kappa}}, \quad \epsilon \sim \mathcal{N}(0, \boldsymbol{I}_d), \kappa \sim \chi^2(\nu)/\nu \tag{167}$$

where $\mathbf{x}_1 \sim p(\mathbf{x}_1)$. Intuitively, at a given time $t$, the flow defined in Eqn. 167 linearly interpolates between data and Student-t noise. Following Albergo et al. (2023), the conditional vector field which induces this interpolant can be specified as (proof in App. A.8)

$$\frac{d\mathbf{x}_t}{dt} = \boldsymbol{b}(\mathbf{x}_t, t) = \frac{\mathbf{x}_t - \mathbb{E}\left[\frac{\epsilon}{\sqrt{\kappa}} \big| \mathbf{x}_t\right]}{t}. \tag{168}$$

We estimate $\mathbb{E}\left[\frac{\epsilon}{\sqrt{\kappa}} \big| \mathbf{x}_t\right]$ by minimizing the objective

$$\mathcal{L}(\theta) = \mathbb{E}_{\mathbf{x}_0 \sim p(\mathbf{x}_0)} \mathbb{E}_{t \sim \mathcal{U}[0,1]} \mathbb{E}_{\epsilon \sim \mathcal{N}(0,\boldsymbol{I}_d)} \mathbb{E}_{\kappa \sim \chi^2(\nu)/\nu} \left[ \left\| \epsilon_{\boldsymbol{\theta}}\left(t\mathbf{x}_0 + (1-t)\frac{\epsilon}{\sqrt{\kappa}}, t\right) - \frac{\epsilon}{\sqrt{\kappa}} \right\|_2^2 \right]. \tag{169}$$

We refer to this flow setup as *t-Flow*. To generate samples from our model, we simulate the ODE in Eq. 168 using Heun's solver. Figure 5 illustrates the ease of transitioning from a Gaussian flow to t-Flow. Similar to Gaussian diffusion, transitioning to t-Flow requires very few lines of code change, making our method readily compatible with existing implementations of flow models.

## C  EXPERIMENTAL SETUP

### C.1  UNCONDITIONAL MODELING

#### C.1.1  HRRR DATASET

We adopt the High-Resolution Rapid Refresh (HRRR) (Dowell et al., 2022) dataset, which is an operational archive of the US km-scale forecasting model. Among multiple dynamical variables in the dataset that exhibit heavy-tailed behavior, based on their dynamic range, we only consider the *Vertically Integrated Liquid* (VIL) and *Vertical Wind Velocity* at level 20 (denoted as w20) channels. How to cope with the especially non-Gaussian nature of such physical variables on km-scales, represents an entire subfield of climate model subgrid-scale parameterization (e.g., Guo et al. (2015)). We only use data for the years $2019 - 2020$ for training (17.4k samples) and the data for 2021 (8.7k samples) for testing; data before 2019 are avoided owing to non-stationaries associated with periodic version changes of the HRRR. Lastly, while the HRRR dataset spans the entire US, for simplicity, we work with regional crops of size $128 \times 128$ (corresponding to $384 \times 384$ km over the Central US). Unless specified otherwise, we perform z-score normalization using precomputed statistics as a preprocessing step. We do not perform any additional data augmentation.

#### C.1.2  BASELINES

**Baseline 1: EDM.** For standard Gaussian diffusion models, we use the recently proposed EDM (Karras et al., 2022) model, which shows strong empirical performance on various image synthesis benchmarks and has also been employed in recent work in weather forecasting and downscaling (Pathak et al., 2024; Mardani et al., 2024). To summarize, EDM employs the following denoising loss during training,

$$\mathcal{L}(\theta) \propto \mathbb{E}_{\mathbf{x}_0 \sim p(\mathbf{x}_0)} \mathbb{E}_{\sigma} \mathbb{E}_{\mathbf{n} \sim \mathcal{N}(0,\sigma^2 \boldsymbol{I}_d)} \left[ \lambda(\sigma) \| \boldsymbol{D}_{\boldsymbol{\theta}}(\mathbf{x}_0 + \mathbf{n}, \sigma) - \mathbf{x}_0 \|_2^2 \right] \tag{170}$$

where the noise levels $\sigma$ are usually sampled from a LogNormal distribution, $p(\sigma) = \text{LogNormal}(\pi_{\text{mean}}, \pi_{\text{std}}^2)$

**Baseline 2: EDM + Inverse CDF Normalization (INC).** It is commonplace to perform z-score normalization as a data pre-processing step during training. However, since heavy-tailed channels usually exhibit a high dynamic range, using z-score normalization for such channels cannot fully compensate for this range, especially when working with diverse channels in downstream tasks in weather modeling. An alternative could be to use a more *stronger* normalization scheme like Inverse CDF normalization, which essentially involves the following key steps:

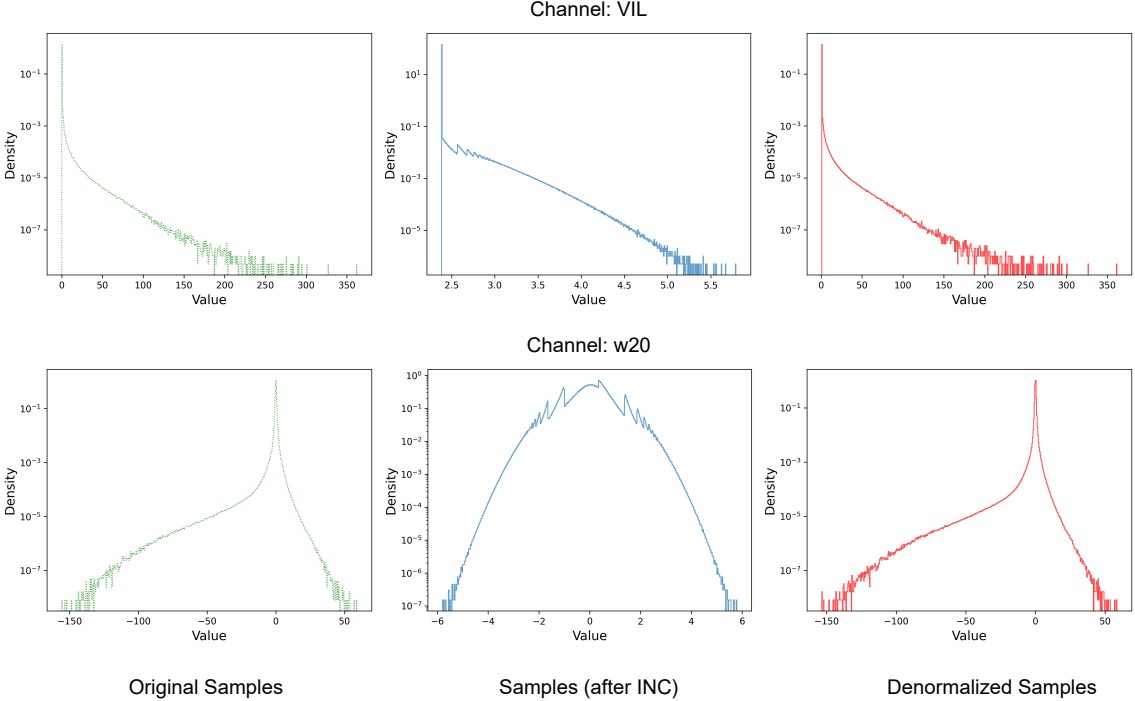

Figure 6: **Inverse CDF Normalization.** Using Inverse CDF Normalization (INC) can help reduce channel dynamic range during training while providing accurate denormalization. (Top Panel) INC applied to the Vertically Integrated Liquid (VIL) channel in the HRRR dataset. (Bottom Panel) INC applied to the Vertical Wind Velocity (w20) channel in the HRRR dataset.

1. Compute channel-wise 1-d histograms of the training data.

2. Compute channel-wise empirical CDFs from the 1-d histograms computed in Step 1.

3. Use the empirical CDFs from Step 2 to compute the CDF at each spatial location.

4. For each spatial location with a CDF value $p$, replace its value by the value obtained by applying the Inverse CDF operation under the standard Normal distribution.

Fig. 6 illustrates the effect of performing normalization under this scheme. As can be observed, using such a normalization scheme can greatly reduce the dynamic range of a given channel while offering reliable denormalization. Moreover, since our normalization scheme only affects data preprocessing, we leave the standard EDM model parameters unchanged for this baseline.

**Baseline 3: EDM + Per Channel-Preconditioning (PCP).** Another alternative to account for extreme values in the data (or high dynamic range) could be to instead add more heavy-tailed noise during training. This can be controlled by modulating the $\pi_{\mathrm{mean}}$ and $\pi_{\mathrm{std}}$ parameters based on the dynamic range of the channel under consideration. Recall that these parameters control the domain of noise levels $\sigma$ used during EDM model training. In this work, we use a simple heuristic to modulate these parameters based on the normalized channel dynamic range (denoted as $d$). More specifically, we set,

$$\pi_{\mathrm{mean}} = -1.2 + \alpha * \mathrm{RBF}(d, 1.0, \beta) \tag{171}$$

where RBF denotes the Radial Basis Function kernel with radius=1.0, parameter $\beta$ and a magnitude scaling factor $\alpha$. We keep $\pi_{\text{std}} = 1.2$ fixed for all channels. Intuitively, this implies that a higher normalized dynamic range (near 1.0) corresponds to sampling the noise levels $\sigma$ from a more heavy-tailed distribution during training. This is natural since a signal with a larger magnitude would need more amount of noise to convert it to noise during the forward diffusion process. In this work, we set $\alpha = 3.0$, $\beta = 2.0$, which yields $\pi_{\text{mean}}^{\text{vil}} = 1.8$ and $\pi_{\text{mean}}^{\text{w20}} = 0.453$ for the VIL and w20 channels, respectively. It is worth noting that, unlike the previous baseline, we use z-score normalization as a preprocessing step for this baseline. Lastly, we keep other EDM parameters during training and sampling fixed.

**Baseline 4. Gaussian Flow.** Since we also extend our framework to flow matching models (Albergo et al., 2023; Lipman et al., 2023), we also compare with a linear one-sided interpolant with a Gaussian base distribution. More specifically,

$$\mathbf{x}_t = t\mathbf{x}_1 + (1-t)\epsilon, \quad \epsilon \sim \mathcal{N}(0, \boldsymbol{I}_d) \tag{172}$$

Similar to t-Flow (Section B), we train the Gaussian flow with the following objective,

$$\mathcal{L}(\theta) \;\; = \;\; \mathbb{E}_{\mathbf{x}_0 \sim p(\mathbf{x}_0)} \mathbb{E}_{t \sim \mathcal{U}[0,1]} \mathbb{E}_{\epsilon \sim \mathcal{N}(0, \boldsymbol{I}_d)} \left[ \left\| \epsilon_{\boldsymbol{\theta}}(t\mathbf{x}_0 + (1-t)\epsilon, t) - \epsilon \right\|_2^2 \right]. \tag{173}$$

### C.1.3 EVALUATION

Here, we describe our scoring protocol used in Tables 2 and 3 in more detail.

**Kurtosis Ratio (KR).** Intuitively, sample kurtosis characterizes the heavy-tailed behavior of a distribution and represents the fourth-order moment. Higher kurtosis represents greater deviations from the central tendency, such as from outliers in the data. In this work, given samples from the underlying train/test set, we generate 20k samples from our model. We then flatten all the samples and compute empirical kurtosis for both the underlying samples from the train/test set (denoted as $k_{\text{data}}$) and our model (denoted as $k_{\text{sim}}$). The Kurtosis ratio is then computed as,

$$\text{KR} = \left| 1 - \frac{k_{\text{sim}}}{k_{\text{data}}} \right| \tag{174}$$

Lower values of this ratio imply a better estimation of the underlying sample kurtosis.

**Skewness Ratio (SR).** Intuitively, sample skewness represents the asymmetry of a tailed distribution and represents the third-order moment. In this work, given samples from the underlying train/test set, we generate 20k samples from our model. We then flatten all the samples and compute empirical skewness for both the underlying samples from the train/test set (denoted as $s_{\text{data}}$) and our model (denoted as $s_{\text{sim}}$). The Skewness ratio is then computed as,

$$\text{SR} = \left| 1 - \frac{s_{\text{sim}}}{s_{\text{data}}} \right| \tag{175}$$

Lower values of this ratio imply a better estimation of the underlying sample skewness.

**Kolmogorov-Smirnov 2-Sample Test (KS).** The KS (Massey, 1951) statistic measures the maximum difference between the CDFs of two distributions. For heavy-tailed distributions, evaluating the KS statistic at the tails could provide a useful measure of the efficacy of our model in estimating tails reliably. To evaluate the KS statistic at the tails, similar to prior metrics, we generate 20k samples from our model. We then flatten all samples in the generated and train/test sets, followed by retaining samples lying above the 99.9th percentile (quantifying right tails/extreme region) or below the 0.1th percentile (quantifying the left tail/extreme region). Lastly, we compute the KS statistic between the retained samples from the generated and the train/test sets individually for each tail and average the KS statistic values for both tails to obtain an average KS score. The final score estimates how well the model might capture both tails. As an exception,. for the VIL channel, we report KS scores only for the right tail due to the absence of a left tail in the underlying samples for this

| | Parameters | EDM (+INC, +PCP) | t-EDM | Flow/t-Flow |
|---|---|---|---|---|
| Preconditioner | $c_{\text{skip}}$ | $\sigma_{\text{data}}^2\big/\left(\sigma^2 + \sigma_{\text{data}}^2\right)$ | $\sigma_{\text{data}}^2\big/\left(\frac{\nu}{\nu-2}\sigma^2 + \sigma_{\text{data}}^2\right)$ | $0$ |
| | $c_{\text{out}}$ | $\sigma \cdot \sigma_{\text{data}}^2\big/\sqrt{\sigma^2 + \sigma_{\text{data}}^2}$ | $\sqrt{\frac{\nu}{\nu-2}}\,\sigma \cdot \sigma_{\text{data}}^2\big/\sqrt{\frac{\nu}{\nu-2}\sigma^2 + \sigma_{\text{data}}^2}$ | $1$ |
| | $c_{\text{in}}$ | $1/\sqrt{\sigma^2 + \sigma_{\text{data}}^2}$ | $1/\sqrt{\frac{\nu}{\nu-2}\sigma^2 + \sigma_{\text{data}}^2}$ | $1$ |
| | $c_{\text{noise}}$ | $\frac{1}{4}\log\sigma$ | $\frac{1}{4}\log\sigma$ | $\sigma$ |
| Training | $\sigma$ | $\log\sigma \sim \mathcal{N}(\pi_{\text{mean}}, \pi_{\text{std}}^2)$ | $\log\sigma \sim \mathcal{N}(\pi_{\text{mean}}, \pi_{\text{std}}^2)$ | $\sigma = 1-t,\, t \sim \mathcal{U}(0,1)$ |
| | $\mu_t$ | $1$ | $1$ | $t$ |
| | $\lambda(\sigma)$ | $1/c_{\text{out}}^2(\sigma)$ | $1/c_{\text{out}}^2(\sigma, \nu)$ | $1$ |
| | Loss | Eq. 170 | Eq. 13 | t-Flow - Eq. 169
Gaussian Flow - Eq. 173 |
| Sampling | Solver | Heun's (2nd order) | Heun's (2nd order) | Heun's (2nd order) |
| | ODE | Eq. 12, $\mathbf{x}_T \sim \mathcal{N}(0, \boldsymbol{I}_d)$ | Eq. 12, $\mathbf{x}_T \sim t_d(0, \boldsymbol{I}_d, \nu)$ | t-Flow: Eq. 168, $\mathbf{x}_0 \sim t_d(0, \boldsymbol{I}_d, \nu)$
Flow: Eq. 168, $\mathbf{x}_0 \sim \mathcal{N}(0, \boldsymbol{I}_d)$ |
| | Discretization | $\left(\sigma_{\max}^{\frac{1}{\rho}} + \frac{i}{N-1}\left[\sigma_{\min}^{\frac{1}{\rho}} - \sigma_{\max}^{\frac{1}{\rho}}\right]\right)^{\frac{1}{\rho}}$ | $\left(\sigma_{\max}^{\frac{1}{\rho}} + \frac{i}{N-1}\left[\sigma_{\min}^{\frac{1}{\rho}} - \sigma_{\max}^{\frac{1}{\rho}}\right]\right)^{\frac{1}{\rho}}$ | $\left(\sigma_{\max}^{\frac{1}{\rho}} + \frac{i}{N-1}\left[\sigma_{\min}^{\frac{1}{\rho}} - \sigma_{\max}^{\frac{1}{\rho}}\right]\right)^{\frac{1}{\rho}}$ |
| | Scaling: $\mu_t$ | $1$ | $1$ | $t$ |
| | Schedule: $\sigma_t$ | $t$ | $t$ | $1-t$ |
| Hyperparameters | $\sigma_{\text{data}}$ | $1.0$ | $1.0$ | N/A |
| | $\nu$ | $\infty$ | VIL: $\{3, 5, 7\}$
w20: $\{3, 5, 7\}$ | Flow: $\infty$
t-Flow ($\downarrow$)
VIL=$\{3, 5, 7\}$
w20=$\{5, 7, 9\}$ |
| | $\pi_{\text{mean}}, \pi_{\text{std}}$ | EDM: -1.2, 1.2
EDM (+INC) : -1.2, 1.2
EDM (+PCP) ($\downarrow$)
VIL: 1.8, 1.2
w20: 0.453, 1.2 | -1.2, 1.2 | N/A |
| | $\sigma_{\max}, \sigma_{\min}$ | 80, 0.002 | 80, 0.002 | 1.0, 0.01 |
| | NFE | 18 | 18 | 25 |
| | $\rho$ | 7 | 7 | 7 |

Table 5: Comparison between design choices and specific hyperparameters between EDM (Karras et al., 2022) (+ related baselines) and t-EDM (Ours, Section 3.5) for unconditional modeling (Section 4.1). INC: Inverse CDF Normalization baselines, PCP: Per-Channel Preconditioning baseline, VIL: Vertically Integrated Liquid channel in the HRRR dataset, w20: Vertical Wind Velocity at level 20 channel in the HRRR dataset, NFE: Number of Function Evaluations

channel (see Fig. 6 (first column) for 1-d intensity histograms for this channel). Lower values of the KS statistic imply better density assignment at the tails by the model.

**Histogram Comparisons.** As a qualitative metric, comparing 1-d intensity histograms between the generated and the original samples from the train/test set can serve as a reliable proxy to assess the tail estimation capabilities of all models.

### C.1.4 DENOISER ARCHITECTURE

We use the DDPM++ architecture from (Karras et al., 2022; Song et al., 2020). We set the base channel multiplier to 32 and the per-resolution channel multiplier to [1,2,2,4,4] with self-attention at resolution 16. The rest of the hyperparameters remain unchanged from Karras et al. (2022), which results in a model size of around 12M parameters.

### C.1.5 TRAINING

We adopt the same training hyperparameters from Karras et al. (2022) for training all models. Model training is distributed across 4 DGX nodes, each with 8 A100 GPUs, with a total batch size of 512. We train all

| Parameter | Model Levels | Height Levels (m) |
|---|---|---|
| Zonal Wind (u) | 1,2,3,4,5,6,7,8,9,10,11,13,15,20 | 10 |
| Meridonal Wind (v) | 1,2,3,4,5,6,7,8,9,10,11,13,15,20 | 10 |
| Geopotential Height (z) | 1,2,3,4,5,6,7,8,9,10,11,13,15,20 | None |
| Humidity (q) | 1,2,3,4,5,6,7,8,9,10,11,13,15,20 | None |
| Pressure (p) | 1,2,3,4,5,6,7,8,9,10,11,13,15,20 | None |
| Temperature (t) | 1,2,3,4,5,6,7,8,9,10,11,13,15,20 | 2 |
| Radar Reflectivity (refc) | N/A | Integrated |

Table 6: Parameters in the HRRR dataset used for conditional modeling tasks.

models for a maximum budget of 60Mimg and select the best-performing model in terms of qualitative 1-D histogram comparisons.

### C.1.6 SAMPLING

For the EDM and related baselines (INC, PCP), we use the ODE solver presented in Karras et al. (2022). For the t-EDM models, as presented in Section 3.5, our sampler is the same as EDM with the only difference in the sampling of initial latents from a Student-t distribution instead (See Fig. 2 (Right)). For Flow and t-Flow, we numerically simulate the ODE in Eq. 168 using the 2nd order Heun's solver with the timestep discretization proposed in Karras et al. (2022). For evaluation, we generate 20k samples from each model.

We summarize our experimental setup in more detail for unconditional modeling in Table 5.

### C.1.7 EXTENDED RESULTS ON UNCONDITIONAL MODELING

**Sample Visualization.** We visualize samples generated from the t-EDM and t-Flow models for the VIL and w20 channels in Figs. 9-12

**Visualization of 1-d histograms.** Similar to Fig. 3, we present additional results on histogram comparisons between different baselines and our proposed methods for the VIL and w20 channels in Figs. 13 and 14.

**Is PCP enough for capturing heavy tails?** In Table 2, we observe that the EDM + PCP baseline outperforms t-EDM in some scenarios. A natural question is whether a dynamic preconditioning scheme based on the dynamic range of a channel is enough for modeling heavy-tailed data. To present some insight into this observation, we train an unconditional EDM + PCP model on 10 channels in the HRRR dataset selected on the basis of the top 10 channels with the highest dynamic range and heavier tails. Our EDM training setup is the same as other experiments presented in Section 4 with the exception that the input has 10 channels instead of a single channel. The updated $\pi_{\mathrm{mean}}$ for all 10 channels is computed based on the formulation in Eqn. 171. Consequently, this is a more difficult modeling task than presented in the main text in Table 2. We compare the generated histograms between the 10-channel run with PCP and the single-channel run with PCP for the VIL and w20 channels in Fig. 7. As can be observed, when scaling in terms of the number of channels, the performance of EDM + PCP takes a big hit. This suggests that merely adjusting the noise schedule during training using heuristics like PCP is not sufficient to resolve the fundamental limitation of Gaussian diffusion models to estimate heavy tails accurately.

### C.2 CONDITIONAL MODELING

### C.2.1 HRRR DATASET FOR CONDITIONAL MODELING

Similar to unconditional modeling (See App. C.1.1), we use the HRRR dataset for conditional modeling at the 128 x 128 resolution. We train the model on HRRR forecast fields 1-hour post-analysis to allow for

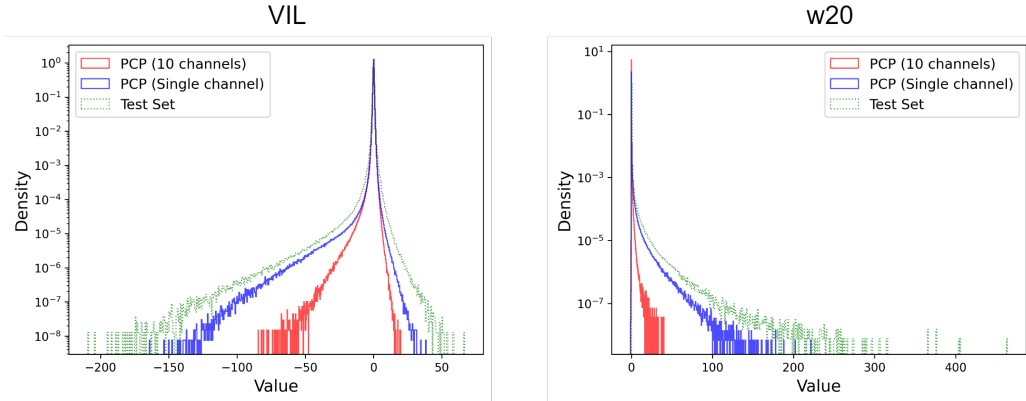

Figure 7: Sample 1-d histogram comparison between EDM + PCP with 10 channels and single-channel training on the test set for the Vertically Integrated Liquid (VIL) channel and Vertical Wind Velocity (w20) channel. Tail estimation suffers when scaling up training in terms of the number of channels in the input. PCP: Per-Channel Preconditioning

some model spin-up following data assimilation. More specifically, for a lead time of 1hr, we sample (input, output) pairs at time $\tau$ and $\tau + 1$, respectively. For the input, at time $\tau$, we use a state vector consisting of a combination of 86 atmospheric channels (including the channel to be predicted at time $\tau$), which are summarized in Table 6. For the output, at time $\tau + 1$, we use either the Vertically Integrated Liquid (VIL) or Vertical Wind Velocity at level 20 (w20) channels, depending on the prediction task. Unless specified otherwise, we perform z-score normalization using precomputed statistics as a preprocessing step without any additional data augmentation.

### C.2.2 BASELINES

We adopt the standard EDM (Karras et al., 2022) for conditional modeling as our baseline.

### C.2.3 DENOISER ARCHITECTURE

We use the DDPM++ architecture from (Karras et al., 2022; Song et al., 2020). We set the base channel multiplier to 32 and the per-resolution channel multiplier to [1,2,2,4,4] with self-attention at resolution 16. Additionally, our noisy state $\mathbf{x}$ is channel-wise concatenated with an 86-channel conditioning signal, increasing the total number of input channels in the denoiser to 87. The number of output channels remains 1 since we are predicting only a single VIL/w20 channel. However, the increase in the number of parameters is minimal since only the first convolutional layer in the denoiser is affected. Therefore, our denoiser is around 12M parameters. The rest of the hyperparameters remain unchanged from Karras et al. (2022).

### C.2.4 TRAINING

We adopt the same training hyperparameters from Karras et al. (2022) for training all conditional models. Model training is distributed across 4 DGX nodes, each with 8 A100 GPUs, with a total batch size of 512. We train all models for a maximum budget of 60Mimg.

### C.2.5 SAMPLING

For both EDM and t-EDM models, we use the ODE solver presented in Karras et al. (2022). For the t-EDM models, as presented in Section 3.5, our sampler is the same as EDM with the only difference in the sampling

of initial latents from a Student-t distribution instead (See Fig. 2 (Right)). For a given input conditioning state, we generate an ensemble of predictions of size 16 by randomly initializing our ODE solver with different random seeds. All other sampling parameters remain unchanged from our unconditional modeling setup (see App. C.1.6).

### C.2.6 EVALUATION

**Root Mean Square Error (RMSE)**. is a standard evaluation metric used to measure the difference between the predicted values and the true values Chai & Draxler (2014). In the context of our problem, let $\boldsymbol{x}$ be the true target and $\hat{\boldsymbol{x}}$ be the predicted value. The RMSE is defined as:

$$\text{RMSE} = \sqrt{\mathbb{E}\left[\|\boldsymbol{x} - \hat{\boldsymbol{x}}\|^2\right]}.$$

This metric captures the average magnitude of the residuals, i.e., the difference between the predicted and true values. A lower RMSE indicates better model performance, as it suggests the predicted values are closer to the true values on average. RMSE is sensitive to large errors, making it an ideal choice for evaluating models where minimizing large deviations is critical.

**Continuous Ranked Probability Score (CRPS)**. is a measure used to evaluate probabilistic predictions Wilks (2011). It compares the entire predicted distribution $F(\hat{\boldsymbol{x}})$ with the observed data point $\boldsymbol{x}$. For a probabilistic forecast with cumulative distribution function (CDF) $F$, and the true value $\boldsymbol{x}$, the CRPS can be formulated as follows:

$$\text{CRPS}(F, \boldsymbol{x}) = \int_{-\infty}^{\infty} \left(F(y) - \mathbb{I}(y \geqslant \boldsymbol{x})\right)^2 \, dy,$$

where $\mathbb{I}(\cdot)$ is the indicator function. Unlike RMSE, CRPS provides a more comprehensive evaluation of both the location and spread of the predicted distribution. A lower CRPS indicates a better match between the forecast distribution and the observed data. It is especially useful for probabilistic models that output a distribution rather than a single-point prediction.

**Spread-Skill Ratio (SSR)**. is used to assess over/under-dispersion in probabilistic forecasts. Spread measures the uncertainty in the ensemble forecasts and can be represented by computing the standard deviation of the ensemble members. Skill represents the accuracy of the mean of the ensemble forecasts and can be represented by computing the RMSE between the ensemble mean and the observations.

**Scoring Criterion.** Since CRPS and SSR metrics are based on predicting ensemble forecasts for a given input state, we predict an ensemble of size 16 for 4000 samples from the VIL/w20 test set. We then enumerate window sizes of 16 x 16 across the spatial resolution of the generated sample (128 x 128). Since the VIL channel is quite sparse, we filter out windows with a maximum value of less than a threshold (1.0 for VIL) and compute the CRPS, SSR, and RMSE metrics for all remaining windows. As an additional caveat, we note that while it is common to roll out trajectories for weather forecasting, in this work, we only predict the target at the immediate next time step.

### C.2.7 EXTENDED RESULTS ON CONDITIONAL MODELING

**Sample Visualization.** We visualize samples generated from the t-EDM and t-Flow models for the VIL and w20 channels in Figs. 9 and 10

**On t-EDM stability for Autoregressive Rollouts.** While we demonstrate the effectiveness of t-EDM for the conditional task of predicting the next frame, in practice, it is more useful to perform autoregressive

| | Parameters | EDM | t-EDM |
|---|---|---|---|
| Preconditioner | $c_{\text{skip}}$ | $\sigma_{\text{data}}^2\big/\left(\sigma^2 + \sigma_{\text{data}}^2\right)$ | $\sigma_{\text{data}}^2\big/\left(\frac{\nu}{\nu-2}\sigma^2 + \sigma_{\text{data}}^2\right)$ |
| | $c_{\text{out}}$ | $\sigma \cdot \sigma_{\text{data}}^2\big/\sqrt{\sigma^2 + \sigma_{\text{data}}^2}$ | $\sqrt{\frac{\nu}{\nu-2}}\sigma \cdot \sigma_{\text{data}}^2\big/\sqrt{\frac{\nu}{\nu-2}\sigma^2 + \sigma_{\text{data}}^2}$ |
| | $c_{\text{in}}$ | $1\big/\sqrt{\sigma^2 + \sigma_{\text{data}}^2}$ | $1\big/\sqrt{\frac{\nu}{\nu-2}\sigma^2 + \sigma_{\text{data}}^2}$ |
| | $c_{\text{noise}}$ | $\frac{1}{4}\log\sigma$ | $\frac{1}{4}\log\sigma$ |
| Training | $\sigma$ | $\log\sigma \sim \mathcal{N}(\pi_{\text{mean}}, \pi_{\text{std}}^2)$ | $\log\sigma \sim \mathcal{N}(\pi_{\text{mean}}, \pi_{\text{std}}^2)$ |
| | $s(t)$ | 1 | 1 |
| | $\lambda(\sigma)$ | $1/c_{\text{out}}^2(\sigma)$ | $1/c_{\text{out}}^2(\sigma, \nu)$ |
| | Loss | Eq. 2 in Karras et al. (2022) | Eq. 13 |
| Sampling | Solver | Heun's (2nd order) | Heun's (2nd order) |
| | ODE | Eq. 12, $\mathbf{x}_T \sim \mathcal{N}(0, \boldsymbol{I}_d)$ | Eq. 12, $\mathbf{x}_T \sim t_d(0, \boldsymbol{I}_d, \nu)$ |
| | Discretization | $\left(\sigma_{\max}^{\frac{1}{\rho}} + \frac{i}{N-1}\left[\sigma_{\min}^{\frac{1}{\rho}} - \sigma_{\max}^{\frac{1}{\rho}}\right]\right)^{\frac{1}{\rho}}$ | $\left(\sigma_{\max}^{\frac{1}{\rho}} + \frac{i}{N-1}\left[\sigma_{\min}^{\frac{1}{\rho}} - \sigma_{\max}^{\frac{1}{\rho}}\right]\right)^{\frac{1}{\rho}}$ |
| | Scaling: $s(t)$ | 1 | 1 |
| | Schedule: $\sigma(t)$ | t | t |
| Hyperparameters | $\sigma_{\text{data}}$ | 1.0 | 1.0 |
| | $\nu$ | $\infty$ | $\mathbf{x}_1 = 20, \mathbf{x}_2 \in \{4, 7, 10\}$ |
| | $\pi_{\text{mean}}, \pi_{\text{std}}$ | -1.2, 1.2 | -1.2, 1.2 |
| | $\sigma_{\max}, \sigma_{\min}$ | 80, 0.002 | 80, 0.002 |
| | NFE | 18 | 18 |
| | $\rho$ | 7 | 7 |

Table 7: Comparison between design choices and specific hyperparameters between EDM (Karras et al., 2022) and t-EDM (Ours, Section 3.5) for the Toy dataset analysis in Fig. 1. NFE: Number of Function Evaluations

rollouts for downstream applications like forecasting. Therefore, we include qualitative results to showcase the stability of t-EDM when performing autoregressive rollouts in Fig. 15

## C.3 TOY EXPERIMENTS

**Dataset.** For the toy illustration in Fig. 1, we work with the Neals Funnel dataset (Neal, 2003), which is commonly used in the MCMC literature (Brooks et al., 2011) due to its challenging geometry. The underlying generative process for Neal's funnel can be specified as follows:

$$p(\mathbf{x}_1, \mathbf{x}_2) = \mathcal{N}(\mathbf{x}_1; 0, 3)\mathcal{N}(\mathbf{x}_2; 0, \exp\mathbf{x}_1/2). \tag{176}$$

For training, we randomly generate 1M samples from the generative process in Eq. 176 and perform z-score normalization as a pre-processing step.

**Baselines and Models.** For the standard Gaussian diffusion model baseline (2nd column in Fig. 1), we use EDM with standard hyperparameters as presented in Karras et al. (2022). Consequently, for heavy-tailed diffusion models (columns 3-5 in Fig. 1), we use the *t-EDM* instantiation of our framework as presented in Section 3.5. Since the hyperparameter $\nu$ is key in our framework, we tune $\nu$ for each individual dimension in our toy experiments. We fix $\nu$ to 20 for the $\mathbf{x}_1$ dimension and vary it between $\nu \in \{4, 7, 10\}$ to illustrate controllable tail estimation along the $\mathbf{x}_2$ dimension.

**Denoiser Architecture.** For modeling the underlying denoiser $\boldsymbol{D}_\theta(\mathbf{x}, \sigma)$, we use a simple MLP for all toy models. At the input, we concatenate the 2-dimensional noisy state vector $\mathbf{x}$ with the noise level $\sigma$. We use two hidden layers of size 64, followed by a linear output layer. This results in around 8.5k parameters for the denoiser. We share the same denoiser architecture across all toy models.

| | Method | $\nu$ | KR $\downarrow$ | SR $\downarrow$ | KS $\downarrow$ |
|---|---|---|---|---|---|
| | | | Toy Dataset (Neal's Funnel) | | |
| Baselines | EDM | $\infty$ | 0.909 | 2.334 | 0.234 |
| Ours | t-EDM | 4 | **0.393** | **1.584** | **0.051** |
| | t-EDM | 7 | 0.795 | 3.671 | 0.194 |
| | t-EDM | 10 | 0.863 | 2.198 | 0.261 |

Table 8: Quantitative comparison between EDM and t-EDM on the Neals Funnel 2-d toy dataset. For all metrics, lower is better. Values in **bold** indicate the best results in a column.

**Training.** We optimize the training objective in Eq. 13 for both t-EDM and EDM (See Fig. 2 (Left)). Our training hyperparameters are the same as proposed in Karras et al. (2022). We train all toy models for a fixed duration of 30M samples and choose the last checkpoint for evaluation.

**Sampling.** For the EDM baseline, we use the ODE solver presented in Karras et al. (2022). For the t-EDM models, as presented in Section 3.5, our sampler is the same as EDM with the only difference in the sampling of initial latents from a Student-t distribution instead (See Fig. 2 (Right)). For visualization of the generated samples in Fig. 1, we generate 1M samples for each model.

Overall, our experimental setup for the toy dataset analysis in Fig. 1 is summarized in Table 7.

**Quantitative Results on Neals Funnel**. We present quantitative results on the Neals funnel dataset in Table 8. t-EDM outperforms EDM on all three metrics, indicating better tail estimation, which also supports our qualitative findings in Fig. 1.

# D    OPTIMAL NOISE SCHEDULE DESIGN

In this section we discuss a strategy for choosing the parameter $\sigma_{\max}$ (denoted by $\sigma$ in this section for notational brevity) in a more principled manner as compared to EDM (Karras et al., 2022). More specifically, our approach involves directly estimating $\sigma$ from the empirically observed samples which circumvents the need to rely on ad-hoc choices of this parameter which can affect downstream sampler performance.

The main idea behind our approach is minimizing the statistical *mutual information* between datapoints from the underlying data distribution, $\mathbf{x}_0 \sim p_{\text{data}}$, and their noisy counterparts $\mathbf{x}_\sigma \sim p(\mathbf{x}_\sigma)$. While a trivial (and non-practical) way to achieve this objective could be to set a large enough $\sigma$ i.e. $\sigma \to \infty$, we instead minimize the mutual information $I(\mathbf{x}_0, \mathbf{x}_\sigma)$ while ensuring the magnitude of $\sigma$ to be as small as possible. Formally, our objective can be defined as,

$$\min_{\sigma^2} \sigma^2 \quad \text{subject to} \quad I(\mathbf{x}_0, \mathbf{x}_\sigma) = 0 \tag{177}$$

As we will discuss later, minimizing this constrained objective provides a more principled way to obtain $\sigma$ from the underlying data statistics for a specific level of mutual information desired by the user. Next, we first simplify the form of $I(\mathbf{x}_0, \mathbf{x}_\sigma)$, followed by a discussion on the estimation of $\sigma$ in the context of EDM and t-EDM. We also extend to the case of non-i.i.d noise.

**Simplification of $I(\mathbf{x}_0, \mathbf{x}_\sigma)$.** The mutual information $I(\mathbf{x}_0, \mathbf{x}_\sigma)$ can be stated and simplified as follows,

$$I(\mathbf{x}_0, \mathbf{x}_\sigma) = D_{\mathrm{KL}}(p(\mathbf{x}_0, \mathbf{x}_\sigma) \,\|\, p(\mathbf{x}_0)p(\mathbf{x}_\sigma)) \tag{178}$$

$$= \int p(\mathbf{x}_0, \mathbf{x}_\sigma) \log \frac{p(\mathbf{x}_0, \mathbf{x}_\sigma)}{p(\mathbf{x}_0)p(\mathbf{x}_\sigma)} d\mathbf{x}_0 d\mathbf{x}_\sigma \tag{179}$$

$$= \int p(\mathbf{x}_0, \mathbf{x}_\sigma) \log \frac{p(\mathbf{x}_\sigma|\mathbf{x}_0)}{p(\mathbf{x}_\sigma)} d\mathbf{x}_0 d\mathbf{x}_\sigma \tag{180}$$

$$= \int p(\mathbf{x}_\sigma|\mathbf{x}_0)p(\mathbf{x}_0) \log \frac{p(\mathbf{x}_\sigma|\mathbf{x}_0)}{p(\mathbf{x}_\sigma)} d\mathbf{x}_0 d\mathbf{x}_\sigma \tag{181}$$

$$= \mathbb{E}_{\mathbf{x}_0 \sim p(\mathbf{x}_0)} \left[ \int p(\mathbf{x}_\sigma|\mathbf{x}_0) \log \frac{p(\mathbf{x}_\sigma|\mathbf{x}_0)}{p(\mathbf{x}_\sigma)} d\mathbf{x}_\sigma \right] \tag{182}$$

$$= \mathbb{E}_{\mathbf{x}_0 \sim p(\mathbf{x}_0)} \left[ D_{\mathrm{KL}}(p(\mathbf{x}_\sigma|\mathbf{x}_0) \,\|\, p(\mathbf{x}_\sigma)) \right] \tag{183}$$

### D.1    DESIGN FOR EDM

Given the simplification in Eqn. 183, the optimization problem in Eqn. 177 reduces to the following,

$$\min_{\sigma^2} \sigma^2 \quad \text{subject to} \quad \mathbb{E}_{\mathbf{x}_0 \sim p(\mathbf{x}_0)} \left[ D_{\mathrm{KL}}(p(\mathbf{x}_\sigma|\mathbf{x}_0) \,\|\, p(\mathbf{x}_\sigma)) \right] = 0 \tag{184}$$

Since at $\sigma_{\max}$ we expect the marginal distribution $p(\mathbf{x}_\sigma)$ to converge to the generative prior (i.e. completely destory the structure of the data), we approximate $p(\mathbf{x}_\sigma) \approx \mathcal{N}(0, \sigma^2 \mathbf{I}_d)$. With this simplification, the Lagrangian for the optimization problem in Eqn. 177 can be specified as,

$$\sigma_*^2 = \arg\min_{\sigma^2} \sigma^2 + \lambda \mathbb{E}_{\mathbf{x}_0 \sim p(\mathbf{x}_0)} \left[ D_{\mathrm{KL}}\big(\mathcal{N}(\mathbf{x}_\sigma; \mathbf{x}_0, \sigma^2) \,\|\, \mathcal{N}(\mathbf{x}_\sigma; 0, \sigma^2)\big) \right] \tag{185}$$

Setting the gradient w.r.t $\sigma^2 = 0$,

$$1 - \frac{\lambda}{\sigma^4} \mathbb{E}_{\mathbf{x}_0}[\mathbf{x}_0^\top \mathbf{x}_0] = 0 \tag{186}$$

which implies,

$$\sigma^2 = \sqrt{\lambda \mathbb{E}_{\mathbf{x}_0}[\mathbf{x}_0^\top \mathbf{x}_0]} \tag{187}$$

For an empirical dataset,

$$\sigma^2 = \sqrt{\frac{\lambda}{N} \sum_{i=1}^{N} \|\mathbf{x}_i\|_2^2} \tag{188}$$

This allows us to choose a $\sigma_{\max}$ from the underlying data statistics during training or sampling. It is worth noting that the choice of the multiplier $\lambda$ impacts the magnitude of $\sigma_{\max}$. However, this parameter can be chosen in a principled manner. At $\sigma_{\max}^2 = \sigma_*^2$, the estimate of the mutual information is given by:

$$I_*(\mathbf{x}_0, \mathbf{x}_\sigma) = \frac{1}{\sigma_*^2} \mathbb{E}_{\mathbf{x}_0}[\mathbf{x}_0^\top \mathbf{x}_0] = \sqrt{\frac{\mathbb{E}_{\mathbf{x}_0}[\mathbf{x}_0^\top \mathbf{x}_0]}{\lambda}} \tag{189}$$

which implies,

$$\lambda = \frac{\mathbb{E}_{\mathbf{x}_0}[\mathbf{x}_0^\top \mathbf{x}_0]}{I_*(\mathbf{x}_0, \mathbf{x}_\sigma)^2} \tag{190}$$

The above result provides a way to choose $\lambda$. Given the dataset stastistics, $\mathbb{E}_{\mathbf{x}_0}[\mathbf{x}_0^\top \mathbf{x}_0]$, the user can specify an acceptable level of mutual information $I(\mathbf{x}_0, \mathbf{x}_\sigma)$ to compute the correponding $\lambda$, which can then be used to find the corresponding minimum $\sigma_{\max}$ required to achieve that level of mutual information. Next, we extend this analysis to t-EDM.

## D.2    EXTENSION TO T-EDM

In the case of t-EDM, we pose the optimization problem as follows,

$$\sigma^* = \arg\min_{\sigma^2} \sigma^2 + \lambda \mathbb{E}_{\mathbf{x}_0 \sim p(\mathbf{x}_0)} \Big[ D_\gamma \big( t_d(\mathbf{x}_0, \sigma^2 \boldsymbol{I}_d, \nu) \,\|\, t_d(0, \sigma^2 \boldsymbol{I}_d, \nu) \big) \Big] \tag{191}$$

where $D_\gamma(q \,\|\, p)$ is the Gamma-power Divergence between two distributions q and p. From the definition of the $\gamma$-Power Divergence in Eqn. 8, we have,

$$D_\gamma \big( t_d(\mathbf{x}_0, \sigma^2 \boldsymbol{I}_d, \nu) \,\|\, t_d(0, \sigma^2 \boldsymbol{I}_d, \nu) \big) = \underbrace{-\frac{1}{\nu\gamma} C_{\nu,d}^{\frac{\gamma}{1+\gamma}} \big(1 + \frac{d}{\nu-2}\big)^{-\frac{\gamma}{1+\gamma}} (\sigma^2)^{-\frac{\gamma d}{2(1+\gamma)} - 1}}_{=f(\nu,d)} \mathbb{E}_{\mathbf{x}_0}[\mathbf{x}_0^\top \mathbf{x}_0] \tag{192}$$

where $C_{\nu,d} = \frac{\Gamma(\frac{\nu+d}{2})}{\Gamma(\frac{\nu}{2})(\nu\pi)^{\frac{d}{2}}}$ and $\gamma = -\frac{2}{\nu+d}$ Solving the optimization problem yields the following optimal $\sigma$,

$$\sigma_*^2 = \Big[ \lambda f(\nu, d) \Big( \frac{\nu-2}{\nu-2+d} \Big) \mathbb{E}[\mathbf{x}_0^\top \mathbf{x}_0] \Big]^{\frac{\nu-2+d}{2(\nu-2)+d}} \tag{193}$$

For an empirical dataset, we have the following simplification,

$$\sigma_*^2 = \Big[ \lambda f(\nu, d) \Big( \frac{\nu-2}{\nu-2+d} \Big) \frac{1}{N} \sum_i \|\mathbf{x}_i\|_2^2 \Big]^{\frac{\nu-2+d}{2(\nu-2)+d}} \tag{194}$$

## D.3    EXTENSION TO CORRELATED GAUSSIAN NOISE

We now extend our formulation for optimal noise schedule design to the case of correlated noise in the diffusion perturbation kernel. This is useful, especially for scientific applications where the data energy is distributed quite non-uniformly across the (data) spectrum. Let $\mathbf{R} = \mathbb{E}_{\mathbf{x_0} \sim p(\mathbf{x_0})}[\mathbf{x_0}\mathbf{x_0}^\top] \in \mathbb{R}^{d \times d}$ denote the data correlation matrix. Let also consider the perturbation kernel $\mathcal{N}(0, \boldsymbol{\Sigma})$ for the postitve-defintie covaraince matrix $\boldsymbol{\Sigma} \in \mathbb{R}^{d \times d}$. Following the steps in equation 185, the Lagrangian for noise covariance estimation can be formulated as follows:

$$\min_{\boldsymbol{\Sigma}} \operatorname{trace}(\boldsymbol{\Sigma}) + \lambda \mathbb{E}_{\mathbf{x}_0 \sim p(\mathbf{x}_0)} \Big[ D_{\mathrm{KL}}(\mathcal{N}(\mathbf{x}_0, \boldsymbol{\Sigma}) \,\|\, \mathcal{N}(0, \boldsymbol{\Sigma})) \Big] \tag{195}$$

$$\min_{\boldsymbol{\Sigma}} \operatorname{trace}(\boldsymbol{\Sigma}) + \lambda \mathbb{E}_{\mathbf{x}_0 \sim p(\mathbf{x}_0)} \Big[ \mathbf{x}_0^\top \boldsymbol{\Sigma}^{-1} \mathbf{x}_0 \Big] \tag{196}$$

$$\min_{\boldsymbol{\Sigma}} \operatorname{trace}(\boldsymbol{\Sigma}) + \lambda \mathbb{E}_{\mathbf{x}_0 \sim p(\mathbf{x}_0)} \Big[ \operatorname{trace}(\boldsymbol{\Sigma}^{-1} \mathbf{x}_0 \mathbf{x}_0^\top) \Big] \tag{197}$$

$$\min_{\boldsymbol{\Sigma}} \operatorname{trace}(\boldsymbol{\Sigma}) + \lambda \operatorname{trace}(\boldsymbol{\Sigma}^{-1} \mathbb{E}_{\mathbf{x}_0 \sim p(\mathbf{x}_0)}[\mathbf{x}_0 \mathbf{x}_0^\top]) \tag{198}$$

$$\min_{\boldsymbol{\Sigma}} \operatorname{trace}(\boldsymbol{\Sigma}) + \lambda \operatorname{trace}(\boldsymbol{\Sigma}^{-1} \mathbf{R}) \tag{199}$$

It can be shown that the optimal solution to this minimization problem is given by, $\boldsymbol{\Sigma}^* = \sqrt{\lambda} \mathbf{R}^{1/2}$, where $\mathbf{R}^{1/2}$ denotes the matrix square root of $\mathbf{R}$. This implies that the noise energy must be distributed along the singular vectors of the correlaiton matrix, where the energy is proportional to the noise singular values. We include the proof below.

*Proof.* We define the objective:

$$f(\boldsymbol{\Sigma}) = \operatorname{trace}(\boldsymbol{\Sigma}) + \lambda \operatorname{trace}(\boldsymbol{\Sigma}^{-1} \mathbf{R}). \tag{200}$$

We compute the gradient of $f(\boldsymbol{\Sigma})$ with respect to $\boldsymbol{\Sigma}$:

$$\nabla_{\boldsymbol{\Sigma}} f(\boldsymbol{\Sigma}) = \frac{\partial}{\partial \boldsymbol{\Sigma}} \text{trace}(\boldsymbol{\Sigma}) + \lambda \frac{\partial}{\partial \boldsymbol{\Sigma}} \text{trace}(\boldsymbol{\Sigma}^{-1} \mathbf{R}). \tag{201}$$

The gradient of the first term is straightforward:

$$\frac{\partial}{\partial \boldsymbol{\Sigma}} \text{trace}(\boldsymbol{\Sigma}) = \mathbf{I}. \tag{202}$$

For the second term, using the matrix calculus identity, the gradient is:

$$\frac{\partial}{\partial \boldsymbol{\Sigma}} \left( \lambda \, \text{trace}(\boldsymbol{\Sigma}^{-1} \mathbf{R}) \right) = -\lambda \boldsymbol{\Sigma}^{-1} \mathbf{R} \boldsymbol{\Sigma}^{-1}. \tag{203}$$

Combining these results, the total gradient is:

$$\nabla_{\boldsymbol{\Sigma}} f(\boldsymbol{\Sigma}) = \mathbf{I} - \lambda \boldsymbol{\Sigma}^{-1} \mathbf{R} \boldsymbol{\Sigma}^{-1}. \tag{204}$$

Setting the gradient to zero to find the critical point:

$$\mathbf{I} - \lambda \boldsymbol{\Sigma}^{-1} \mathbf{R} \boldsymbol{\Sigma}^{-1} = 0. \tag{205}$$

$$\boldsymbol{\Sigma}^{-1} \mathbf{R} \boldsymbol{\Sigma}^{-1} = \frac{1}{\lambda} \mathbf{I}. \tag{206}$$

which implies,

$$\mathbf{R} = \lambda \boldsymbol{\Sigma}^2. \tag{207}$$

$$\boldsymbol{\Sigma} = \sqrt{\lambda} \mathbf{R}^{1/2}. \tag{208}$$

which completes the proof.

## E  LOG-LIKELIHOOD FOR T-EDM

Here, we present a method to estimate the log-likelihood for the generated samples using the ODE solver for t-EDM (see Section 3.5). Our analysis is based on the likelihood computation in continuous-time diffusion models as discussed in Song et al. (2020) (Appendix D.2). More specifically, given a probability-flow ODE,

$$\frac{d\mathbf{x}_t}{dt} = \boldsymbol{f}_{\boldsymbol{\theta}}(\mathbf{x}_t, t), \tag{209}$$

with a vector field $\boldsymbol{f}_{\boldsymbol{\theta}}(\mathbf{x}_t, t)$, Song et al. (2020) propose to estimate the log-likelihood of the model as follows,

$$\log p(\mathbf{x}_0) = \log p(\mathbf{x}_T) + \int_0^T \nabla \cdot \boldsymbol{f}_{\boldsymbol{\theta}}(\mathbf{x}_t, t) dt. \tag{210}$$

The divergence of the vector field $\boldsymbol{f}_{\boldsymbol{\theta}}(\mathbf{x}_t, t)$ is further estimated using the Skilling-Hutchinson trace estimator (Skilling, 1989; Hutchinson, 1990) as follows,

$$\nabla \cdot \boldsymbol{f}_{\boldsymbol{\theta}}(\mathbf{x}_t, t) = \mathbb{E}_{p(\boldsymbol{\epsilon})}[\boldsymbol{\epsilon}^\top \nabla \boldsymbol{f}_{\boldsymbol{\theta}}(\mathbf{x}_t, t) \boldsymbol{\epsilon}] \tag{211}$$

where usually, $\epsilon \sim \mathcal{N}(\mathbf{0}, \boldsymbol{I}_d)$. For t-EDM ODE, this estimate can be further simplified as follows,

$$\nabla \cdot \boldsymbol{f_\theta}(\mathbf{x}_t, t) = \mathbb{E}_{p(\epsilon)}[\boldsymbol{\epsilon}^\top \nabla \boldsymbol{f_\theta}(\mathbf{x}_t, t)\boldsymbol{\epsilon}] \tag{212}$$

$$= \mathbb{E}_{p(\epsilon)}\Big[\boldsymbol{\epsilon}^\top \nabla \Big(\frac{\mathbf{x}_t - \boldsymbol{D_\theta}(\mathbf{x}_t, t)}{t}\Big)\boldsymbol{\epsilon}\Big] \tag{213}$$

$$= \mathbb{E}_{p(\epsilon)}\Big[\boldsymbol{\epsilon}^\top \Big(\frac{\boldsymbol{I}_d - \nabla_{\mathbf{x}_t} \boldsymbol{D_\theta}(\mathbf{x}_t, t)}{t}\Big)\boldsymbol{\epsilon}\Big] \tag{214}$$

$$= \frac{1}{t}\mathbb{E}_{p(\epsilon)}\Big[\boldsymbol{\epsilon}^\top \boldsymbol{\epsilon} - \boldsymbol{\epsilon}^\top \nabla_{\mathbf{x}_t} \boldsymbol{D_\theta}(\mathbf{x}_t, t)\boldsymbol{\epsilon}\Big] \tag{215}$$

$$= \frac{1}{t}\mathbb{E}_{p(\epsilon)}\Big[\boldsymbol{\epsilon}^\top \boldsymbol{\epsilon} - \boldsymbol{\epsilon}^\top \nabla_{\mathbf{x}_t} \boldsymbol{D_\theta}(\mathbf{x}_t, t)\boldsymbol{\epsilon}\Big] \tag{216}$$

$$= \frac{1}{t}\Big[d - \mathbb{E}_{p(\epsilon)}\big(\boldsymbol{\epsilon}^\top \nabla_{\mathbf{x}_t} \boldsymbol{D_\theta}(\mathbf{x}_t, t)\boldsymbol{\epsilon}\big)\Big] \tag{217}$$

where $d$ is the data dimensionality. Thus, the log-likelihood can be specified as,

$$\log p(\mathbf{x}_0) = \log p(\mathbf{x}_T) + \int_0^T \frac{1}{t}\Big[d - \mathbb{E}_{p(\epsilon)}\big(\boldsymbol{\epsilon}^\top \nabla \boldsymbol{D_\theta}(\mathbf{x}_t, t)\boldsymbol{\epsilon}\big)\Big]dt. \tag{218}$$

When $\epsilon \sim \mathcal{N}(0, \sigma^2 \boldsymbol{I}_d)$, the above result can be re-formulated as,

$$\log p(\mathbf{x}_0) = \log p(\mathbf{x}_T) + \int_0^T \frac{1}{t}\Big[d - \frac{1}{\sigma^2}\mathbb{E}_{p(\epsilon)}\big(\boldsymbol{\epsilon}^\top \nabla \boldsymbol{D_\theta}(\mathbf{x}_t, t)\boldsymbol{\epsilon}\big)\Big]dt. \tag{219}$$

Moreover, using the first-order taylor series expansion

$$D_\theta(\mathbf{x} + \epsilon) = D_\theta(\mathbf{x}) + \nabla D_\theta \epsilon + \mathcal{O}(\|\epsilon\|^2) \tag{220}$$

For a sufficiently small $\sigma$, higher-order terms in $\mathcal{O}(\|\epsilon\|^2)$ can be ignored since $\mathbb{E}[\|\epsilon^2\|] = \sigma^2 d$. Therefore,

$$D_\theta(\mathbf{x} + \epsilon) \approx D_\theta(\mathbf{x}) + \nabla D_\theta \epsilon \tag{221}$$

$$\epsilon^\top \nabla D_\theta \epsilon \approx \epsilon^\top [D_\theta(\mathbf{x} + \epsilon) - D_\theta(\mathbf{x})] \tag{222}$$

$$\mathbb{E}_\epsilon[\epsilon^\top \nabla D_\theta \epsilon] \approx \mathbb{E}_\epsilon[\epsilon^\top (D_\theta(\mathbf{x} + \epsilon) - D_\theta(\mathbf{x}))] \tag{223}$$

$$\mathbb{E}_\epsilon[\epsilon^\top \nabla D_\theta \epsilon] \approx \mathbb{E}_\epsilon[\epsilon^\top D_\theta(\mathbf{x} + \epsilon)] \tag{224}$$

Therefore, the log-likelihood expression can be further simplified as,

$$\log p(\mathbf{x}_0) = \log p(\mathbf{x}_T) + \int_0^T \frac{1}{t}\Big[d - \frac{1}{\sigma^2}\mathbb{E}_{p(\epsilon)}\big(\boldsymbol{\epsilon}^\top \nabla \boldsymbol{D_\theta}(\mathbf{x}_t, t)\boldsymbol{\epsilon}\big)\Big]dt \tag{225}$$

$$\log p(\mathbf{x}_0) = \log p(\mathbf{x}_T) + \int_0^T \frac{1}{t}\Big[d - \frac{1}{\sigma^2}\mathbb{E}_{p(\epsilon)}\big(\boldsymbol{\epsilon}^\top D_\theta(\mathbf{x}_t + \epsilon, t)\big)\Big]dt \tag{226}$$

The advantage of this simplification is that we don't need to rely on expensive jacobian-vector products in Eq. 218. However, since the denoiser now depends on $\epsilon$, monte-carlo approximation of the expectation in the above equation could be computationally expensive for many samples $\epsilon$

## F    DISCUSSION AND LIMITATIONS

### F.1    RELATED WORK

**Connections with Denoising Score Matching.** For the perturbation kernel $q(\mathbf{x}_t|\mathbf{x}_0) = t_d(\mu_t \mathbf{x}_0, \sigma_t^2 \boldsymbol{I}_d, \nu)$, the denoising score matching (Vincent, 2011; Song et al., 2020) loss, $\mathcal{L}_{\text{DSM}}$, can be formulated as,

$$\mathcal{L}_{\text{DSM}}(\theta) \propto \mathbb{E}_{\mathbf{x}_0 \sim p(\mathbf{x}_0)}\mathbb{E}_t \mathbb{E}_{\epsilon \sim \mathcal{N}(0, \boldsymbol{I}_d)}\mathbb{E}_{\kappa \sim \chi^2(\nu)/\nu}\Big[\lambda(\mathbf{x}_t, \nu, t)\Big\|\boldsymbol{D_\theta}\big(\mu_t \mathbf{x}_0 + \sigma_t \frac{\epsilon}{\sqrt{\kappa}}, \sigma_t\big) - \mathbf{x}_0\Big\|_2^2\Big] \tag{227}$$

with the scaling factor $\lambda(\mathbf{x}_t, \nu, t) = [(\nu + d)/(\nu + d_1)]^2$ where $d_1 = (1/\sigma_t^2)\|\mathbf{x}_t - \mu_t \mathbf{x}_0\|_2^2$ (proof in App. A.9). Therefore, the denoising score matching loss in our framework is equivalent to the simplified training objective in Eq. 10 scaled by a data-dependent coefficient. However, in this work, we do not explore this loss formulation and leave further exploration to future work.

**Prior work in Heavy-Tailed Generative Modeling.** The idea of exploring heavy-tailed priors for modeling heavy-tailed distributions has been explored in several works in the past. More specifically, Jaini et al. (2020) argue that a Lipschitz flow map cannot change the tails of the base distribution significantly. Consequently, they use a heavy-tailed prior (modeled using a Student-t distribution) as the base distribution to learn Tail Adaptive flows (TAFs), which can model the tails more accurately. In this work, we make similar observations where standard diffusion models fail to accurately model the tails of real-world distributions. Consequently, Laszkiewicz et al. (2022) assess the *tailedness* of each marginal dimension and set the prior accordingly. On a similar note, we note that learning the tail parameter $\nu$ spatially and across channels can provide greater modeling flexibility for downstream tasks and will be an important direction for future work on this problem. More recently, Kim et al. (2024) introduce heavy-tailed VAEs (Kingma & Welling, 2022; Rezende & Mohamed, 2016) based on minimizing $\gamma$-power divergences (Eguchi, 2021). This is perhaps the closest connection of our method with prior work since we rely on $\gamma$-power divergences to minimize the divergence between heavy-tailed forward and reverse diffusion posteriors. However, VAEs often have scalability issues and tend to produce blurry artifacts (Dosovitskiy & Brox, 2016; Pandey et al., 2022). On the other hand, we work with diffusion models, which are known to scale well to large-scale modeling applications (Pathak et al., 2024; Mardani et al., 2024; Esser et al., 2024; Podell et al., 2023).

Within diffusion models, Deasy et al. (2021) show that denoising score matching can be extended to Generalized Normal distributions and refer to the resulting training as heavy-tailed denoising score matching (HTDSM). Moreover, Deasy et al. (2021) rely on annealed Langevin dynamics for sampling. However, an exact formulation of the reverse process is non-trivial and would involve Levy formulations of Kolmogorov's forward and reverse equations, which can be quite complex. In contrast, due to the conditional properties of multivariate Student-t distributions, our method circumvents such complex formulations and enables a simple framework for designing samplers for Student-t based diffusion models (see Proposition 2 in the main text). Similarly, Yoon et al. (2023) present a framework for modeling heavy-tailed distributions using $\alpha$-stable Levy processes while Shariatian et al. (2024) simplify the framework proposed in Yoon et al. (2023) and instantiate it for more practical diffusion models like DDPM. In contrast, our work deals with Student-t noise, which in general (with the exceptions of Cauchy and the Gaussian distribution) is not $\alpha$-stable and, therefore, a distinct category of diffusion models for modeling heavy-tailed distributions. Moreover, prior works like Yoon et al. (2023); Shariatian et al. (2024) rely on empirical evidence from light-tailed variants of small-scale datasets like CIFAR-10 (Krizhevsky, 2009) and their efficacy on actual large-scale scientific datasets like weather datasets remains to be seen.

**Prior work in Diffusion Models.** Our work is a direct extension of standard diffusion models in the literature (Karras et al., 2022; Ho et al., 2020; Song et al., 2020). Moreover, since it only requires a few lines of code change to transition from standard diffusion models to our framework, our work is directly compatible with popular families of latent diffusion models (Pandey et al., 2022; Rombach et al., 2022) and augmented diffusion models (Dockhorn et al., 2022; Pandey & Mandt, 2023; Singhal et al., 2023). Our work is also related to prior work in diffusion models on a more theoretical level. More specifically, PFGM++ (Xu et al., 2023b) is a unique type of generative flow model inspired by electrostatic theory. It treats d-dimensional data as electrical charges in a $D + d$-dimensional space, where the electric field lines define a bijection between a heavy-tailed prior and the data distribution. $D$ is a hyperparameter controlling the shape of the electric fields that define the generative mapping. In essence, their method can be seen as utilizing a perturbation kernel:

$$p(\mathbf{x}_t|\mathbf{x}_0) \propto (\|\mathbf{x}_t - \mathbf{x}_0\|_2^2 + \sigma_t^2 D))^{-\frac{D+d}{2}} = t_d(\mathbf{x}_0, \sigma_t^2 \mathbf{I}_d, D)$$

When setting $\nu = D$, the perturbation kernel becomes equivalent to that of t-EDM, indicating the Student-t perturbation kernel can be interpreted from another physical perspective — that of electrostatic fields and

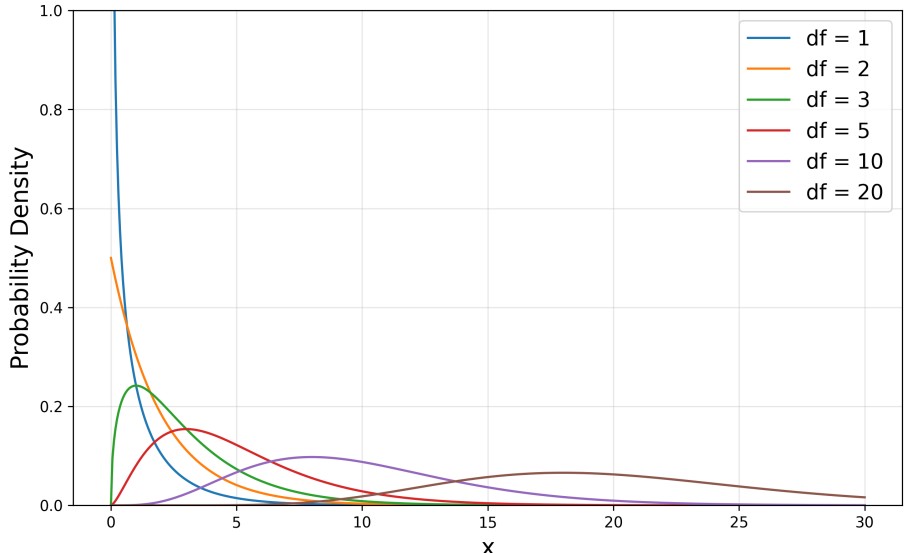

Figure 8: Illustration of the pdf for a $\chi^2$ distribution with varying degrees of freedom.

charges. The authors demonstrated that using an intermediate value for $D$ (or $\nu$) leads to improved robustness compared to diffusion models (where $D \to \infty$), due to the heavy-tailed perturbation kernel.

### F.2    LIMITATIONS AND FUTURE WORK

While our proposed framework works well for modeling heavy-tailed data, it is not without its limitations.

**On tuning $\nu$.** Firstly, while the parameter $\nu$ offers controllability for tail estimation using diffusion models, it also increases the tuning budget by introducing an extra hyperparameter. Moreover, for diverse data channels, tuning $\nu$ per channel could be key to good estimation at the tails. This could result in a combinatorial explosion with manual tuning. We think there might be several ways to learn $\nu$ from the data.

- Firstly, it is worth noting that the parameter $\nu$ directly influences training in heavy-tailed diffusions. This is because the noising process $\mathbf{x}_t = \mu_t \mathbf{x}_0 + \frac{\mathbf{z}}{\sqrt{\kappa}}$ where $\kappa \sim \chi^2(\nu)/\nu$. Since the $\chi^2$-squared distribution is a special case of a Gamma distribution, we can perhaps use reparameterization (Ruiz et al., 2016) to estimate the parameters (i.e. $\nu$) of this distribution using a neural network to learn this parameter end-to-end. One important caveat in learning $\nu$ end-to-end is that the optimization process could just select a divergence which reduces the loss while not fitting $\nu$ on the data. Therefore, exploring these caveats for learning $\nu$ are an important direction for further research.

- Alternatively, one can also estimate $\nu$ from the data in a separate stage and use it as a perturbation kernel parameter. We highlight one simple way to do this in Appendix D where we attempt to derive the optimal noise schedule for EDM and t-EDM by minimizing the mutual information between the $\mathbf{x}_0$ and the noisy data point $\mathbf{x}_\sigma$ w.r.t $\sigma$. In principle, for t-EDM, we can also solve this minimization problem jointly over $\nu$ and $\sigma$ to learn an initial $\nu$ from the data.

**On Evaluation metrics.** Secondly, our evaluation protocol relies primarily on comparing the statistical properties of samples obtained by flattening the generated or train/test set samples. One disadvantage of this approach is that our current evaluation metrics ignore the structure of the generated samples. On this

note, it may be tempting to use metrics like FID (Heusel et al., 2018), precision/recall (Sajjadi et al., 2018) to assess the performance between the generated and dataset samples. However, a key caveat in using these metrics is the reliance on a pre-trained feature extractor, which is commonly trained on natural images (like ImageNet). However, in this work, we focus on weather data, which has a much different structure than natural images. Due to this distribution shift, it is unclear if a feature extractor pre-trained on natural images can extract meaningful structural features from weather datasets. Therefore, computing metrics like FID using a pre-trained network trained on natural images for weather modeling is dubious at best. In this spirit, training these feature extractors on large-scale weather data using classification or self-supervised losses so that downstream metrics like FID can be reported reliably can be an interesting direction for future work.

**Applications in other domains.** In this work, while we explore the application of heavy-tailed diffusion models like t-EDM in the context of weather forecasting, the methods developed in this work are quite generic, and it will be interesting to apply these in the context of other domains like finance, which rely on heavy-tailed modeling of assets. Lastly, our conditional synthesis experiments are limited to the next step prediction and do not perform autoregressive rollouts like in the forecasting setup. Extending t-EDM with improved preconditioning and automatic $\nu$ tuning for forecasting would be an interesting research direction.

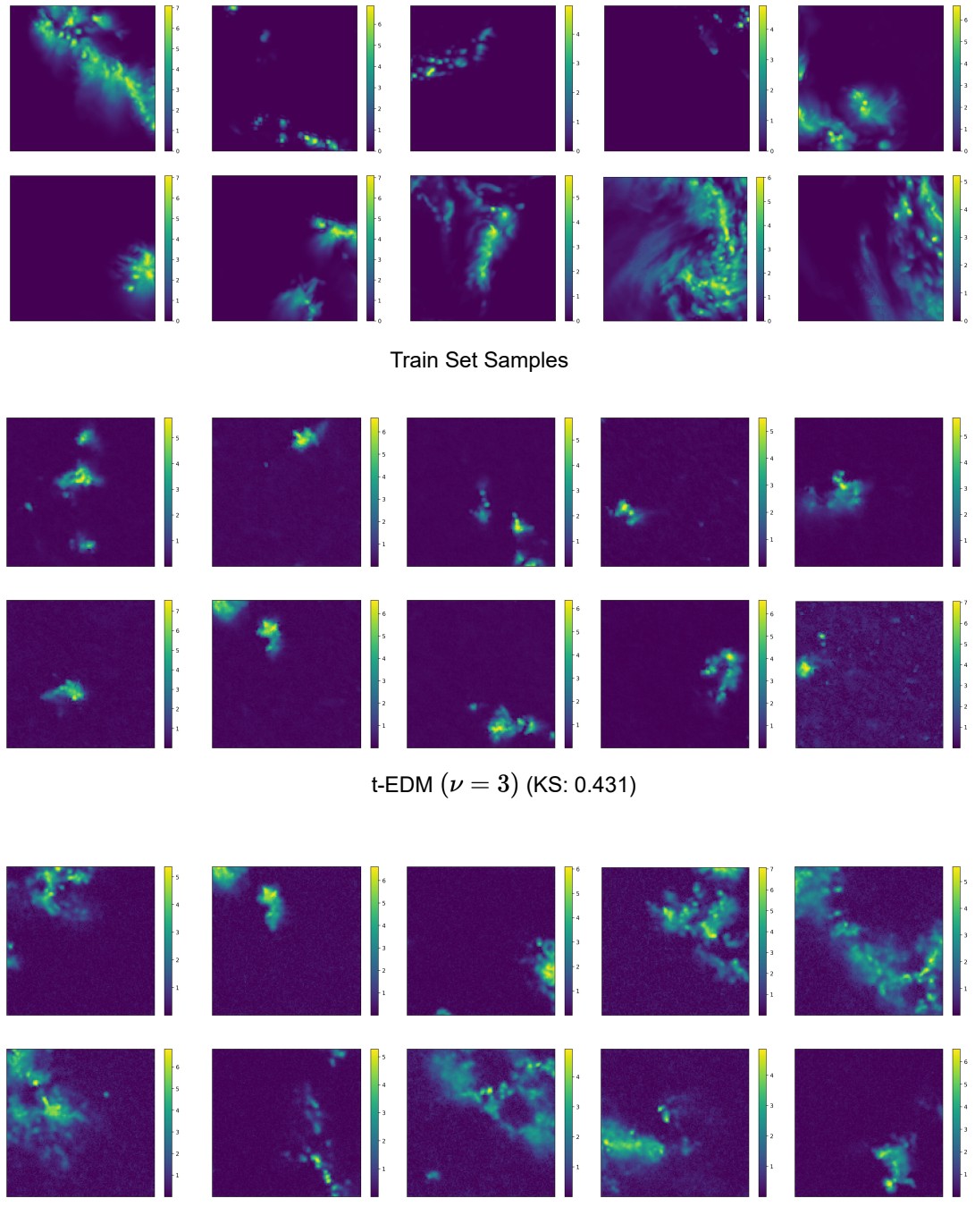

Figure 9: Random samples generated from t-EDM (Top Panel) and EDM (Bottom Panel) for the Vertically Integrated Liquid (VIL) channel. KS: Kolmogorov-Smirnov 2-sample statistic. Samples have been scaled logarithmically for better visualization

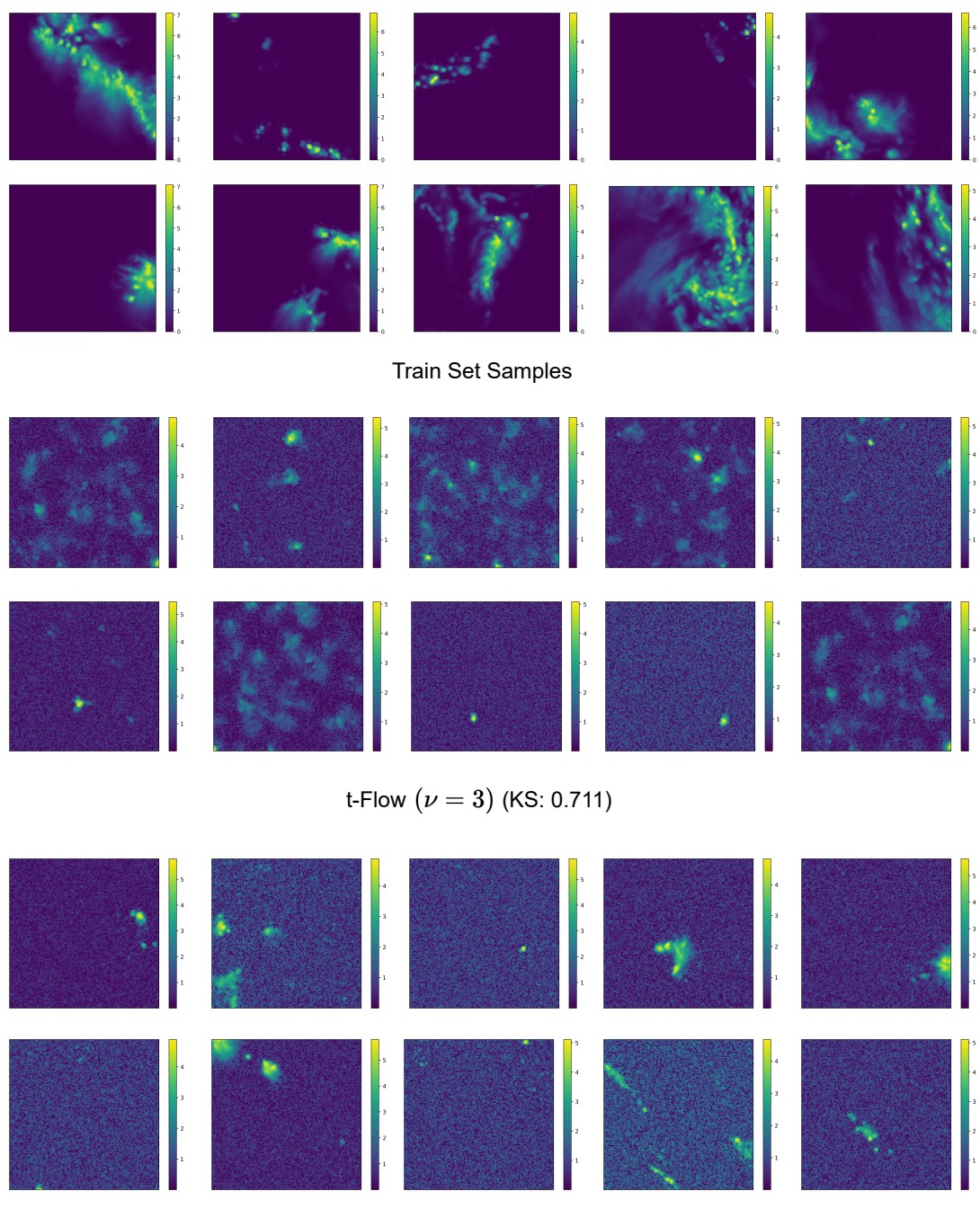

Figure 10: Random samples generated from t-Flow (Top Panel) and Gaussian Flow (Bottom Panel) for the Vertically Integrated Liquid (VIL) channel. KS: Kolmogorov-Smirnov 2-sample statistic. Samples have been scaled logarithmically for better visualization

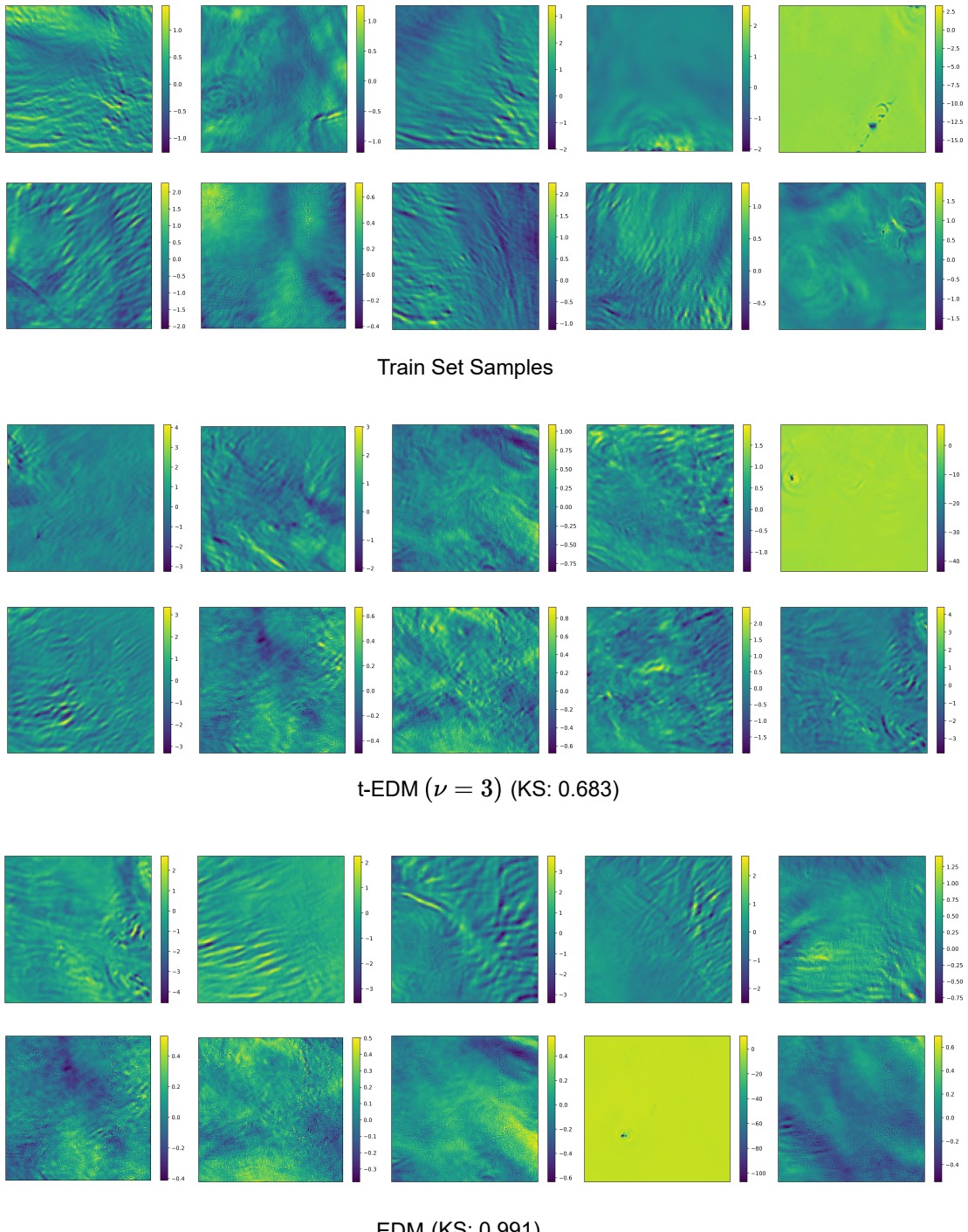

Figure 11: Random samples generated from t-EDM (Top Panel) and EDM (Bottom Panel) for the Vertical Wind Velocity (w20) channel. KS: Kolmogorov-Smirnov 2-sample statistic

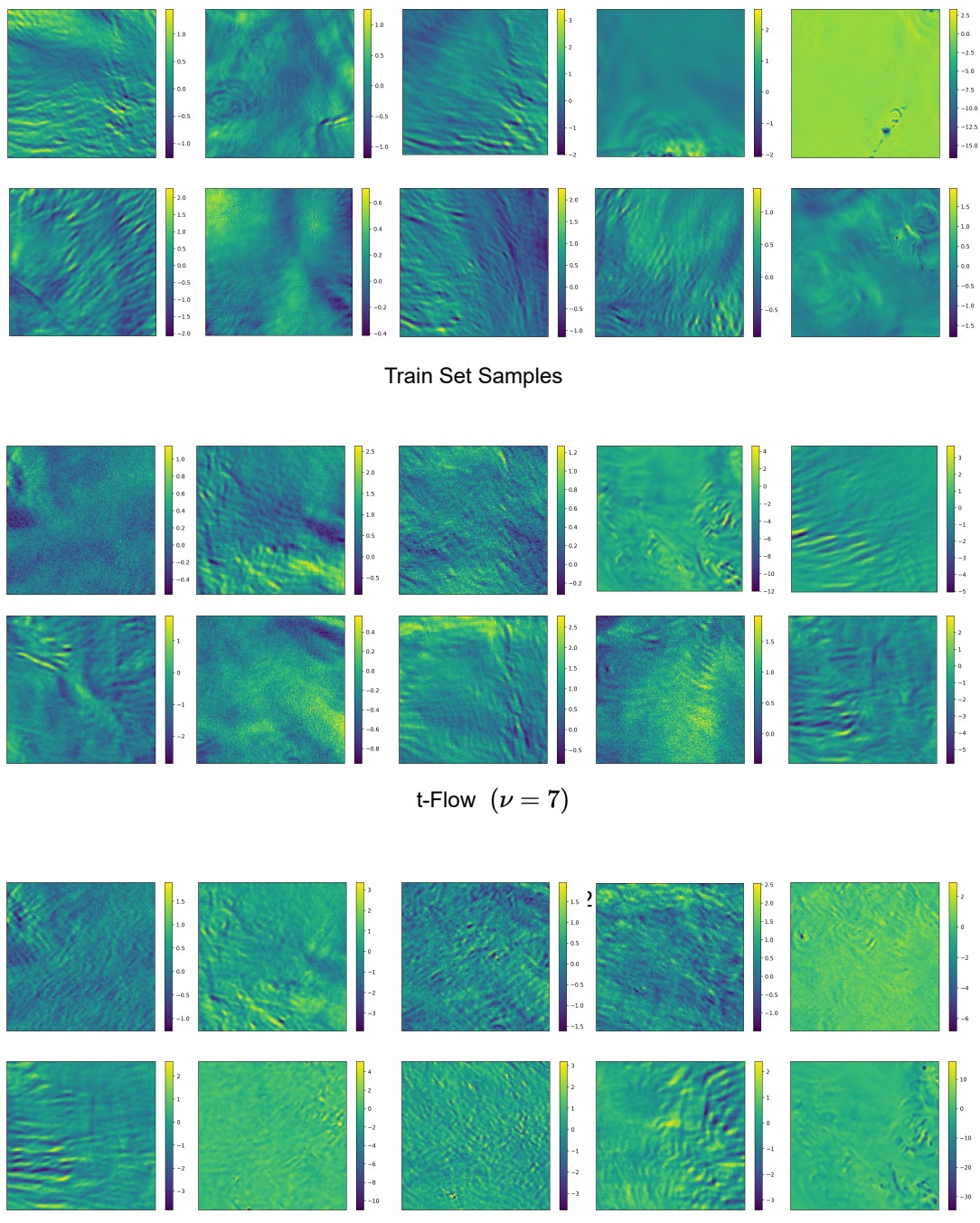

Figure 12: Random samples generated from t-Flow (Top Panel) and Gaussian Flow (Bottom Panel) for the Vertical Wind Velocity (w20) channel. KS: Kolmogorov-Smirnov 2-sample statistic

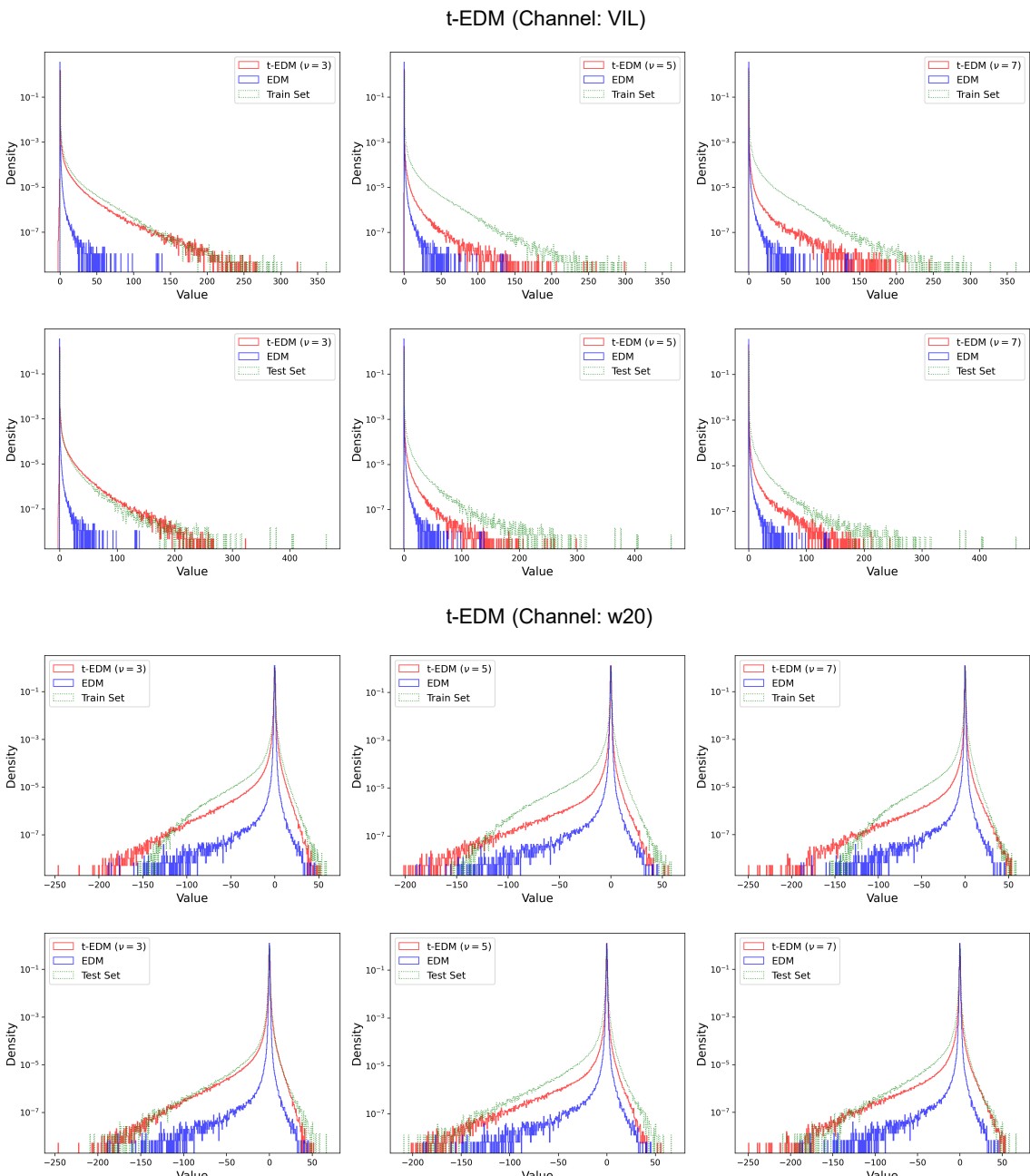

Figure 13: 1-d Histogram Comparisons between samples from the generated and the Train/Test set for the Vertically Integrated Liquid (VIL, see Top Pandel) and Vertical Wind Velocity (w20, see Bottom Panel) channels using t-EDM (with varying $\nu$).

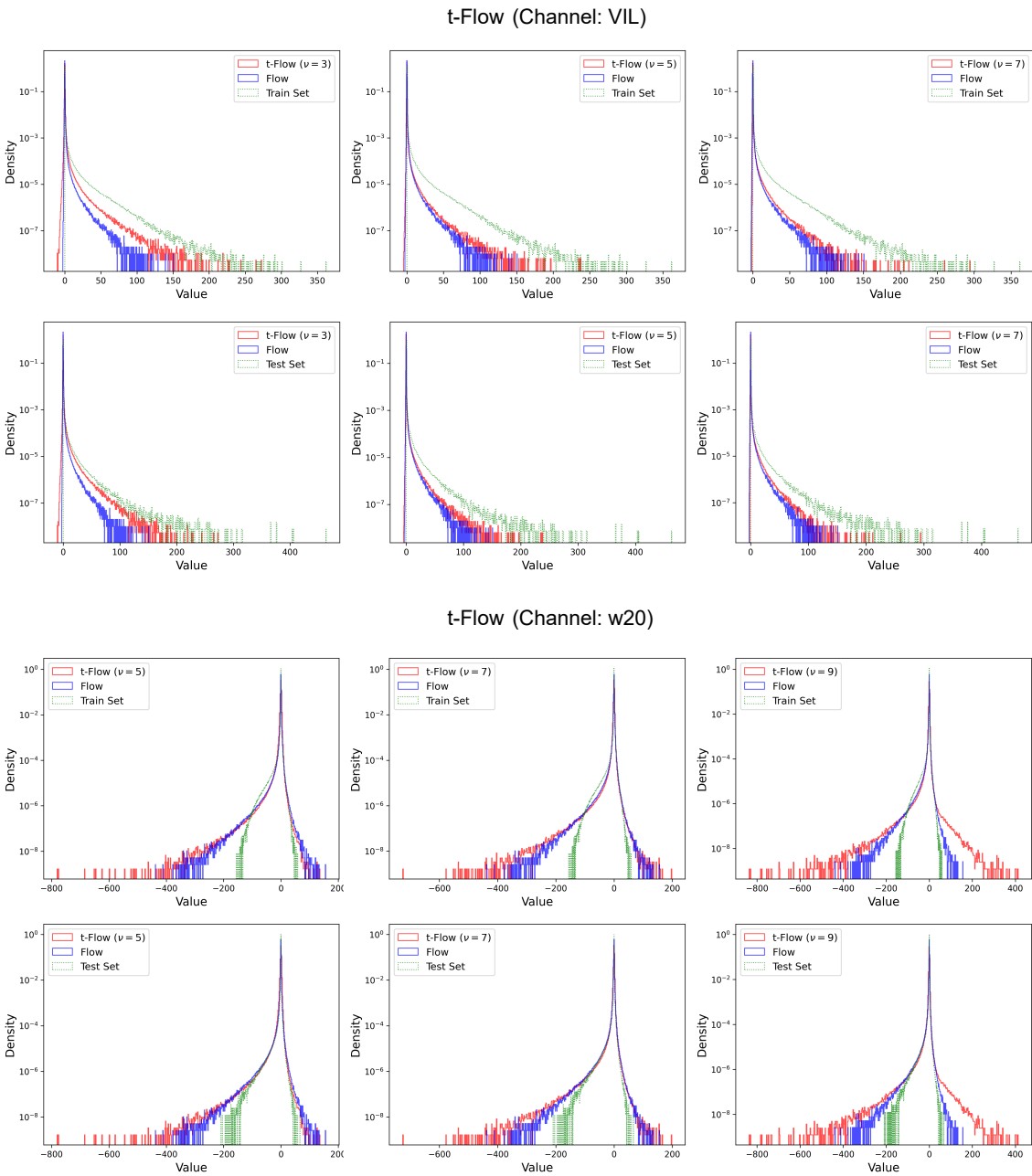

Figure 14: 1-d Histogram Comparisons between samples from the generated and the Train/Test set for the Vertically Integrated Liquid (VIL, see Top Pandel) and Vertical Wind Velocity (w20, see Bottom Panel) channels using t-Flow (with varying $\nu$).

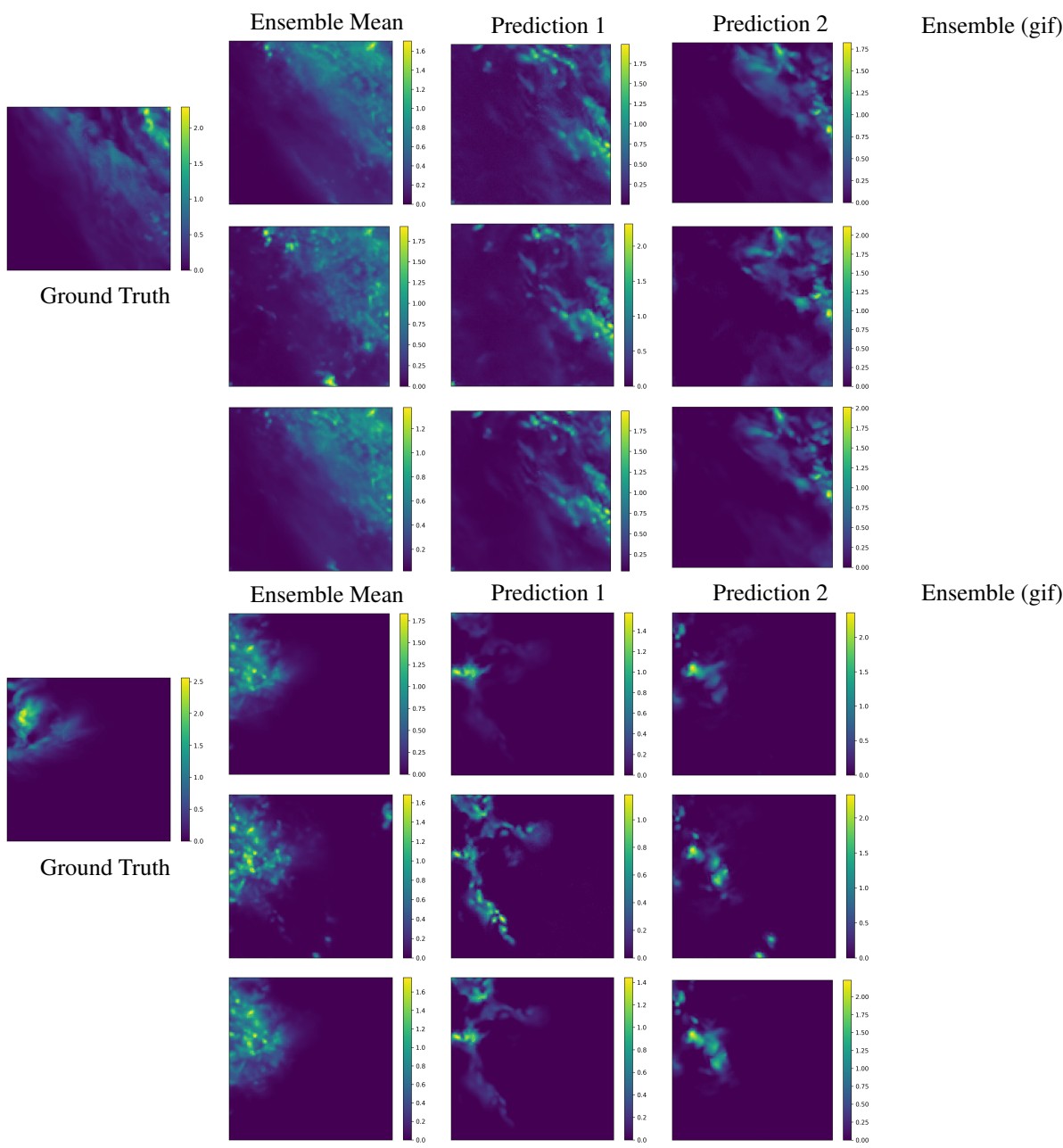

Figure 9: Qualitative visualization of samples generated from our conditional modeling for predicting the next state for the Vertically Integrated Liquid (VIL) channel. The ensemble mean represents the mean of ensemble predictions (16 in our case). Columns 2-3 represent two samples from the ensemble. The last column visualizes an animation of all ensemble members (Best viewed in a dedicated PDF reader). For each sample, the rows correspond to predictions from EDM, t-EDM ($\nu = 3$), and t-EDM ($\nu = 5$) from top to bottom, respectively. Samples have been scaled logarithmically for better visualization

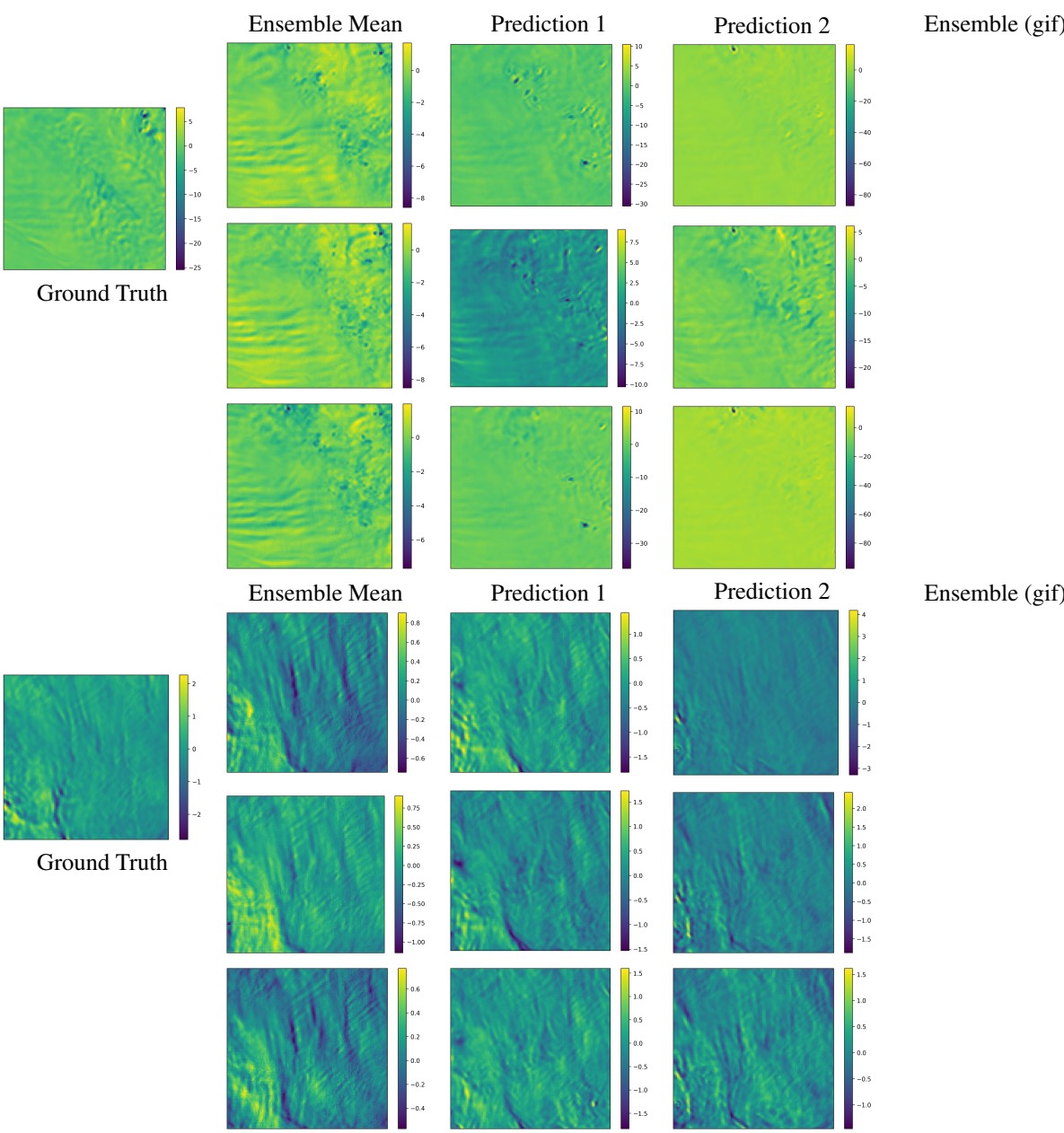

Figure 10: Qualitative visualization of samples generated from our conditional modeling for predicting the next state for the Vertical Wind Velocity (w20) channel. The ensemble mean represents the mean of ensemble predictions (16 in our case). Columns 2-3 represent two samples from the ensemble. The last column visualizes an animation of all ensemble members (Best viewed in a dedicated PDF reader). For each sample, the rows correspond to predictions from EDM, t-EDM ($\nu = 3$), and t-EDM ($\nu = 5$) from top to bottom, respectively.

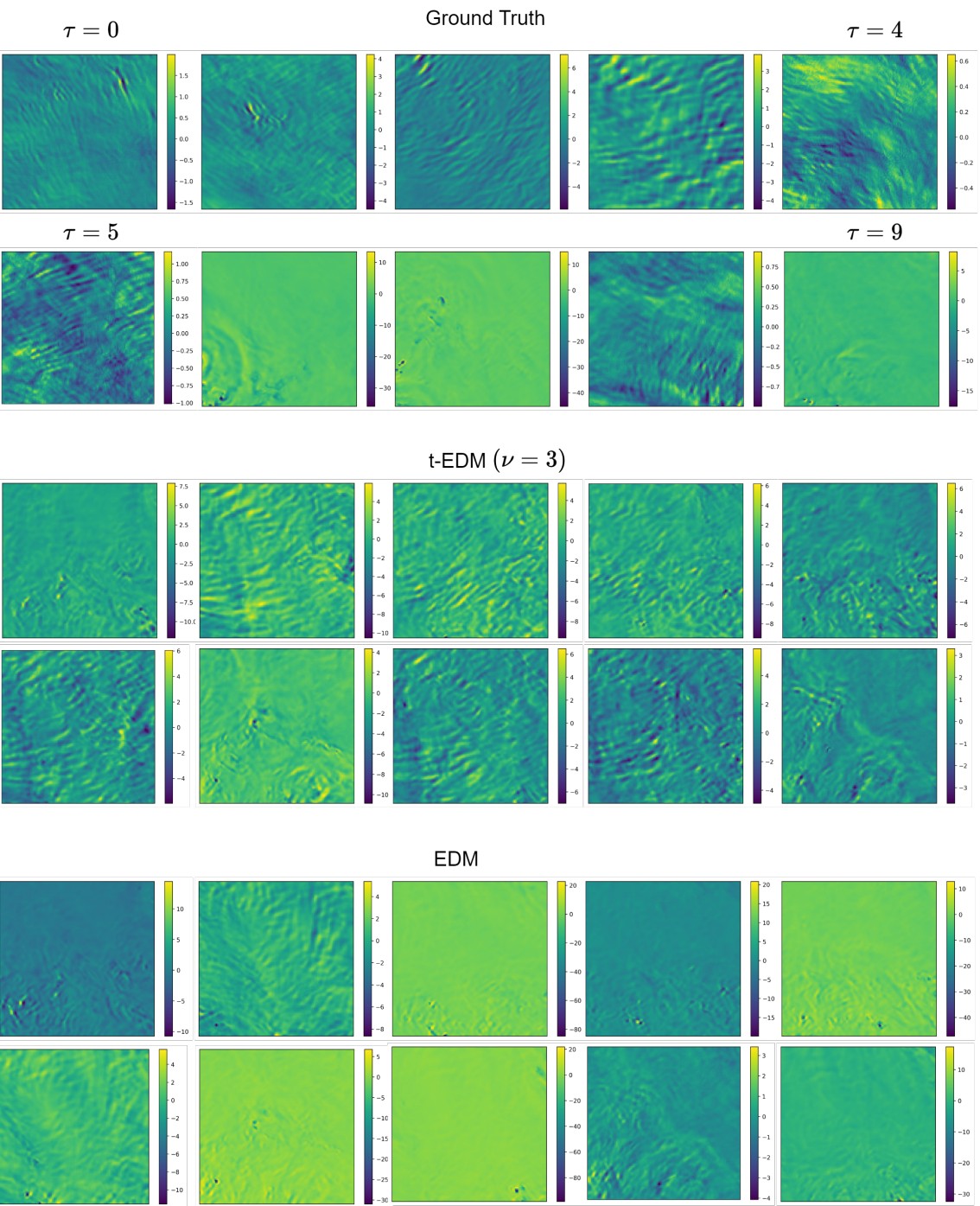

Figure 15: Qualitative illustration of autoregressive rollouts using t-EDM ($\nu = 3$) and EDM on the w20 channel. The conditional diffusion models were initialized using HRRR validation data and predictions were made with an interval of 1 hour up to a lead time of 10 hours. We do not observe any instabilities when generating trajectories using t-EDM. $\tau$ represents lead-time with $\tau = 0$ denoting initial state.

