# OpenReview forum: "Heavy-Tailed Diffusion Models"
_ICLR.cc/2025/Conference — ICLR 2025 Poster_

### Official Review · Reviewer_EPoh · 2024-10-27

**Soundness:** 4
**Presentation:** 4
**Contribution:** 3
**Rating:** 8
**Confidence:** 4

**Summary:**

This paper proposes a method for adapting denoising diffusion models to sample from heavy tailed distributions. The core idea is to replace the Gaussian prior and transition distributions from standard DDPM with Student-t distributions. The authors show how to construct a heavy-tailed diffusion model by defining the relevant noising process in terms of the Student-t distribution, calculating the corresponding reverse noising process, showing how it can be appropriately parameterized in terms of a prediction of $x_0$, and deriving a training objective defined in terms of the $\gamma$-Power Divergence by analogy to the standard KL objective used to train diffusion models. They then demonstrate the empirical success of their method on distributions from the HRRR dataset, showing that it outperforms Gaussian-based diffusion models in most settings they study.

**Strengths:**

My overall feeling is that this is a very technically solid paper which takes a relatively simple core idea - exchanging the Gaussian distributions in diffusion models for Student-t distributions in order to have a heavy-tailed sampling distribution - and executes on it methodically and very well. The presentation and flow of the argument is clear throughout, and all the obvious questions (parameterization of the reverse process, tractable training objective, continuous-time sampling with SDEs and ODEs, empirical performance) are dealt with comprehensively in my opinion.

The problem of sampling from heavy-tailed distributions is one of significant interest to the field, and I'm not aware of other work providing such a nice and integrated framework for heavy-tailed diffusion modeling, so I consider the contribution of an appropriate size for the conference.

Particular highlights were the connection of the diffusion modeling loss to the $\gamma$-Power Divergence, showing that this choice of divergence makes the loss tractable and reduces to a denoising $L^2$ loss similar to that of DDPM, and the discussion in Section 5 on various theoretical intuitions and insights the authors have concerning their framework.

**Weaknesses:**

I do not think this paper has any major weaknesses. The addition I would be most keen to see is a more thorough discussion of the results in Tables 2 and 3 (empirical results in the unconditional setting). For example, it would appear from Table that in the test setting for VIL, their method actually underperforms compared to EDM (+PCP) and compared to EDM (+INC) in the other setting, and the authors do not discuss this in the text as far as I can tell. (Though, Figure 3 tells a somewhat different story, and I can't work out how to reconcile the two - see question below.) The authors also don't really discuss the results in Table 3 in the main body, and a short commend about what we should take from those results might be appropriate.

Secondly, I also feel that many of the derivations in the work are somewhat routine once one has the core idea of using Student-t distributions to parameterize the forward process, and the novelty of the work is therefore not outstanding. Nevertheless, the problem is of sufficient interest and the execution of the authors is sufficiently good that I believe this still clearly merits acceptance to the conference.

**Questions:**

- How do we tune the parameter $\nu$ in practice? It looks from Table 3 like the choice matters a lot, so it might be nice to say something about how one should choose a suitable value of $\nu$ or know that one has found a good value.
- How confident are you that the 1-dimensional summary statistics you are looking in Table 2 (and maybe Table 3 also) at appropriate? From looking at Figure 3, t-EDM seems clearly qualitatively superior to the other methods. The fact that this is not reflected in the summary statistics in Table 2 makes me suspect the summary statistics are not capturing the most relevant aspects of the distributions.

---

> ### Author Response · Authors · 2024-11-23
> **Official Author Response - Reviewer EPoh**
>
> We thank the reviewer for their comments and address each point in more detail below.
>
> **Weaknesses/Questions**
>
> **Weakness 1a:** *The addition I would be most keen to see is a more thorough discussion of the results in Tables 2 and 3 (empirical results in the unconditional setting). For example, it would appear from Table that in the test setting for VIL, their method actually underperforms compared to EDM (+PCP) and compared to EDM (+INC) in the other setting, and the authors do not discuss this in the text as far as I can tell. (Though, Figure 3 tells a somewhat different story, and I can't work out how to reconcile the two - see question below.)*
>
> **Response:** We provide a detailed response for this point in our response to Question 2 below.
>
> **Weakness 1b:** *The authors also don't really discuss the results in Table 3 in the main body, and a short commend about what we should take from those results might be appropriate.*
>
> **Response:** We have added a short clarification on the results in Table 3 in the main text (lines 409-411).
>
> **Question 1:** *How do we tune the parameter \nu in practice? It looks from Table 3 like the choice matters a lot, so it might be nice to say something about how one should choose a suitable value of  \nu or know that one has found a good value.*
>
> **Response:** In this work, we tune $\nu$ manually for different datasets. However, we also highlight several principled ways to learn $\nu$ directly from the data.
>
> Firstly, it is worth noting that the parameter $\nu$ directly influences training in heavy-tailed diffusions. This is because the noising process $x_t = \mu_t x_0 + \frac{z}{\sqrt{\kappa}}$ where $\kappa \sim \chi^2(\nu)/nu$. Since the Chi-squared distribution is a special case of a Gamma distribution, we can perhaps use reparameterization [1] to estimate the parameters (i.e., $\nu$) of this distribution using a neural network and backprop end-to-end to learn this parameter. This would correspond to learning $\nu$ directly from the data.
>
> [1] The Generalized Reparameterization Gradient, Ruiz et al.
>
> Alternatively, one can also estimate $\nu$ from the data in a separate stage and use it as a perturbation kernel parameter. We highlight one simple way to do this in Appendix D, where we attempt to derive the optimal noise schedule for EDM and t-EDM by minimizing the mutual information between the $x_0$ and the noisy data point $x_\sigma$ w.r.t $\sigma$. In principle, for t-EDM, we can also solve this minimization problem jointly over $\nu$ and $\sigma$ to learn an initial $\nu$ from the data.
>
> We have added these points in Appendix F.2 in our revision. Overall, our proposed framework can be readily extended to learn $\nu$ in an end-to-end or separate stage fashion. Nevertheless, we leave this exploration to future work due to space constraints.
>
> **Question 2:** *How confident are you that the 1-dimensional summary statistics you are looking in Table 2 (and maybe Table 3 also) at appropriate? From looking at Figure 3, t-EDM seems clearly qualitatively superior to the other methods. The fact that this is not reflected in the summary statistics in Table 2 makes me suspect the summary statistics are not capturing the most relevant aspects of the distributions.*
>
> **Response:** This is an interesting observation. We elaborate more upon this aspect as follows.
>
> Firstly, our method exhibits better performance for the KS metric. This could be due to the fact that the KS metric, in addition to flattening the data, is computed over 99.9th and 0.1th percentiles for the right and left tails, respectively. This implies that the KS metric specifically examines the tail values and thus represents an appropriate metric to capture the difference between the tail estimation for the generated and the data samples. In contrast, the KR and SR metrics are based on computing higher-order moments over the entire domain of flattened empirical samples and thus might be more coarse in nature, hence the discrepancy.
>
> However, it is worth noting that while we present EDM baselines with improved preconditioning and preprocessing as additional baselines, these can also be applied to t-EDM, potentially improving results. Therefore, a more fair comparison should be between t-EDM and the baseline EDM for which t-EDM outperforms EDM by a significant margin in all metrics. We added this observation in Section 4.1.
>
> Secondly, due to the flattening operation, the metrics considered in this work disregard the spatial structure of the generated samples, which is undesirable. The qualitative histogram plots in Figure 3 help alleviate this to some extent. However, designing metrics that can quantify heavy-tailed behavior while considering the spatial structure of the generated samples is an interesting direction for further work but is outside the scope of this paper. We discuss these limitations in more detail in Appendix F.2.

---

### Official Review · Reviewer_e3KL · 2024-11-01

**Soundness:** 2
**Presentation:** 2
**Contribution:** 2
**Rating:** 3
**Confidence:** 4

**Summary:**

The paper proposes a novel generative model that, in a sense, mirrors traditional diffusion models (DM) by substituting Gaussian noise in the noising process with noise following a Student’s t-distribution.

More precisely, the authors introduce a sequence of joint conditional distributions $q_{t-\Delta t, t}(x_{t-\Delta t, t} \,|\, x_0)$ over $\mathbb{R}^{2d}$, which are Student distributions with mean $[\mu_{t - \Delta t} x_0, \mu_t x_0]$, covariance matrix $\Sigma_t$, and degrees of freedom parameter $\nu > 1$. Leveraging properties of conditional Student distributions, they demonstrate that the marginals of $q_{t-\Delta t, t}(x_{t-\Delta t, t} \,|\, x_0)$ are also Student-distributed, as are the conditional distributions. Given a family of transitions
$p_{t-\Delta t \, |\, t}$, the authors fit this family by minimizing a modified DM loss, where the standard Kullback-Leibler divergence is replaced with a power divergence. They then incorporate their chosen conditionals $q_{t-\Delta t, t}(x_{t-\Delta t, t} \,|\, x_0)$. Finally, the developed methodology is benchmarked against standard EDM and several of its variations for generating dynamical variables in the High-Resolution Rapid Refresh dataset.

**Strengths:**

The approach appears interesting and addresses a relevant problem. Also, the methodology can be straightforwardly implemented using existing DM frameworks.

**Weaknesses:**

In my opinion, there are several key points the authors should address.

First, a major concern is the soundness of the proposed method. In particular, it is unclear to me that the sequence of conditional distributions $ q $ introduced by the authors genuinely defines a Markov noising process. As such, the loss function they employ does not appear to serve as a meaningful metric. More precisely, it remains uncertain why this loss function would serve as a good proxy for measuring discrepancies between the model marginal $p_0^{\theta}$ and the data distribution. The authors acknowledge that the loss function is not a proper ELBO and claim that the limiting diffusion model is a special case of their method, but further justification of the loss function is needed.

My second concern relates to the results and mathematical formalism of the paper. The mathematical treatment seems quite weak. For instance, the proof in the first section of the appendix contains several typos and appears incorrect in its current form. The reference to Ding (2016) would be sufficient, as it already includes the result. I also found Proposition 2 challenging to interpret, and the accompanying proof does not clarify matters. For example, it is unclear what $f$ and $g$ represent, whether they are any functions satisfying the proposition's conditions, and whether such functions exist. Similarly, the term $ dS_t \sim t_d(0, dt, \nu + d)$ lacks clarity. I am also unclear on the main conclusions to be drawn from Proposition 2.

My third concern is related to the experiments and the metrics used. First, there is a large gap in performance when using $\nu = 3$ versus $\nu = 5$, raising questions about the robustness and reliability of the method. This gap suggests that the chosen metrics (KR, KS, SR) might be sensitive only to the tails of the target distributions. Additional metrics such as precision, recall, and density coverage could provide a more balanced assessment. Moreover, the claims regarding t-EDM's performance appear somewhat overstated, given the tables and the likely margin of error. In my opinion, the experiments should include toy examples for a deeper comparison with existing diffusion models.

Finally, the authors should provide a comprehensive literature review on extensions of diffusion models, acknowledging previous relevant works. The current literature review is too limited.

**Questions:**

See weakness

---

> ### Author Response · Authors · 2024-11-23
> **Official Author Response - Reviewer e3KL**
>
> We thank the reviewer for their comments and address each point in more detail below.
>
> **Weaknesses/Questions**
>
> **Weakness 1a:** *First, a major concern is the soundness of the proposed method. In particular, it is unclear to me that the sequence of conditional distributions q introduced by the authors genuinely defines a Markov noising process.*
>
> **Response**: From our construction of the forward process, the distribution $q(x_t|x_{t-\Delta t})$  can be specified as follows,
>
> $q(x_t|x_{t-\Delta t}) = \int q(x_t|x_{t-\Delta t}, x_0)q(x_0|x_{t-\Delta t}) dx_0$. Therefore, the forward process in our design is non-markovian. However, for most practical scenarios, since diffusion model training solely depends on the perturbation kernel $q(x_t|x_0)$, it does not matter if the forward noising process is Markovian or not. Indeed, prior work in diffusion models like DDIM [1] also relies on using a non-markovian forward process design to achieve sampling speedups.
>
> [1] Denoising Diffusion Implicit Models, Song et al.
>
> **Weakness 1b:** *As such, the loss function they employ does not appear to serve as a meaningful metric. More precisely, it remains uncertain why this loss function would serve as a good proxy for measuring discrepancies between the model marginal $p_0^θ$ and the data distribution. The authors acknowledge that the loss function is not a proper ELBO and claim that the limiting diffusion model is a special case of their method, but further justification of the loss function is needed*
>
> **Response**: This is good feedback that we address in Section 3.3 (lines 220-234) in our revised paper. Below, we argue (in more detail) that minimizing our proposed loss function in Eq. 8 indeed serves as a good proxy for measuring discrepancies between the model and the data distributions for the following reasons.
>
> 1. Firstly, the loss function in Eq. 8 is obtained by replacing the KL divergence in the DDPM objective in Eq. 1  with the gamma power divergence instead. Since the gamma power divergence is a valid divergence (See [2]), in principle, the objective in Eq. 8 is minimized when matching the learnable denoiser with the forward process posterior, which in turn implies $p_0^\theta = p_\text{data}$. This divergence-based perspective of the ELBO objective is also utilized in prior works in generative models. For instance, in the context of diffusion models, [1] replace the KL divergence in the DDPM objective using a GAN instead, which provides significant sampling speedups. Similarly, [2] utilizes a similar perspective for training VAEs for heavy-tailed data distributions. Therefore, replacing the KL divergence with alternative divergences still provides a valid proxy for measuring the discrepancies between the model and the data distributions. However, it is worth noting that different divergences would indeed result in different properties of the learned score function and are suitable for different applications. In our context of heavy-tailed diffusion models, using gamma power divergences is natural since, unlike the KL divergence, they can be computed tractably for Student-t distributions.
> 2. Secondly, in Proposition 1, we formally establish the connection between the KL and gamma power divergence. More specifically, the following equality holds: $D_\gamma = D_{\text{kl}} + O(\gamma)$ where $\gamma = -2/(\nu + d)$ in our work. This formally relates the objective in Eq. 8 to the DDPM objective in Eq. 1. More precisely, both objectives are equivalent in the $\nu \rightarrow \infty$ limit.
> 3. Lastly, the use of alternative divergences like the gamma power divergences to measure the discrepancies between the model and the data distributions in the presence of outliers/contamination is common in robust statistics [3] (referred to as M-estimators). It serves as a solid alternative to MLE-based objectives like ELBO. We also highlight this connection in Section 5 in the main text.
>
> Therefore, using a divergence like the gamma power divergence in Eq. 8 can serve as a valid proxy for measuring the discrepancies between the model and the data distributions
>
> 1] Tackling the Generative Learning Trilemma with Denoising Diffusion GANs, Xiao et al., ICLR’22.
>
> [2] $ t^ 3$-Variational Autoencoder: Learning Heavy-tailed Data with Student's t and Power Divergence, Kim et al., ICLR’24.
>
> [3] Variational inference based on robust divergences, Futami et al.

---

> ### Author Response · Authors · 2024-11-23
> **Official Author Response - Reviewer e3KL (Contd.)**
>
> **Weakness 2a:** *My second concern relates to the results and mathematical formalism of the paper. The mathematical treatment seems quite weak. For instance, the proof in the first section of the appendix contains several typos and appears incorrect in its current form. The reference to Ding (2016) would be sufficient, as it already includes the result*
>
> **Response:** We assume that the reviewer is referring to the proof in Appendix A.1. We have fixed some typos in the proof and made it more elaborate. We also respectfully disagree with the assessment that our mathematical treatment is weak. Some of our core mathematical contributions are:
>
> 1. Designing a student-t noising and denoising process (Sections 3.1-3.2)
> 2. Formulating a training objective for t-diffusions regarding the gamma power divergences (Section 3.3). We show that the training objective for t-diffusions converges to Gaussian diffusion in the limit of $\nu \rightarrow \infty$ by establishing a relation between the KL and the gamma power divergences (Proposition 1). We also show the connection of our loss functions to robust estimators in statistics (Section 5).
> 3. We derive a novel family of continuous-time samplers for our discrete denoising process (Proposition 2) and instantiate it in a way that provides minimal implementation overhead on top of prior diffusion models like EDM (Section 3.5)
> 4. We highlight the proof for all propositions and claims in Appendix A.
>
> Overall, we believe our methodological contributions are solid (as also acknowledged by Reviewers EPoh, Nnew, and uzb2).
>
> **Weakness 2b**: *I also found Proposition 2 challenging to interpret, and the accompanying proof does not clarify matters. For example, it is unclear what $f$ and $g$ represent, whether they are any functions satisfying the proposition's conditions, and whether such functions exist. Similarly, the term $ dS_t \sim t_d(0, dt, \nu + d)$ lacks clarity. I am also unclear on the main conclusions to be drawn from Proposition 2.*
>
> **Response:** As also highlighted in the main text (lines 261-264), continuous-time formulations of diffusion models can be preferred over their discrete-time analogues due to advances in fast sampling for the former. Therefore, the main idea behind Proposition 2 is to develop a family of continuous-time samplers corresponding to the discrete-time formulation of the diffusion model defined in Sections 3.1-3.3. In this context, firstly, the functions $f$ and $g$ represent design choices that a user can specify to construct continuous time samplers. We mention one plausible choice of such functions in lines 274-279 where we set $f=-\frac{\dot{\sigma}_t}{\sigma_t}$ and $g=0$. This results in our ODE solver, which we use throughout our work. Overall, there do exist functions $f$ and $g$ which can provide useful instantiations for sampler design.
>
> **Weakness 2c**: *Similarly, the term $dS_t \sim t_d(0, dt, \nu + d)$ lacks clarity. I am also unclear on the main conclusions to be drawn from Proposition 2.*
>
> **Response:** The term $dS_t \sim t_d(0, dt, \nu + d)$ intuitively means that we add Student-t noise when sampling from this continuous time SDE. More formally, it denotes the stochastic noise differential, which is distributed as a Student-t random variable and ensures that independent increments are Student-t distributed. This is intuitive since our denoising process is stochastic, with the reverse process posterior modeled as a Student-t distribution. This is also clarified in lines 279-281 in our revision.
>
> Overall, the main idea behind Proposition 2 is to design a family of continuous-time samplers for the proposed heavy-tailed diffusion models, which can provide added advantages over the discrete-time case.

---

> ### Author Response · Authors · 2024-11-23
> **Official Author Response - Reviewer e3KL (Contd.)**
>
> **Weakness 3a:** *My third concern is related to the experiments and the metrics used. First, there is a large gap in performance when using $\nu = 3$ versus $\nu = 5$, raising questions about the robustness and reliability of the method.*
>
> **Response:** To address this question, we want to highlight a couple of observations.
>
> Firstly, as highlighted in the main text (Proposition 1), our proposed model t-EDM converges to Gaussian diffusion (EDM) in the limit of $\nu \rightarrow \infty$. This implies that as nu increases, the performance of our method on different metrics should converge towards that of EDM. This is indeed what we observe in the results for unconditional generation in Table 2, where for larger nu (nu=5 or 7), the performance of t-EDM (let's say for the VIL channel) starts moving towards that of EDM. Therefore, this performance deviation is expected, and $\nu$ needs to be tuned accordingly.
>
> Secondly, during training, the noising process in t-EDM can be specified as,
> $x_t = x_0 + \frac{\sigma}{\sqrt{\kappa}}$, where $\kappa \sim \chi^2(\nu)/\nu$. For lower values of $\nu$ (like 3), the quantity $\frac{\sigma}{\sqrt{\kappa}}$ can take large values during training (we present a pdf of the Chi-squared distribution for different nu values in Fig. 8). This effect becomes less pronounced as the value of $\nu$ increases, eventually converging to Gaussian diffusion as $\nu \rightarrow \infty$. Therefore, intuitively, the variability in the model performance for lower values of $\nu$ would be more than the variability at higher $\nu$ values. This is indeed what we observe in Table 1, where the variability in the results for nu=3 and 5 is much more than between nu=5 and 7. This also underscores the importance of tuning $\nu$ correctly for different datasets.
>
> Overall, both these observations justify that our method is reliable for capturing heavy tails, as is also demonstrated in our empirical results in Fig. 3 and Tables 1,2, provided $\nu$ can be tuned reliably for a given dataset.
>
> **Weakness 3b:** *This gap suggests that the chosen metrics (KR, KS, SR) might be sensitive only to the tails of the target distributions. Additional metrics such as precision, recall, and density coverage could provide a more balanced assessment.*
>
> **Response:** Since we focus on comparing different diffusion models based on their ability to capture the tails of the data distribution in this work, our choice of metrics reflects this goal. We agree with the reviewer that additional metrics used in standard generative modeling for images like FID, IS, precision, and recall can provide a more balanced comparison between different models. However, a key caveat in using all these metrics is the reliance on a pre-trained feature extractor (like Inceptionv3/VGG), which is commonly trained on natural images (like ImageNet) for generative modeling of images. However, this work focuses on weather data, which has a much different structure than natural images. Due to this distribution shift, it is unclear if a feature extractor pre-trained on natural images can extract meaningful structural features from weather datasets. Therefore, computing metrics like FID, precision, or recall using a pre-trained network trained on natural images for weather modeling is dubious at best.
>
> In this spirit, training these feature extractors on large-scale weather data using classification or self-supervised losses so that downstream metrics like FID or IS can be reported reliably could be an interesting study on its own, which is outside the scope of this paper. We have also clarified this point in Appendix F.2. Alternatively, we present qualitative histograms between the generated and the train/test set distributions (see Figs. 3,11), which also provide a nice comparison between different methods and demonstrate that t-EDM outperforms other baselines in capturing heavy tails while also modeling the distribution of non-heavy tailed structures accurately.

---

> ### Author Response · Authors · 2024-11-23
> **Official Author Response - Reviewer e3KL (Contd.)**
>
> **Weakness 3c:** *Moreover, the claims regarding t-EDM's performance appear somewhat overstated, given the tables and the likely margin of error.*
>
> **Response:** We would like to highlight that t-EDM largely outperforms the baseline EDM (Row 1) model across all metrics in Table 1. Moreover, our qualitative histogram comparisons in Fig. 3 illustrate that our proposed method outperforms EDM and its improved versions in modeling heavy-tailed variables, as also acknowledged by Reviewers uzb2 and zbkB. We agree that with improved preconditioning (PCP) and preprocessing (INC), EDM outperforms t-EDM in some cases on the KR and SR metrics. However, for the KS metric, which specifically examines the left and right tails by filtering 0.1 and 99.9 percentiles of the data (and is thus arguably a better metric for comparing tail estimation), t-EDM outperforms all other baselines.
>
> Lastly, we would like to highlight that these techniques can also be readily combined with the t-EDM model and should yield similar improvements. Therefore, the main point of comparison in Table 1 should be primarily between t-EDM and the standard baseline EDM (row 1). The current claim in lines 82-83 is based on this observation. We have added this caveat in the main text in Section 4.1. In this context, we feel that we do not overclaim our contributions in this work.
>
> **Weakness 3d:** *In my opinion, the experiments should include toy examples for a deeper comparison with existing diffusion models.*
>
> **Response:** We would like to point the reviewer to Fig. 1, where we compare the performance of the baseline EDM with t-EDM for different $\nu$ values. Fig. 1 highlights that t-EDM provides a parameter $\nu$, which can help control tail estimation and is thus better able to model heavy-tailed distributions as compared to other baselines like standard Gaussian diffusion. We also include additional quantitative results in Table 8 for the same 2-D toy dataset in Appendix C.3 (section “Quantitative Results on Neals Funnel”). From Table 8, our quantitative results support our qualitative results, implying that our proposed model can definitively help with heavy-tailed data modeling. We also include the results from Table 8 below for convenience.
>
> |           | Method |   $\nu$  | KR $\downarrow$ | SR $\downarrow$ | KS $\downarrow$ |
> |:---------:|:------:|:--------:|:---------------:|:---------------:|:---------------:|
> | Baselines |   EDM  | $\infty$ |      0.909      |      2.334      |      0.234      |
> |    Ours   |  t-EDM |     4    |      **0.393**      |      **1.584**      |      **0.051**      |
> |           |  t-EDM |     7    |      0.795      |      3.671      |      0.194      |
> |           |  t-EDM |    10    |      0.863      |      2.198      |      0.261      |
>
> The quantitative results in the above table for the toy dataset support our qualitative results in Figure 1 in the main text and imply that our proposed model can definitively help with heavy-tailed data modeling as compared to baselines like Gaussian Diffusion.
>
> **Weakness 4:** *Finally, the authors should provide a comprehensive literature review on extensions of diffusion models, acknowledging previous relevant works. The current literature review is too limited.*
>
> **Response:** We thank the reviewer for pointing this out. We have updated Appendix F.1 in the paper to add a more thorough literature review.
>
> Overall, we thank the reviewer for their constructive suggestions and request the reviewer to reconsider their evaluation. We look forward to the reviewers’ response and would be happy to respond to additional questions/comments.

---

> > ### Author Response · Authors · 2024-11-27
> > **Revision Upload Deadline**
> >
> > Dear Reviewer e3KL,
> >
> > Since the deadline for uploading a revision is coming up, it would be great if you could acknowledge our response and let us know if further clarifications on any of your questions is required. Thanks for your help in the review process.
> >
> > Best,
> >
> > Authors

---

### Official Review · Reviewer_Nnew · 2024-11-03

**Soundness:** 4
**Presentation:** 3
**Contribution:** 3
**Rating:** 8
**Confidence:** 4

**Summary:**

The paper presents a novel approach to adapting diffusion models for heavy-tail estimation by leveraging multivariate Student-t distributions. This method addresses the limitations of traditional Gaussian-based diffusion models in capturing rare or extreme events, which is particularly relevant for applications such as weather forecasting. The authors introduce extensions of existing diffusion and flow models, termed t-EDM and t-Flow, which utilize a Student-t prior to enable controllable tail generation through a single scalar hyperparameter. The framework is compatible with standard Gaussian diffusion models and requires minimal code changes. Empirical validation on high-resolution datasets, such as the HRRR weather dataset, demonstrates that the proposed t-EDM outperforms existing diffusion models in accurately modeling heavy-tailed distributions.

**Strengths:**

1. The use of multivariate Student-t distributions for diffusion models is innovative, allowing the model to better capture rare or extreme events with controllable tail behavior through a hyperparameter. It is a very brilliant idea.
2. The overall presentation of this paper is very clear and understandable.
3. As an extension of previous EDM and Rectified Flow, the authors established a new framework for Student-T distribution and flow, which is able to capture the rare events more effectively.
4. The authors provide theoretical backing for their approach, linking it to robust statistical estimators and demonstrating its foundation in existing heavy-tailed distribution literature.
5. Choosing the weather dataset to verify the effectiveness is a great idea, and is novel.

**Weaknesses:**

I like this paper a lot and I believe there are no major weakness. The only thing is that, there are some duplicated references in the reference list.

**Questions:**

1.  Can the authors discuss potential applications in other domains where rare events or extreme values are important, such as finance, seismology, or network traffic analysis. This would help demonstrate the broader impact of the method beyond just weather forecasting.
2. Do you think it possible to extend your framework into other heavy-tailed distributions beyond the Student-t, such as the Cauchy or Lévy distributions, or even unifying all these classes of perturbation kernels? Are there any theoretical or practical challenges you foresee in such extensions?

**Details Of Ethics Concerns:**

Ethics Concerns have been clearly written in this paper.

---

> ### Author Response · Authors · 2024-11-23
> **Official Author Response - Reviewer Nnew**
>
> We thank the reviewer for insightful comments and address each point in more detail below.
>
> **Weaknesses/Questions**
>
> **Weakness 1:** *The only thing is that, there are some duplicated references in the reference list.*
>
> **Response:** We have removed duplicated references in our revised version.
>
> **Question 1:** *Can the authors discuss potential applications in other domains where rare events or extreme values are important, such as finance, seismology, or network traffic analysis. This would help demonstrate the broader impact of the method beyond just weather forecasting*.
>
> **Response:** Thanks for the suggestion. We have added a note mentioning other potential applications in Appendix F.2 in lines 2097-2102.
>
> **Question 2:** *Do you think it possible to extend your framework into other heavy-tailed distributions beyond the Student-t, such as the Cauchy or Lévy distributions, or even unifying all these classes of perturbation kernels? Are there any theoretical or practical challenges you foresee in such extensions?*
>
> **Response:** This is a very interesting question. There has been some work on formulating heavy-tailed diffusion models using $\alpha$-stable Levy distributions [Yoon et al.]. However, except Gaussian and Cauchy distributions, Student-t distributions are not $\alpha$-stable in general (refer to Appendix F.1). Thus, a unification of all these classes of diffusions might be non-trivial. Regardless, this is an interesting direction for further theoretical work in this area, which we have specified in our revision in Appendix F.1.
>
> Overall, we thank the reviewer for their positive comments and feedback. We would be happy to answer any follow-up questions.

---

> > ### Comment · Reviewer_Nnew · 2024-11-25
> > **Official Comment**
> >
> > Thanks so much for your clear response. I think it answers my questions and I keep my original score.

---

### Official Review · Reviewer_zbkB · 2024-11-03

**Soundness:** 3
**Presentation:** 3
**Contribution:** 3
**Rating:** 6
**Confidence:** 4

**Summary:**

Diffusion models are widely used, but there has been little work studying how well they capture extreme events in the tail of distributions. This is especially important in environmental prediction where extreme events are often the ones that we care most about. This paper shows that standard approaches can fail to capture the extremes accurately. It develops a new form of diffusion by leveraging the multivariate Student-t distribution to form new noising and denoising processes in discrete and continuous time. It develops a method to fit these new processes using a power-divergence which avoids the intractability of the ELBO.  It then applies the new methods to modelling a pair of variables from HRRR, NOAA’s high-resolution local area system for the continental US. This product has recently been a focus of AI diffusion emulators e.g. in NVIDIA’s StormCast.

**Strengths:**

I really liked the problem the paper was looking into — better characterisation of extremes in diffusion and flow based generative models — especially in the context of environmental forecasting.  I think that this is an important practical problem. I also liked the use of a simple example to analyse how well diffusion models capture tail behaviour. (In fact I would have liked to have seen more investigation to toy examples of this sort to pin down the failure models of diffusion more generally.)

 I thought the writing was very clear and struck a good balance between communicating technical content and the high level idea.

The technical contribution is interesting. I liked the fact that the authors provide both the discrete-time version and the continuous-time version of the new approach. The use of the power-divergence for producing a more tractable training objective was creative.

HRRR was a sensible domain to test the new method on. A range of sensible metrics that are used by practitioners e.g. CRPS for the forecasting experiments. I liked the simple modifications to EDM using the inverse CDF transform and per-channel preconditioning.

**Weaknesses:**

I would have liked to have seen a bit more clarity on the formulation of the denoising process (the sub-optimalities here seem significantly more severe than in the Gaussian case, see questions below for more detail). The use of the power-divergence for training was creative, but connecting the parameters of the model to the divergence parameters also seemed sub-optimal as it does not allow the framework to learn the degree of freedom parameter in the student-t distribution.

The experimentation was a little light in that only two variables were considered in the conditional and unconditional settings, there were no experiments that considered forecasting multiple variables at one time, or autoregressive rollouts.

Setting the degree of freedom parameter via a parameter sweep is a bit painful. It’s a shame that it can’t be learned directly, especially if it is going to be different for different dimensions of x  (locations / variables / pressure levels etc.).

**Questions:**

Figure 1 - I liked this! I thought it could be improved by adding axis labels including a description of what the colour map shows (log ground truth density?), calling the original density “ground truth” ("original hists” in the legend was slightly cryptic),  would add axis labels.

I was a bit confused by the notation around the degree of freedom parameter v. Initially in figure 1 it is a vector, then in equation 2 it is a scalar, but then in the equations on line 155 we switch back to a vector with t_{2d}() being a composition of d independent 2-dimensional multivariate student-t distributions (although this isn’t explained). The issue around clarity here bleeds into the discussion of the divergences where then the divergence parameter \gamma would depend on the dimension. Balancing notational clarity and correctness is tricky here — perhaps say that "we’ll consider v to be constant across dimensions for the mathematical development, but the generalisation is simple" somewhere early on to clarify these issues?

There is a lack of clarity in my mind around the noising process and the approximations made when arriving at the denoising equation (equation 5). Wouldn’t the true denoising process be non-Markov? If you consider the student-t as a Gaussian scale mixture, I believe the latent scale parameter in your noising construction is effectively shared across all time-steps (a consequence of the equations on line 155). So, when generating x_t the whole history of previous x_{t+1:T} will inform us about the latent scale and so p(x_t | x_{t+1:T}) \ne  p(x_t | x_{t+1}). Have I misunderstood? Moreover, dropping the data dependent coefficient (v+d_1)/(v+d) is rather glossed over. In the Gaussian setting, dropping this can be somewhat justified by 1. it being an exact choice in the limit of small step sizes, and 2. at least it’s exact when the true distribution of x_0 is Gaussian. In the current case neither of these statements hold I believe. I felt that this should be discussed as the suboptimality of this choice in this case is likely far larger than in the usual case.  Finally, could you clarify how the continuous case fits in with this? I presume, both of the approximations mentioned above are made, and then we take the continuum limit?

Connecting the specific divergence used to a parameters in the model / denoising process (i.e. \gamma = 2/(v+d)) might be limiting e.g. the objective now can’t be used to learn the degree of freedom, the optimisation objective (8) cannot be used to compare between models that use different settings for v. Also, the correspondence to the traditional objective only then holds as q becomes Gaussian (ideally you’d like to recover the original ELBO without having to limit the model back to the DDPM). I’d have liked to have seen more discussion of this point.

In table 1, it might be useful to add p(x_T) for the two cases (by the way do you mention somewhere that q(x_T) is a product of student-ts and that you set p(x_T) = q(x_T))?

Why not consider surface fields, arguably where modelling extremes have most impact?

Have you considered a baseline that trains the EDM and then uses a base distribution for generation which is zero-mean unit-variance student-t, sweeping the degree of freedom parameter to find a suitable one?

For the HRRR experiments, I presume that you are training off HRRR analysis (rather than, say forecasts) this should be mentioned (I didn’t see it in the main text or in the appendix). Also, is the time interval between inputs mentioned? Is it 1hr like in StormCast?

The spatial fields shown in the appendix were useful. For the forecasts results shown in figure 8 and 9, it would be useful to show the persistence forecast.

For the unconditional sampling, it looks like on the test set t-EDM outperforms the baselines 1/3 of the time (and only on the KS metric).

Is the KS metric to be more trusted? Why weren’t the other metrics applied to the forecasting case too?

It would be interesting to see how t-Flow behaved during auto-regressive rollouts. I could imagine that the under-dispersion of EDM might actually help maintain stability in these cases, whereas heavier tails could lead to greater instability.

---

> ### Author Response · Authors · 2024-11-25
> **Official Author Response - Reviewer zbkB**
>
> We thank the reviewer for their constructive comments and address each point in more detail below.
>
> **Weaknesses/Questions**
>
> **Weakness 1:** *I would have liked to have seen a bit more clarity on the formulation of the denoising process (the sub-optimalities here seem significantly more severe than in the Gaussian case, see questions below for more detail). The use of the power-divergence for training was creative, but connecting the parameters of the model to the divergence parameters also seemed sub-optimal as it does not allow the framework to learn the degree of freedom parameter in the student-t distribution.*
>
> **Response:** Regarding learning the degree of freedom parameter $\nu$, while we don’t learn this parameter in this work, there can be several ways to learn this parameter directly from the data.
>
> Firstly, it is worth noting that the parameter $\nu$ directly influences training in heavy-tailed diffusions. This is because the noising process $x_t = \mu_t x_0 + \frac{z}{\sqrt{\kappa}}$ where $\kappa \sim \chi^2(\nu)/nu$. Since the Chi-squared distribution is a special case of a Gamma distribution, we can perhaps use reparameterization [1] to estimate the parameters (i.e., $\nu$) of this distribution using a neural network and backprop end-to-end to learn this parameter. This would correspond to learning $\nu$ directly from the data.
>
> [1] The Generalized Reparameterization Gradient, Ruiz et al.
>
> Alternatively, one can also estimate $\nu$ from the data in a separate stage and use it as a perturbation kernel parameter. We highlight one simple way to do this in Appendix D, where we attempt to derive the optimal noise schedule for EDM and t-EDM by minimizing the mutual information between the $x_0$ and the noisy data point $x_\sigma$ w.r.t $\sigma$. In principle, for t-EDM, we can also solve this minimization problem jointly over $\nu$ and $\sigma$ to learn an initial $\nu$ from the data.
>
> We have added these points in Appendix F.2 in our revision. Overall, while manual tuning of $\nu$ might seem to be a sub-optimality of our framework, we would like to point out that the proposed framework can be readily extended to learn $\nu$ in an end-to-end or separate stage fashion.
>
> **Weakness 2:** *The experimentation was a little light in that only two variables were considered in the conditional and unconditional settings, there were no experiments that considered forecasting multiple variables at one time, or autoregressive rollouts.*
>
> **Response:** Regarding the choice of variables, we chose variables that are most heavy-tailed in the HRRR dataset to illustrate the efficacy of our framework. Regarding multiple variables, since tuning $\nu$ per variable can be quite tedious, we leave exploration of this aspect to further work. We also highlight some methods for automatically tuning this parameter in our response to Weakness 1 above. Since we can combine techniques like per-channel-preconditioning, i.e., PCP and automatic $\nu$ tuning with heavy-tailed diffusion models, the design space for experimentation when exploring these models in the context of autoregressive rollouts in weather forecasting is quite extensive and, therefore, beyond the scope of this work. We mention this as a limitation in Appendix F.2 in our revised paper.
>
> **Weakness 3:** *Setting the degree of freedom parameter via a parameter sweep is a bit painful. It’s a shame that it can’t be learned directly, especially if it is going to be different for different dimensions of x (locations / variables / pressure levels etc.).*
>
> **Response:** Kindly, refer to our response to Weakness 1 above, where we clarify some ways to tune $\nu$ directly from the data.

---

> > ### Author Response · Authors · 2024-11-25
> > **Official Author Response - Reviewer zbkB (Contd.)**
> >
> > **Question 1:** *Figure 1 - I liked this! I thought it could be improved by adding axis labels including a description of what the colour map shows (log ground truth density?), calling the original density “ground truth” ("original hists” in the legend was slightly cryptic), would add axis labels.*
> >
> > **Response:** We thank the reviewer for pointing this out. We have added axis labels, including a description for the color maps, and updated the label for the original density. Kindly, refer to Fig. 1 in our revised version.
> >
> > **Question 2:** *I was a bit confused by the notation around the degree of freedom parameter v. Initially in figure 1 it is a vector, then in equation 2 it is a scalar, but then in the equations on line 155 we switch back to a vector with t_{2d}() being a composition of d independent 2-dimensional multivariate student-t distributions (although this isn’t explained). The issue around clarity here bleeds into the discussion of the divergences where then the divergence parameter \gamma would depend on the dimension. Balancing notational clarity and correctness is tricky here — perhaps say that "we’ll consider v to be constant across dimensions for the mathematical development, but the generalisation is simple" somewhere early on to clarify these issues?*
> >
> > **Response:** Thanks for the suggestion. We have added this note in our revision in lines 135-136.
> >
> > **Question 3a:** *There is a lack of clarity in my mind around the noising process and the approximations made when arriving at the denoising equation (equation 5). Wouldn’t the true denoising process be non-Markov? If you consider the student-t as a Gaussian scale mixture, I believe the latent scale parameter in your noising construction is effectively shared across all time-steps (a consequence of the equations on line 155). So, when generating x_t the whole history of previous x_{t+1:T} will inform us about the latent scale and so p(x_t | x_{t+1:T}) \ne p(x_t | x_{t+1}). Have I misunderstood?*
> >
> > **Response:** While the perspective of treating the Student-t distribution as a scale-mixture could be useful, more formal treatment of examining whether the underlying denoising process is Markov or not is to consider the reverse SDE formulation as considered in Song et al. However, for Student-t distributions, this would likely involve working with Levy processes and their corresponding Planck equations, which is outside the scope of this work. Regardless, our choice of the denoising process as a Markov chain is inspired by prior work in diffusion models. Indeed, this choice results in a simple algorithm that works well in practice.
> >
> > **Question 3b:** *Moreover, dropping the data dependent coefficient $\frac{v+d_1}{v+d}$ is rather glossed over. In the Gaussian setting, dropping this can be somewhat justified by 1. it being an exact choice in the limit of small step sizes, and 2. at least it’s exact when the true distribution of $x_0$ is Gaussian. In the current case neither of these statements hold I believe.*
> >
> > **Response:** There might be some misunderstanding here. We note that the data-dependent coefficient $(v+d_1)/(v+d)$ in the forward process posterior arises from the conditional properties of multivariate Student-t distributions. In the Gaussian setting, the parameter $\nu \rightarrow \infty$, and therefore the coefficient $(v+d_1)/(v+d)$ converges to 1. Thus, the scalar coefficient can be dropped without any additional caveats for Gaussian diffusion models.
> >
> > **Question 3c:** *I felt that this should be discussed as the suboptimality of this choice in this case is likely far larger than in the usual case*
> >
> > **Response:** In the reverse process parameterization in Eqn. 5, we argue that dropping this term is a practical design choice, primarily inspired by simplicity and good empirical results. More specifically,
> >
> > Firstly, dropping this term leads to ease of designing preconditioners and more flexible training like EDM while enabling modifications on top of EDM during training.
> >
> > Secondly, when dropping the data-dependent term, our ODE sampler presented in Section 3.4 has the same formulation as EDM with the main difference in sampling the latents from a Student-t prior. This leads to minimal implementation overhead during training and sampling.
> >
> > Lastly, more importantly, our empirical results justify this design choice.
> > All these factors were crucial in selecting the reverse process parameterization in Eqn. 5. In this context, our parameterization in Eqn. 5 does not seem sub-optimal (though this is quite subjective since more “efficient” parameterizations not explored in this work might exist).

---

> > > ### Comment · Reviewer_zbkB · 2024-11-26
> > > **Response to question 3a and 3b.**
> > >
> > > Thanks again for expanding on this.
> > >
> > > With regard to 3a: The specific issue I have is that it appears that your equations are inconsistent. Doesn't the equation on line 156 imply that all variables x_{1:T} jointly follow a multivariate Student-t? If so, then the noising process and therefore the optimal denoising process will not be Markov, but you then assume a Markov denoising process.
> > >
> > > With regard to 3b: my point is that when you say "This aligns with prior works in diffusion models (Ho et al., 2020; Song et al., 2020; Karras et al., 2022) where it is common to only parameterize the denoiser mean" this choice in prior work is somewhat justified as, in certain limits and special cases, this term does not depend on the data and can simply be fixed to a value that depends on the noising process. However, your case is very different as you have shown it does depend on the data. So, this seems far more risky and potentially misleading to compare to the standard case.

---

> > ### Comment · Reviewer_zbkB · 2024-11-26
> > **Response on learning the degree of freedom parameter**
> >
> > Thanks for the responses.
> >
> > The issue I was raising is that the divergence you use has a free parameter which you then tie to the degree of freedom parameter. If you then optimise this objective wrt the degree of freedom parameter, you change the divergence as the parameter changes. This means you don't know whether this will actually result in a degree of freedom which fits the data -- rather it could just select a divergence which is more forgiving. I looked at the Appendix F.2 and D.2 and it seems that this fact is overlooked.

---

> ### Author Response · Authors · 2024-11-25
> **Official Author Response - Reviewer zbkB (Contd.)**
>
> **Question 3d:** *Finally, could you clarify how the continuous case fits in with this? I presume, both of the approximations mentioned above are made, and then we take the continuum limit?*
>
> **Response:** We assume that the assumptions that the reviewer refers to are the markovianity of the reverse posterior and the parameterization in Eqn. 5. In this case, the continuum limit is taken after these two design choices.
>
> **Question 4a:** *Connecting the specific divergence used to a parameters in the model / denoising process (i.e. $\gamma = 2/(v+d)$) might be limiting e.g. the objective now can’t be used to learn the degree of freedom, the optimisation objective (8) cannot be used to compare between models that use different settings for $\nu$*
>
> **Response:** Please see our response to Weakness 1, highlighting several methods to learn $\nu$ from the data.
>
> **Question 4b:** *Also, the correspondence to the traditional objective only then holds as q becomes Gaussian (ideally, you’d like to recover the original ELBO without having to limit the model back to the DDPM). I’d have liked to have seen more discussion of this point.*
>
> **Response:** In addition to Proposition 1, which formally presents a connection between the objectives in Eq. 1 (i.e. standard ELBO) and Eq. 8 (t-Diffusion objective), prior works in generative models have also considered the divergence-based perspective of the ELBO objective in Eq. 1. For instance, in the context of diffusion models, [1] replace the KL divergence in the DDPM objective using a GAN instead, which provides significant sampling speedups. Similarly, [2] utilizes a similar perspective for training VAEs for heavy-tailed data distributions. Moreover, replacing the KL divergence with robust divergences for estimation under contamination is commonplace in robust statistics (commonly referred to as M-estimators) [3]. Therefore, replacing the KL divergence with alternative divergences is still principled. We have clarified this point in more detail in Section 3.3 (lines 220-234) of our revision.
>
> 1] Tackling the Generative Learning Trilemma with Denoising Diffusion GANs, Xiao et al., ICLR’22.
>
> [2] $ t^ 3$-Variational Autoencoder: Learning Heavy-tailed Data with Student's t and Power Divergence, Kim et al., ICLR’24.
>
> [3] Variational inference based on robust divergences, Futami et al.
>
> **Question 5:** *In table 1, it might be useful to add p(x_T) for the two cases (by the way do you mention somewhere that q(x_T) is a product of student-ts and that you set p(x_T) = q(x_T))?*
>
> **Response:** We added another entry in Table 1 indicating the generative prior for Gaussian and t-diffusions. We also highlight that we set the generative prior to $p(x_T)=q(x_T|x_0)$ (this is due to the first term in the t-diffusion objective in Eqn. 8).
>
> **Question 6**: *Why not consider surface fields, arguably where modelling extremes have most impact?*
>
> **Response:** We agree that many impactful surface fields like precipitation are important heavy-tailed variables in atmospheric sciences. Our initial analysis uncovered that liquid cloud concentration was especially heavy-tailed in the dataset at hand – therefore we focused on this most challenging edge case to model. We feel it is natural to expect associated benefits to precipitation modeling. Meanwhile, it is reasonable for people to disagree about how to choose a particular field to demonstrate this method. We hope the reviewer appreciates our own rationale and very much share their interest in seeing these methods extended to benefit surface hydrological modeling.
>
> **Question 7:** *Have you considered a baseline that trains the EDM and then uses a base distribution for generation which is zero-mean unit-variance student-t, sweeping the degree of freedom parameter to find a suitable one?*
>
> **Response:** Thanks for the suggestion. We assume the main objective with such a baseline could be to tune the $\nu$ parameter. While we did not consider such a baseline, we highlight some computational methods that are more principled and can be used for learning $\nu$ in our response to Weakness 1. In contrast, tuning this parameter during sampling could be computationally expensive due to the iterative sampling procedure of diffusion models.

---

> ### Author Response · Authors · 2024-11-25
> **Official Author Response - Reviewer zbkB (Contd.)**
>
> **Question 8:** *For the HRRR experiments, I presume that you are training off HRRR analysis (rather than, say forecasts) this should be mentioned (I didn’t see it in the main text or in the appendix). Also, is the time interval between inputs mentioned? Is it 1hr like in StormCast?*
>
> **Response:** The reviewer is correct that we used data similar to what was trained in *StormCast*. To be clear, this is not the analysis but rather model forecast fields 1-hour post-analysis. Such a time lag is necessary to allow for some additional model spin-up; cloud condensate is not directly assimilated. We have added a sentence in Appendix C.2.1 to clarify this. We expect the methods presented to work equally well with model forecast or heavy-tailed fields available in the analysis without losing generality.
>
> **Question 9:** *The spatial fields shown in the appendix were useful. For the forecasts results shown in figure 8 and 9, it would be useful to show the persistence forecast.*
>
> **Response:** Thanks for the suggestion. We will include persistence forecasts in a subsequent revision.
>
> **Question 10:** *For the unconditional sampling, it looks like on the test set t-EDM outperforms the baselines 1/3 of the time (and only on the KS metric).*
>
> *Is the KS metric to be more trusted? Why weren’t the other metrics applied to the forecasting case too?*
>
> **Response:** This is an interesting observation. One reason for our method exhibiting better performance for the KS metric could be the fact that the KS metric, in addition to flattening the data, is computed over 99.9th and 0.1th percentiles for the right and left tails, respectively. This implies that the KS metric specifically examines the tail values and thus represents an appropriate metric to capture the difference between the tail estimation for the generated and the data samples. In contrast, the KR and SR metrics are based on computing higher-order moments over the entire domain of flattened empirical samples and thus might be more coarse in nature, hence the discrepancy.
>
> However, it is also worth noting that while we present EDM baselines with improved preconditioning and preprocessing as additional baselines, these can also be applied to t-EDM, potentially improving results. Therefore, a more fair comparison should be between t-EDM and the baseline EDM for which t-EDM outperforms EDM by a significant margin in all metrics. We added this observation in Section 4.1. (lines 406-411).
>
> For conditional modeling, since standard metrics like CRPS, RMSE, and SSR already capture forecasting performance while the KS metric captures tail coverage, we do not use KR and SR metrics.
>
> **Question 11:** *It would be interesting to see how t-Flow behaved during auto-regressive rollouts. I could imagine that the under-dispersion of EDM might actually help maintain stability in these cases, whereas heavier tails could lead to greater instability.*
>
> **Response:** In this work we only consider the next-frame prediction task for which we did not observe any instability issues when working with t-EDM or t-Flow models. While examining this behavior for autoregressive rollouts is indeed interesting, it feels beyond the scope of this work due to space constraints.
>
> Overall, we thank the reviewer for their constructive feedback and for highlighting many interesting points in their review. We would also be happy to answer any subsequent questions.

---

> > ### Comment · Reviewer_zbkB · 2024-11-26
> > **Response to question 11**
> >
> > Thanks for the response. I agree that auto-regressive rollouts introduce complexity, but aren't they absolutely key for the downstream application? Although predicting the next hour ahead is useful, the key application is to rollout for multiple time-steps to make a full forecast. Standard diffusion has been shown to be extremely stable under rollouts (e.g. GenCast) and it would be an issue if the heavy-tailed variant was not.

---

> > > ### Author Response · Authors · 2024-11-27
> > > **Common Response**
> > >
> > > We thank the reviewer for presenting additional clarifications and questions. Please find our response below:
> > >
> > > **Re. tuning the nu parameter:** Thanks for the additional clarification. We have added the following explanation in Appendix F.2 (Lines 2077-2079):
> > > ```
> > > One important caveat in learning $\nu$ end-to-end is that the optimization process could just select a divergence which reduces the loss while not fitting $\nu$ on the data. Therefore, exploring these caveats for learning $\nu$ are an important direction for further research.
> > > ```
> > >
> > > **On Question 3a**: It is safe to argue that the forward noising process is non-markovian. However, the joint distribution of variables $x_{1:T}$ will not be a multivariate Student-t as: $q(x_{1:T}|x_0) = \prod_{t=1} q(x_t|x_{t-1})$. Since the distribution $q(x_t|x_{t-1}) = \int q(x_t|x_{t-1},x_0)q(x_0) dx_0$, the resulting joint will not be a multivariate Student-t. However, the non-markovian argument still holds. Consequently, modeling the denoiser as a Markovian process might appear sub-optimal from a theoretical perspective, but it often works well in practice. In fact, popular diffusion models like DDIM [Song et al.] have a non-markovian forward process but a Markovian denoiser. We have added the following clarification to the main text:
> > >
> > > ```
> > > It is worth noting that while the noising process defined in Eq. 3 is non-markovian, our parameterization of the posterior is still Markovian. However, similar to DDIM, this choice works well empirically (see Section 4).
> > > ```
> > >
> > > **On Question 3b**: We agree and replace this text by the following clarification:
> > > ```
> > > Moreover, when parameterizing the reverse posterior scale, we drop the data-dependent coefficient $(\nu + d_1)/(\nu + d)$. This choice is primarily inspired by simplicity in deriving preconditioners (Sec. 3.5) and developing continuous-time sampling methods (Sec. 3.4) for heavy-tailed diffusions, resulting in models that require minimal implementation overhead over standard diffusion models during training and sampling (see Fig. 2)
> > > ```
> > > Our changes can be found in lines 185-192.
> > >
> > > **Re. Question 11:** We add qualitative rollout trajectories (for lead times up to 10 hours) for the w20 channel in Figure 15. We find that autoregressive rollouts using t-EDM show no signs of instability (as evident from qualitative visualizations on this channel). We are working on additional quantitative results for the same and will try to upload the next revision by the deadline.
> > >
> > > Thanks again for your valuable feedback, and we look forward to further discussions.

---

### Official Review · Reviewer_M1e3 · 2024-11-03

**Soundness:** 3
**Presentation:** 2
**Contribution:** 2
**Rating:** 6
**Confidence:** 3

**Summary:**

This paper adds to the line of work investigating training diffusion models using heavy-tailed noise. Specifically the paper investigates the setting where the increments of noise added in a discrete diffusion model are drawn from a student T distribution.

New methods are developed for training the diffusion model based on power divergences.

**Strengths:**

- The paper introduces a new way of training a score based model with heavy tailed noise, based on the student t distribution, and shows clear benefits in training.
- The experimental results are reasonably convincing.

**Weaknesses:**

- The paper fails to discuss previous work, for example "Heavy-tailed denoising score matching" covering the subject matter.
- The paper lacks experiments on synthetic data which is generated as heavy tailed. While not needed, I believe it would help make the case that this method definitively helps with modeling heavy tails by demonstrating it on definitively heavy tailed data.

**Questions:**

- Can the authors comment on the draw backs of using student t noise discussed in "Heavy-tailed denoising score matching" and why this makes sense in the context proposed?
- proposition 2 to me is not well defined. The construction of the stochastic process with student t increments. Can the authors discuss how such a process is constructed in more detail?
- In the proof of proposition 2 the authors derive the last term converging to Brownian motion, not a student process, can they please comment on this discrepancy.
- It seems unintuitive to define a diffusion model via its finite increments, only to show it converges to a continuous time diffusion and later make use of this perspective. Can the authors comment on why the did not simply define the noising process as a student-t diffusion in the first place and later discretise this?

---

> ### Author Response · Authors · 2024-11-23
> **Official Author Response - Reviewer M1e3**
>
> We thank the reviewer for insightful comments and address each point in more detail below.
>
> **Weaknesses/Questions**
>
> **Weakness 1:** *The paper fails to discuss previous work, for example "Heavy-tailed denoising score matching" covering the subject matter.*
>
> **Response**: We thank the reviewer for mentioning this work. However, several key differences exist between our proposed methods and the Generalized Normal distribution-based score matching introduced in the mentioned paper, which we have clarified in Section F.1 in our revision with a more detailed explanation (also see our response to Question 1 below).
>
> **Weakness 2:** *The paper lacks experiments on synthetic data which is generated as heavy tailed. While not needed, I believe it would help make the case that this method definitively helps with modeling heavy tails by demonstrating it on definitively heavy tailed data.*
>
> **Response:** We would like to point the reviewer to Fig. 1, where we compare the performance of the baseline EDM with t-EDM for different $\nu$ values on a 2-D toy dataset (also referred to as the Neal’s Funnel dataset) with heavy tails along the x-axis dimension. Fig. 1 highlights that t-EDM provides a parameter $\nu$, which can help control tail estimation and is thus better able to model heavy-tailed distributions as compared to other baselines like standard Gaussian diffusion. We also include additional quantitative results in Table 8 for the same 2-D toy dataset in Appendix C.3 (section “Quantitative Results on Neals Funnel”). We also include the results from Table 8 below for convenience.
>
> |           | Method |   $\nu$  | KR $\downarrow$ | SR $\downarrow$ | KS $\downarrow$ |
> |:---------:|:------:|:--------:|:---------------:|:---------------:|:---------------:|
> | Baselines |   EDM  | $\infty$ |      0.909      |      2.334      |      0.234      |
> |    Ours   |  t-EDM |     4    |      **0.393**      |      **1.584**      |      **0.051**      |
> |           |  t-EDM |     7    |      0.795      |      3.671      |      0.194      |
> |           |  t-EDM |    10    |      0.863      |      2.198      |      0.261      |
>
> The quantitative results in the above table for the toy dataset support our qualitative results in Figure 1 in the main text, and imply that our proposed model can definitively help with heavy-tailed data modeling as compared to baselines like Gaussian Diffusion.
>
> **Question 1:** *Can the authors comment on the drawbacks of using student t noise discussed in "Heavy-tailed denoising score matching" and why this makes sense in the context proposed?*
>
> **Response:** We assume the reviewer refers to the paper [1] "Heavy-tailed denoising score matching" by Deasy et al. The motivation to use heavy-tailed noising distributions is similar between [1] and our work, i.e., using base Gaussian distribution for noising concentrates on a narrow shell and neglects the tails (lines 045-046 in the main text). However, in this context, it is worth noting a few key differences from [1]:
>
> 1. **Relevance of using Student-t distributions**: The choice of a Generalized Normal Distribution in [1] as a noising process is mostly inspired by the convenience of working with finite moments of the underlying noisy state vector. The choice of the Student-t distribution in [1] is discouraged due to non-finite moments. However, we argue that this is not representative of whether a Student-t kernel can/cannot be used for modeling heavy-tailed distributions. In fact, as we show in this work, the Student-t based noising process outperforms traditional Gaussian diffusion models for modeling heavy-tailed data distributions (for instance, see Fig. 1). Moreover, as we discuss in Appendix F.1, several prior works have utilized Student-t distributions for other model families like VAEs [2] and likelihood-based flows [3], justifying our choice of the heavy-tailed distribution.
> 2. **Easy Sampler Design**: In [1], the authors rely on annealed Langevin dynamics for sampling and briefly discuss the existence of a reverse SDE for sampling in Appendix B.3. However, an exact formulation of such a reverse process is missing from [1] and would involve levy formulations of Kolmogorov’s forward and reverse equations which can be quite complex. In contrast, due to the conditional properties of multivariate Student-t distributions, our method does not rely on a reverse process formulation based on Levy processes (see Proposition 2 in the main text). Therefore, designing samplers using our proposed Student-t diffusion formulation is easier than the Generalized Normal distribution-based noising formulation discussed in [1].
>
> Therefore, using a Student-t formulation makes sense in the proposed context. We have included these details in Appendix F.1.
>
> [2] Tails of Lipschitz triangular flows, Jaini et al.
>
> [3] $ t^ 3$-Variational Autoencoder: Learning Heavy-tailed Data with Student's t and Power Divergence, Kim et al.

---

> ### Author Response · Authors · 2024-11-23
> **Official Author Response - Reviewer M1e3 (Contd.)**
>
> **Question 2** *proposition 2 to me is not well defined. The construction of the stochastic process with student t increments. Can the authors discuss how such a process is constructed in more detail?*
>
> **Response:** We re-highlight the overall construction for our heavy-tailed diffusion process:
>
> 1. **Forward Process:** We parameterize the forward process using Student-t distributions. More specifically, we define a joint distribution over $x_t$ and $x_{t-\Delta t}$ and then compute the perturbation kernel $q(x_t|x_0)$ using properties of multivariate Student-t distributions, which is utilized during training. Kindly refer to Sections 3.1-3.3 for more details on constructing the forward noising process. We would also like to refer the reviewer to Figure 2 (Algorithm 1) for an overview of the noising process and how it is utilized during training.
> 2. **Reverse Process:** We model the reverse process using a Student-t distribution based stochastic process. More specifically, we parameterize the denoising diffusion posterior $p(x_{t-\Delta t}|x_t)$ using a denoiser $D_\theta(x_t, t)$. Given a pre-trained denoiser, we can perform ancestral sampling from the posterior $p(x_{t-\Delta t}|x_t)$ during sampling. Alternatively, we can derive a continuous time formulation of our reverse process and perform sampling using off-the-shelf fast samplers for diffusion models like DDIM, EDM-ODE, etc. The latter approach is preferable since it provides better sampling speedups. Proposition 2 provides a way to construct such continuous time samplers in this context. While the proof for Proposition 2 is provided in Appendix A.5, the main idea is to “de-discretize” the posterior update in Eqn. 5 to design a family of continuous-time samplers parameterized using user-specified functions $f$ and $g$, which satisfy the constraint (as stated in Proposition 2). We also provide an instantiation of these functions and derive an ODE solver, which we use throughout this work. An overview of the sampling process using our ODE solver is given in Figure 2 (Algorithm 2).
>
> We hope that this high-level overview clarifies the construction of the process with Student-t increments and the significance of the result in Proposition 2.
>
> **Question 3:** *In the proof of proposition 2 the authors derive the last term converging to Brownian motion, not a student process, can they please comment on this discrepancy.*
>
> **Response:** There might be some misunderstanding here. The noise increments in Proposition 2 are indeed distributed as Student-t random variables, as also stated in the statement for Proposition 2 on line 276. We do notice that there is a final step missing in the proof in Appendix A.5 where it can be stated that the random variable $dS_t = dW_t / \sqrt{\kappa_t}$ is Student-t distributed as $\kappa_t \sim \chi^2$ distributed. We have clarified this line in more detail in the proof in Appendix A.5 to make this more explicit for the readers.

---

> ### Author Response · Authors · 2024-11-23
> **Official Author Response - Reviewer M1e3 (Contd.)**
>
> **Question 4:** *It seems unintuitive to define a diffusion model via its finite increments, only to show it converges to a continuous time diffusion and later make use of this perspective. Can the authors comment on why the did not simply define the noising process as a student-t diffusion in the first place and later discretise this?*
>
> **Response:** We understand different presentation styles might be possible for the methods introduced in this work. However, our presentation of the discrete-time version followed by the continuous-time formulation is based on the following rationale:
>
>
>
> 1. Firstly, presenting the discrete-time version highlights key design choices involved in the noising process design. For instance, when designing the forward noising process, we directly compute the perturbation kernel $q(x_t|x_0)$ without specifying the transition kernel $q(x_t|x_{t-\Delta t})$, which is non-markovian under our design. This is justified since only the former is utilized during diffusion model training. We also highlight this aspect in the main text in lines 149-150. Similarly, the discrete-time perspective highlights the choice of $\sigma_{12}(t)$ and $\sigma_{21}(t)$ as important for sampler design. The latter might be ignored in the continuous time presentation.
> 2. Secondly, presenting the discrete-time version also highlights the interpretation of our “x0-prediction” objective as minimizing the gamma-power divergence between the forward and the reverse posterior distributions, providing additional intuition on why our proposed diffusion model can capture heavy tails. On the contrary, this observation is non-trivial when directly defining the continuous-time denoising process followed by the “x0-prediction” training objective.
> 3. Lastly, presenting the discrete-time version first might be more accessible to readers with a background in discrete-time DDPM [Ho et al.] and not very familiar with the continuous-time theory of diffusion processes. This is because continuous treatment of t-diffusion models would most likely involve dealing with Kolmogorov equations for Levy-like processes, making the presentation very dense.
>
> Overall, we thank the reviewer for their constructive suggestions and questions. We look forward to the reviewers’ response and would be happy to respond to additional questions/comments.

---

> > ### Author Response · Authors · 2024-11-27
> > **Revision Upload Deadline**
> >
> > Dear Reviewer M1e3,
> >
> > Since the deadline for uploading a revision is coming up, it would be great if you could acknowledge our response and let us know if further clarifications on any of your questions is required. Thanks for your help in the review process.
> >
> > Best,
> >
> > Authors

---

### Official Review · Reviewer_uzb2 · 2024-11-04

**Soundness:** 3
**Presentation:** 3
**Contribution:** 3
**Rating:** 6
**Confidence:** 3

**Summary:**

The authors provide a framework, termed t-EDM, for diffusion models where the added noise is heavy-tailed, following a Student-t distribution. They build upon the EDM introduced by Karras et al 2022. Interestingly, starting with the EDM framework, replacing the Gaussian noise by the Student-t distribution and changing the KL divergence in the objective function to $\gamma-$Power Divergence for a specific $\gamma$ leads to same loss function as that of the EDM, except that the noisy state is now drawn from a Student-t distribution. This is of practical interest since their framework can be obtained by only performing minimal changes to previous implementations of EDM. It is of theoretical interest that since the Student-t distribution with $\nu$ degrees of freedom converges to a Gaussian distribution as $\nu$ goes to infinity, t-EDM is a generalization of the EDM. The authors provide experiments on weather prediction showing that t-EDM outperforms EDM for both unconditional and conditional generation for certain metrics.

**Strengths:**

The t-EDM provides a straightforward and mathematically justified extension for diffusion models to heavy-tailed data. This is to be compared, for instance, to Yoon et al (2023) who also provide a framework for heavy-tailed diffusion models (using Levy processes), but whose implementation requires heavier mathematics and to make the model work in practice they make theoretically unjustified changes to the model.

The authors do a good job presenting the concepts as natural extensions of what was already known in the field.

The experiments make the case for t-EDM being useful for weather modeling, an application that requires capturing tails and extreme events. Although the present reviewer is not knowledgable in this application, it is clear that having an extension of EDM to heavy-tails is an important contribution to scientists working on weather forecasting and several other natural science applications. It is especially useful that this implementation requires only minimal changes over previous EDM implementation, so it is quick to implement for practitioners.

One advantage of the t-EDM is that it has a free parameter $\nu$ that can be tuned to model different datasets (see results in Table 4), providing a flexible framework useful for practitioners.

Yoon, E. B., Park, K., Kim, S., & Lim, S. (2023). Score-based Generative Models with Lévy Processes. In A. Oh, T. Naumann, A. Globerson, K. Saenko, M. Hardt, & S. Levine (Eds.), Advances in Neural Information Processing Systems (Vol. 36, pp. 40694–40707). Curran Associates, Inc.

**Weaknesses:**

The experiments for unconditional generation in Table 2 show that the EDM (using preconditioning) outperforms the t-EDM for the KR and SR metric in the VIL channel with test data. This hinders the claim the authors make that standard diffusion models, even with preconditioning, fail to capture the heavy-tailed distributions (line 043) and that t-EDM outperforms standard diffusion models. Qualitatively, the plots in Figure 3 do show that t-EDM outperforms EDM with preconditioning in capturing the heavy-tailed behavior. However, I believe a justification of why the KR and SR metrics do not reflect this should be included. As addressed in Appendix F.2, this metrics rely on flattening the data. This may be one reason, but the metric KS also relies on flattening the data but does show that t-EDM outperforms EDM. In any case, I believe a more throrough justification or explanation is needed here.

Also see questions regarding preconditioning for EDM vs t-EDM. If these questions are successfully addressed I will raise my score.

Smaller comments:
* in line 198 there is a typo, the equal is in the subscript and it should not.

**Questions:**

Questions
* Since the preconditioning for EDM helps a good amount for unconditional generation (see Table 2), is not there some preconditioning for EDM in a similar spirit to the unconditional case that could help for the conditional generation case?
* As mentioned in strengths the t-EDM has a free parameter $\nu$ that can be tuned to predict more accurately the heavy-tail behavior for different datasets. However, the INC preconditioning for the EDM could be given a parameter by reparameterizing into a normal with variance $\sigma^2$ instead of a standard normal. (More precisely, this means that in step 4 in Baseline 2 in Appendix C.1.2. we would replace 'standard Normal distribution' to centered normal distribution with variance $\sigma^2.$) This change in variance is qualitatively different from a change in decay (which is what happens when we vary $\nu$) so this approach may not be theoretically justified, but it may work in practice. The PCP preconditioning already has parameters. Is there any claim for INC/PCP that can be made about whether tuning this parameters helps or not in capturing different type of heavy-tail distributions?

In summary, I am wondering if the performance of EDM can be increased only by preconditioning to reach that of t-EDM or if t-EDM does provide a significant difference not achievable by preconditioning.

---

> ### Author Response · Authors · 2024-11-23
> **Official Author Response - Reviewer uzb2**
>
> We thank the reviewer for insightful comments and address each point in more detail below.
>
> **Weaknesses/Questions**
>
> **Weakness 1***: The experiments for unconditional generation in Table 2 show that the EDM (using preconditioning) outperforms the t-EDM for the KR and SR metric in the VIL channel with test data. This hinders the claim the authors make that standard diffusion models, even with preconditioning, fail to capture the heavy-tailed distributions (line 043) and that t-EDM outperforms standard diffusion models. Qualitatively, the plots in Figure 3 do show that t-EDM outperforms EDM with preconditioning in capturing the heavy-tailed behavior. However, I believe a justification of why the KR and SR metrics do not reflect this should be included. As addressed in Appendix F.2, this metrics rely on flattening the data. This may be one reason, but the metric KS also relies on flattening the data but does show that t-EDM outperforms EDM. In any case, I believe a more throrough justification or explanation is needed here.*
>
> **Response**: This is an interesting observation. We elaborate upon this aspect as follows.
>
> Firstly, another reason for our method exhibiting better performance for the KS metric could be the fact that the KS metric, in addition to flattening the data, is computed over 99.9th and 0.1th percentiles for the right and left tails, respectively (see Appendix C.1.3). This implies that the KS metric specifically examines the tail values and thus represents an appropriate metric to capture the difference between the tail estimation for the generated and the data samples. In contrast, the KR and SR metrics are based on computing higher-order moments over the entire domain of flattened empirical samples and thus might be more coarse in nature, hence the discrepancy.
>
> However, it is worth noting that while we present EDM baselines with improved preconditioning and preprocessing as additional baselines, these can also be applied to t-EDM, potentially improving results. Therefore, a more fair comparison should be between t-EDM and the baseline EDM for which t-EDM outperforms EDM by a significant margin in all metrics. We also clarify this in the main text in Section 4.1 (lines 406-411).
>
> Secondly, due to the flattening operation, the metrics considered in this work disregard the spatial structure of the generated samples, which is undesirable. The qualitative histogram plots in Figure 3 help alleviate this to some extent. However, designing metrics that can quantify heavy-tailed behavior while considering the spatial structure of the generated samples is an interesting direction for further work but is outside the scope of this paper. We discuss these limitations in more detail in Appendix F.2.

---

> ### Author Response · Authors · 2024-11-23
> **Official Author Response - Reviewer uzb2 (Contd.)**
>
> **Question 1:** *In summary, I am wondering if the performance of EDM can be increased only by preconditioning to reach that of t-EDM or if t-EDM does provide a significant difference not achievable by preconditioning.*
>
> **Response**: This is a valid question and, in fact, was also a motivation for us to explore the PCP and INC techniques as comparison baselines in the first place. While the possible design space for constructing more efficient preconditioning and preprocessing techniques is quite large, and it is not feasible to experiment with a lot of these different combinations, several observations justify the use of heavy-tailed diffusion models for modeling extreme events rather than relying solely on preconditioner like heuristics for improved modeling performance:
>
> 1. **Theoretical arguments**: [1] show that a class of Lipschitz-continuous transport maps acting on the source distribution leads to target distributions with tail properties similar to the source distribution. Therefore, a diffusion/flow model trained using a Gaussian base distribution (which is light-tailed) is unlikely to capture a target heavy-tailed distribution. Moreover, unlike a heavy-tailed prior, a light-tailed distribution like a Gaussian prior lacks the capacity to encode usual and extreme events [2]. Similarly, [3] argues that standard denoising score matching fails to capture heavy tails due to spherical Gaussian shells in high dimensions. Therefore, the inability of Gaussian diffusion models to capture tail events is likely a more fundamental modeling issue rather than one that can be resolved by updating preconditioners (PCP) or data preprocessing schemes (INC).
> 2. **Empirical arguments**: The theoretical arguments presented above are also supported by empirical results. For unconditional generation, we experimented with different PCP schemes for modulating the $\pi_\text{mean}$ parameter for controlling the noise schedule in EDM training and stuck with the RBF kernel due to better performance. However, we also observed that heuristics like PCP do not scale when training on data with multiple channels, with performance degrading rapidly (in terms of qualitative histograms) as opposed to a single-channel setting.
>
>     As an example of this observation, we trained an EDM model on HRRR data with 10 channel inputs (as opposed to single-channel training presented in the main text). We compare qualitative histograms between the 10-channel and single-channel settings for the VIL and w20 channels in Figure 7 in Appendix C.1.7. It can be seen clearly that while PCP might prove to be effective on single-channel training (possibly due to easier modeling), on multi-channel runs, modeling tails takes a severe degradation. Therefore, under such scenarios, PCP/INC can only help so much, and t-EDM can possibly provide a more principled alternative (provided there is a principled mechanism to tune the hyperparameter nu) to model heavy tail scenarios. (We include full experimental details for our 10-channel run in Appendix C.1.7).
>
>
>     Moreover, while heuristics like PCP/INC are illustrated in the main text as competing baselines, it is worth noting that they can also be used under the t-EDM framework and should be considered complementary to our proposed heavy-tailed diffusion framework for more challenging modeling tasks. Below, we also provide a more detailed response to the individual questions posed by the reviewer.
>
>
> [1] Tails of Lipschitz triangular flows, Jaini et al.
>
> [2] $ t^ 3$-Variational Autoencoder: Learning Heavy-tailed Data with Student's t and Power Divergence, Kim et al.
>
> [3] Heavy-tailed denoising score matching, Deasy et al.

---

> > ### Author Response · Authors · 2024-11-23
> > **Official Author Response - Reviewer uzb2 (Contd.)**
> >
> > **Question 2:** *Since the preconditioning for EDM helps a good amount for unconditional generation (see Table 2), is not there some preconditioning for EDM in a similar spirit to the unconditional case that could help for the conditional generation case?*
> >
> > **Response:** We assume that the reviewer refers to the per-channel preconditioning (PCP) scheme here for unconditional generation. In principle, a similar PCP scheme (as in the unconditional case) could also help the conditional synthesis results. However, in this work, the results for conditional synthesis are more akin to a proof of concept rather than a more detailed comparison with different baselines commonly used in weather forecasting. In this context, we think the comparison between t-EDM and EDM presented in the main text (Table 4) is natural and fair (since both EDM and t-EDM can benefit from techniques like improved preconditioning, albeit with some tuning). Moreover, such a comparison is probably also desirable since the optimal set of parameters for preconditioning will likely be different for both EDM and t-EDM, adding another level of complexity to our evaluations. Therefore, we believe that combining techniques like improved preconditioning (PCP) and preprocessing (INC) with our heavy-tailed diffusion framework for downstream tasks like forecasting deserves a more detailed analysis, which we do not include in this work due to space and time constraints and leave as an important direction for future work. We highlight this point in Appendix F.2 in the revised paper.
> >
> > **Question 3:** *As mentioned in strengths the t-EDM has a free parameter ν  that can be tuned to predict more accurately the heavy-tail behavior for different datasets. However, the INC preconditioning for the EDM could be given a parameter by reparameterizing into a normal with variance σ2 instead of a standard normal. (More precisely, this means that in step 4 in Baseline 2 in Appendix C.1.2. we would replace 'standard Normal distribution' to centered normal distribution with variance σ2.) This change in variance is qualitatively different from a change in decay (which is what happens when we vary ν) so this approach may not be theoretically justified, but it may work in practice. The PCP preconditioning already has parameters. Is there any claim for INC/PCP that can be made about whether tuning this parameters helps or not in capturing different type of heavy-tail distributions?*
> >
> > **Response:** We thank the reviewer for their suggestion on improving the INC scheme by including a tunable variance parameter. Kindly refer to our response to Question 1 regarding whether specific claims can be made on the efficacy of the preconditioning or the preprocessing schemes.
> >
> > **Minor Comments**
> >
> > **Comment 1:** *in line 198 there is a typo, the equal is in the subscript and it should not.*
> >
> > **Response:** Thanks for pointing this out. We have fixed this issue in our revision.
> >
> > We thank the reviewer for their constructive suggestions (which helped improve our revision) and willingness to reconsider their evaluation. We look forward to the reviewer's response and would be happy to respond to additional questions/comments.

---

> > > ### Author Response · Authors · 2024-11-27
> > > **Revision Upload Deadline**
> > >
> > > Dear Reviewer uzb2,
> > >
> > > Since the deadline for uploading a revision is coming up, it would be great if you could acknowledge our response and let us know if further clarifications on any of your questions is required. Thanks for your help in the review process.
> > >
> > > Best,
> > >
> > > Authors

---

> > > > ### Comment · Reviewer_uzb2 · 2024-12-02
> > > >
> > > > Several of my concerns were addressed. The comparisons between EDM with preconditioning and t-EDM with/without preconditioning are still not fully solved and left as future work by the authors, although they provide partial progress towards them.
> > > >
> > > > I keep my positive score of 6.

---

### Comment · Area_Chair_Pqma · 2024-11-25
**Reviewers' Response**

Dear Reviewers,

As the author-reviewer discussion period is approaching its end, I would strongly encourage you to read the authors' responses and acknowledge them, while also checking if your questions/concerns have been appropriately addressed.

This is a crucial step, as it ensures that both reviewers and authors are on the same page, and it also helps us to put your recommendation in perspective.

Thank you again for your time and expertise.

Best,

AC

---

### Meta-Review · Area_Chair_Pqma · 2024-12-20

**Metareview:**

This paper introduces t-EDM, a diffusion model framework that enhances the modeling of heavy-tailed distributions, particularly relevant for applications in extreme-weather prediction.  Building upon the existing EDM framework, t-EDM replaces the Gaussian noise typically used in diffusion models with a Student-t distribution, allowing for controllable tail generation via a single hyperparameter.  This change can be implemented with minimal modifications to existing EDM implementations. The paper also introduces a new training method based on power divergence to fit the model. Empirical evaluations on the HRRR weather dataset show that t-EDM outperforms standard Gaussian-based diffusion models, including EDM, in capturing and generating extreme values, making it a valuable contribution to the field of generative modeling, especially for data with heavy-tailed characteristics.

In general the reviewers found the paper to be well-written, while providing a mathematically justified extension to diffusion models. The numerical methods back up the claims with some caveats raised by the reviewers (preconditioned versus non-preconditioned diffusion models). The implementation of the methods is quite straight forward, the mathematical ration is clear. Even though some of the more rigorous mathematical justifications are a bit wobbly, the numerical experiments are deemed sufficient. As such I recommend acceptance.

**Additional Comments On Reviewer Discussion:**

With the exception of reviewer e3KL, most of the reviewers engaged in fairly fruitful discussion (from which I actually learned several new references). Unfortunately, reviewer e3KL, who raised many good theoretical points, didn't respond to the rebuttal, so it is not clear if their points were addressed, although from my perspective, some of the criticism remain.

---

### Decision · Program_Chairs · 2025-01-22

Accept (Poster)